# OPTIMIZATION FOR NEURAL OPERATOR LEARNING: WIDER NETWORKS ARE BETTER

## ABSTRACT

Neural Operators, such as Deep Operator Networks (DONs) (Lu et al., 2021) and Fourier Neural Operators (FNOs) (Li et al., 2021a), that directly learn mappings between function spaces have received considerable recent attention. Despite the universal approximation guarantees for DONs (Lu et al., 2021; Chen & Chen, 1995) and FNOs (Kovachki et al., 2021), there is currently no optimization convergence guarantee for learning such networks using gradient descent (GD). In this paper, we present a unified framework for optimization based on GD and apply the framework to DONs and FNOs, establishing convergence guarantees for both. In particular, we show that as long two conditions—restricted strong convexity (RSC) and smoothness—are satisfied by the loss, GD is guaranteed to decrease the loss geometrically. Subsequently, we show that the two conditions are indeed satisfied by the DON and FNO losses, but because of rather different reasons that arise as a result of differences in the structure of the respective models. One takeaway that emerges is that wider networks lead to better optimization convergence for both DONs and FNOs. We present empirical results on several canonical operator learning problems to show that wider DONs and FNOs lead to lower training losses, thereby supporting the theoretical results.

## 1 INTRODUCTION

Replicating the success of deep learning in scientific computing such as developing neural PDE solvers, constructing surrogate models, and developing hybrid numerical solvers, has recently captured interest of the broader scientific community. In relevant applications to scientific computing, we often need to learn mappings between function spaces. Neural Operators have emerged as the prominent class of deep learning models used to learn such mappings. While there have been a plethora of attempts, the two most widely adopted neural operators are the Fourier Neural Operators (FNOs) (Li et al., 2021a;b) and Deep Operator Networks (DONs) (Lu et al., 2021; Wang et al., 2021). The fundamental idea of a neural operator is to parameterize these mappings as a deep neural network and proceed with its learning—also known as its optimization or training—as in a standard supervised learning setup. However, contrary to a classical supervised learning setting where we learn mappings between two finite-dimensional vector spaces, here we learn mappings between mappings between *infinite-dimensional* function spaces.

Since a neural operator directly learns the mapping between the input and output function spaces (Lu et al., 2021), it is a natural choice for learning solution operators of (i) parametric PDEs where the PDE solution needs to be inferred for multiple combinations of these "input parameters" or (ii) inverse problems where the forward problem needs to be solved multiple times to optimize a given functional. While there exist results on the universal approximation properties of neural operators; see, e.g., Deng et al. (2021); Kovachki et al. (2021) for universal approximation results of DONs and FNOs, there does not exist any optimization result on when and why gradient descent (GD) converges during the optimization of these Neural Operators.

In this paper, we establish convergence guarantees for GD for learning DONs and FNOs. We first present two conditions for the convergence of GD on neural operator learning and show that as long as these two conditions are satisfied by a loss function, GD will decrease the loss in every iteration. One of the conditions is based on restricted strong convexity (RSC) on a non-empty set $Q^t$, a recently introduced Banerjee et al. (2023) alternative to the widely used NTK (neural tangent kernel) based

analysis Liu et al. (2021a; 2022b); Allen-Zhu et al. (2019). The key novelty and asscoiated heavy lifting in our work is on showing that DONs and FNOs in fact satisfy these conditions for over-parameterized wide networks, though the analyses for DONs and FNOs are substantially different, and need to consider specifics of how these models are structured. Our results are the first of its kind to show GD convergence on DONs and FNOs, that too using a unified analysis, and the first to theoretically show the benefits of width in these popular neural operators. To complement our theoretical results, we present empirical evaluation of our guarantees and benefits of width on both DONs and FNOs on a set of popular operator learning problems, including antiderivative, diffusion-reaction, and Burger's equation.

The rest of the paper is organized as follows. We briefly review related literature in Section 2 and present specifics on DONs and FNOs in Section 3. In Section 4, we present technical conditions under which GD optimization guarantees can be established for learning neural operators and show that these conditions are indeed satisfied by DONs and FNOs respectively in Section 5 and 6. We present empirical results in Section 7, with additional results and proofs in the Appendix.

## 2    RELATED WORK

**Learning Operators.** Constructing operator networks for ordinary differential equations (ODEs) using learning-based approaches was first studied in Chen & Chen (1995) where the authors showed that a neural network with a single hidden layer can approximate *a nonlinear continuous functional* to arbitrary accuracy. This was, in essence, akin to the Universal Approximation Theorem for classical neural networks (see, e.g., Cybenko (1989); Hornik et al. (1989); Hornik (1991); Lu et al. (2017)). While the theorem only guaranteed the existence of a neural architecture, it was not practically realized until Lu et al. (2021) provided an extension of the theorem to deep networks. Since then a number of works have pursued applications of DONs to different problems (e.g., see Goswami et al. (2022); Wang et al. (2021); Wang & Perdikaris (2021)). The operator learning paradigm has also been explored in parallel by a number of other works, most notably Bhattacharya et al. (2021b;a); Li et al. (2021a; 2020b; 2021b) which seek to directly parameterize the integral kernel in the Fourier space using a deep network. A number of subsequent extensions that explore different architectures tailored to different problems have been proposed in Li et al. (2020a); Liu et al. (2022a); Wen et al. (2022); Pathak et al. (2022). Recently Kontolati et al. (2022) studied the influence of over-parameterization on neural surrogates based on DONs in the context of dynamical systems. While their paper studies the effects of over-parameterization on the generalization properties of DONs, an optimization analysis of DONs is a largely open problem. Similarly, an optimization analysis of FNOs has not been pursued to the best of our knowledge.

**Optimization Analysis of Neural Networks.** Optimization of over-parameterized deep networks have been studied extensively (see, e.g., Du et al. (2019); Arora et al. (2019b;a); Allen-Zhu et al. (2019); Liu et al. (2021a)). In particular, Jacot et al. (2018) showed that the neural tangent kernel (NTK) of a deep network converges to an explicit kernel in the limit of infinite network width and stays constant during training. Liu et al. (2021a) showed that this constancy arises due to the scaling properties of the Hessian of the predictor as a function of network width. Du et al. (2019); Allen-Zhu et al. (2019) showed that gradient descent converges to zero training error in polynomial time for a deep over-parameterized model, with Du et al. (2019) showing it for a deep model with residual connections (ResNet) and Allen-Zhu et al. (2019) showing it in the context of feed-forward models, CNNs and ResNets. Karimi et al. (2016) showed that the Polyak-Lojasiewicz (PL) condition, a much weaker condition than strong convexity can be used to explain the linear convergence of gradient-based methods. Banerjee et al. (2023) showed convergence of feedforward networks using restricted strong convexity (RSC) in order to derive a variant of the PL condition, and thus provide convergence guarantees of gradient descent.

## 3    LEARNING NEURAL OPERATORS

In this section, we briefly introduce the DON and FNO aproaches to learning operators to setup some notation. For a more detailed exposition, we refer the reader to Appendix B.

## 3.1 Learning Deep Opertor Networks (DONs)

A DON is an operator network that learns a parametric operator $G_{\boldsymbol{\theta}}$ such that $G_{\boldsymbol{\theta}}(\boldsymbol{u}) \approx G^{\dagger}(\boldsymbol{u})$, where $\boldsymbol{u}$ denotes the input function, and $G^{\dagger}$ denotes the "true" operator. Following Lu et al. (2021), a DON predictor is defined as the inner product of two deep networks: the branch net $\boldsymbol{f} = \{f_k\}_{k=1}^{K}$ and the trunk net $\boldsymbol{g} = \{g_k\}_{k=1}^{K}$, namely

$$G_{\boldsymbol{\theta}}(\boldsymbol{u})(\boldsymbol{y}) := \sum_{k=1}^{K} f_k(\boldsymbol{\theta}_f; \boldsymbol{u}) g_k(\boldsymbol{\theta}_g; \boldsymbol{y}), \tag{1}$$

where $\boldsymbol{u} \in \mathbb{R}^{d_u}$ is the input function and $\boldsymbol{y} \in \mathrm{dom}(G_{\boldsymbol{\theta}}(\boldsymbol{u})) \subseteq \mathbb{R}^{d_y}$ the output location on which the operator will be evaluated[1]. The training data comprises of $n$ input functions $\{\boldsymbol{u}^{(i)}\}_{i=1}^{n}$ and $q_i$ output locations for each $G(u^{(i)})$, i.e., $\{\{\boldsymbol{y}_j^{(i)}\}_{j=1}^{q_i}\}_{i=1}^{n}$ with $\boldsymbol{y}_j^{(i)}$ denoting the $j$-th output location for $G_{\boldsymbol{\theta}}(u^{(i)})$. The input functions $\boldsymbol{u}^{(i)}$ are represented in $R$ locations $\{\boldsymbol{x}_r\}_{r=1}^{R} \in \mathrm{dom}(\boldsymbol{u}) \subseteq \mathbb{R}^{d}$ so that $\boldsymbol{u}^{(i)}(x_r) \in \mathbb{R}^{d_u}, \forall r \in [R]$. For scalar functions $u^{(i)} \in \mathbb{R}$, the branch net takes input $\{u^{(i)}(x_r)\}_{r=1}^{R}$, which implies $\boldsymbol{f} : \mathbb{R}^{d_u} \to \mathbb{R}^{K}$. Similarly, for scalar output locations $y_j^{(i)} \in \mathbb{R}$ we have $\boldsymbol{g} : \mathbb{R} \to \mathbb{R}^{K}$. The branch net $\boldsymbol{f}$ has parameters $\boldsymbol{\theta}_f \in \mathbb{R}^{p_f}$ with the $k^{th}$ output denoted as $\boldsymbol{f}_k(\boldsymbol{\theta}_f; \boldsymbol{u}), k \in [K]$. Similarly, the trunk net $\boldsymbol{g}$ has parameters $\boldsymbol{\theta}_g \in \mathbb{R}^{p_g}$ with the $k^{th}$ output denoted as $\boldsymbol{g}_k(\boldsymbol{\theta}_g; \boldsymbol{y}), k \in [K]$. The entire set of parameters for the DON is given by $\boldsymbol{\theta} = [\boldsymbol{\theta}_f^{\top} \ \boldsymbol{\theta}_g^{\top}]^{\top} \in \mathbb{R}^{p_f + p_g}$. The DON learning problem can then be cast as the minimization of the following empirical risk:

$$\boldsymbol{\theta}_{(\mathrm{don})}^{\dagger} \in \underset{\boldsymbol{\theta} \in \Theta}{\mathrm{argmin}} \ \mathcal{L}\left(G_{\boldsymbol{\theta}}(\boldsymbol{u}), G^{\dagger}(\boldsymbol{u})\right) := \frac{1}{n} \sum_{i=1}^{n} \frac{1}{q_i} \sum_{j=1}^{q_i} \left(G_{\boldsymbol{\theta}}(u^{(i)})(y_j^{(i)}) - G^{\dagger}(u^{(i)})(y_j^{(i)})\right)^2, \tag{2}$$

with

$$G_{\boldsymbol{\theta}}(u^{(i)})(y_j^{(i)}) = \sum_{k=1}^{K} f_k\left(\boldsymbol{\theta}_f; \{u^{(i)}(x_r)\}_{r=1}^{R}\right) g_k\left(\boldsymbol{\theta}_g; y_j^{(i)}\right). \tag{3}$$

Note that the "true" operator $G^{\dagger}$ whose approximation is sought in (2) can either be explicit, e.g. integral of a function, or implicit, e.g. the solution to a nonlinear partial differential equation (PDE).

## 3.2 Learning Fourier Neural Operators (FNOs

Given an input function $\boldsymbol{u}$ and corresponding output $f(\boldsymbol{x}) = G^{\dagger}(\boldsymbol{u})(\boldsymbol{x})$, the FNO learns a parametric map $G_{\boldsymbol{\theta}}$ such that $G_{\boldsymbol{\theta}}(\boldsymbol{u}) \approx G^{\dagger}(\boldsymbol{u})$ where $G^{\dagger}$ denotes the "true" operator. We recall the definition of the FNO model in (Li et al., 2021a), i.e,

$$\boldsymbol{\alpha}^{(0)}(\boldsymbol{x}) = P(\boldsymbol{u}; \boldsymbol{\theta}_p)(\boldsymbol{x}), \quad \boldsymbol{\alpha}^{(l)}(\boldsymbol{x}) = \mathcal{F}^{(l)}(\boldsymbol{\alpha}^{(l-1)}(\boldsymbol{x}); \boldsymbol{\theta}_{F^{(l)}}) \quad \text{and} \quad f(\boldsymbol{x}) = Q(\boldsymbol{\alpha}^{(L+1)}; \boldsymbol{\theta}_q)(\boldsymbol{x}), \tag{4}$$

where $\{\mathcal{F}^{(l)}\}_{l=1}^{L+1}$ are the nonlinear transformations with learnable parameters $\boldsymbol{\theta}_F = [\boldsymbol{\theta}_{F^{(1)}}^{\top}, \dots, \boldsymbol{\theta}_{F^{(l)}}^{\top}]^{\top}$, $P$ denotes an encoder that maps the input function to an ambient space (often higher dimensional), and $Q$ denotes the decoder that maps the output from the last FNO block to the desired output space with parameters $\boldsymbol{\theta}_p$ and $\boldsymbol{\theta}_q$ respectively. The entire set of parameters for the FNO can be written as $\boldsymbol{\theta} = \left[\boldsymbol{\theta}_p^{\top} \ \boldsymbol{\theta}_F^{\top} \ \boldsymbol{\theta}_q^{\top}\right]^{\top}$. Following the approach in (Li et al., 2021a), we write

$$\mathcal{F}^{(l)}\left(\boldsymbol{\alpha}^{(l-1)}(\boldsymbol{x}); \boldsymbol{\theta}_{F^{(l)}}\right) := \phi(W^{(l)} \boldsymbol{\alpha}^{(l-1)} + \left(\mathcal{K}^{(l)}(\boldsymbol{u}; R^{(l)}) \boldsymbol{\alpha}^{(l-1)}\right))(\boldsymbol{x}), \tag{5}$$

where $\phi$ is a pointwise activation, $W^{(l)}$ is an affine transformation, $\mathcal{K}^{(l)}$ denotes the parametric kernel operator with parameters $R^{(l)}$. The kernel can be written as a scalar function as $(\mathcal{K}^{(l)}(\boldsymbol{u}; R^{(l)}) w)(x) := \int_{\mathcal{T}} k(x, y, \boldsymbol{u}(x), \boldsymbol{u}(y); R^{(l)}) w(y) dy$, where $\boldsymbol{\theta}_{\mathcal{F}^{(l)}}$ a set of unknown parameters and $w$ any appropriate function with domain $\mathcal{T}$. With a slight abuse of notation, the FNO applied on the input $\boldsymbol{u}$ can be implicitly written as $\boldsymbol{f}(\boldsymbol{x}) = G_{\boldsymbol{\theta}}(\boldsymbol{u})(\boldsymbol{x})$. Considering $n$ input-output pairs $(\boldsymbol{f}^{(i)})$ and input $(\boldsymbol{u}^{(i)})$ pairs on a computational grid $(\boldsymbol{x}_{r_{r=1}^{R}})$ allows us to write

$$\boldsymbol{f}^{(i)}(\boldsymbol{x}_r) = G_{\boldsymbol{\theta}}(\boldsymbol{u}^{(i)})(\boldsymbol{x}_r), \quad \forall i \in [n], \quad \forall r \in [R]. \tag{6}$$

---

[1]The original DON paper (Lu et al., 2021) puts forth the above model and another one with a bias term added to the inner product. For definiteness, we restrict our attention to the model without bias.

Then, the FNO learning problem can be written as the minimization of the following empirical risk:

$$\boldsymbol{\theta}^{\dagger}_{(\text{fno})} \in \underset{\boldsymbol{\theta} \in \Theta}{\text{argmin}} \; \mathcal{L}(G_{\boldsymbol{\theta}}(\boldsymbol{u}), G^{\dagger}(\boldsymbol{u})) = \frac{1}{n} \sum_{i=1}^{n} \frac{1}{R} \sum_{r=1}^{R} \left( G_{\boldsymbol{\theta}}(\boldsymbol{u}^{(i)})(\boldsymbol{x}_r) - G^{\dagger}(\boldsymbol{u}^{(i)})(\boldsymbol{x}_r) \right)^2 . \quad (7)$$

## 4 OPTIMIZATION CONVERGENCE ON NEURAL OPERATORS MODELS

We now focus on establishing *two conditions* for the convergence of gradient descent (GD) on neural operator (NO) models, in particular showing that as long as the two conditions are satisfied, the loss will decrease geometrically. The development is independent of the type of neural operator under consideration. Then, we show that the required conditions for convergence are satisfied by DONs (Section 5) and by FNOs (Section 6) using the structure and properties specific to these models. For convenience, we will denote the empirical loss of the neural operator model as $\mathcal{L}(\boldsymbol{\theta})$. The specific losses corresponding to DONs and FNOs are in (2) and (7), respectively.

The first condition is based on the concept of Restricted Strong Convexity (RSC).

**Definition 1** (Restricted Strong Convexity (RSC)). *A function $\mathcal{L}$ is said to satisfy $\alpha$-restricted strong convexity ($\alpha$-RSC) w.r.t. the tuple $(B, \boldsymbol{\theta})$ if for any $\boldsymbol{\theta}' \in B \subseteq \mathbb{R}^p$ and some fixed $\boldsymbol{\theta} \in \mathbb{R}^p$, we have $\mathcal{L}(\boldsymbol{\theta}') \geq \mathcal{L}(\boldsymbol{\theta}) + \langle \boldsymbol{\theta}' - \boldsymbol{\theta}, \nabla_{\boldsymbol{\theta}} \mathcal{L}(\boldsymbol{\theta}) \rangle + \frac{\alpha}{2} \|\boldsymbol{\theta}' - \boldsymbol{\theta}\|_2^2$, with $\alpha > 0$.*

Let $\boldsymbol{\theta}_0$ denote a suitable (random) initialization and $\{\boldsymbol{\theta}_t\}_{t \geq 1}$ denote the sequence of iterates obtained from GD on loss $\mathcal{L}(\boldsymbol{\theta})$, i.e.,

$$\boldsymbol{\theta}_{t+1} = \boldsymbol{\theta}_t - \eta_t \nabla_{\boldsymbol{\theta}} \mathcal{L}(\boldsymbol{\theta}_t) . \quad (8)$$

Under suitable assumptions, the iterates stay within a suitable ball $B_{\rho}^{\text{Euc}}(\boldsymbol{\theta}_0)$ around the initialization that will be individually specified for DONs and FNOs. The first conditions of interest stipulates that at step $t$, the loss $\mathcal{L}$ satisfies $\alpha$-RSC.

**Condition 1** (RSC). *At step $t$, there exists a set $Q^t \subseteq \mathbb{R}^p$ such that*

(a) *the set $B^t := Q^t \cap B_{\rho}^{\text{Euc}}(\boldsymbol{\theta}_0) \cap B_{\rho_2}^{\text{Euc}}(\boldsymbol{\theta}_t)$ is non-empty for some suitable constant radii $\rho, \rho_2 > 0$; and*

(b) *the loss function $\mathcal{L}$ satisfies $\alpha_t$-RSC w.r.t. $(B_t, \boldsymbol{\theta}_t)$ for some $\alpha_t > 0$.*

Note that $\mathcal{L}$ need not be convex for it to satisfy $\alpha_t$-RSC. In essence, when the iterate is $\boldsymbol{\theta}_t$, the loss needs to be strongly convex on a suitable set $B_t$. The analysis for establishing that such a non-empty $B_t$ exists are substantially different for DONs and FNOs, and takes more care DONs (Section 5) as it involves two different networks. On the other hand, the analysis for establishing the $\alpha_t$-RSC takes considerably more care for FNOs (Section 6) as it involves Fourier transforms which are not there in typical feedforward networks. The second condition stipulates that the loss $\mathcal{L}$ is $\beta$-smooth.

**Condition 2** (Smoothness). *The loss is $\beta$-smooth, i.e., for $\boldsymbol{\theta}', \boldsymbol{\theta} \in \mathcal{N}(\boldsymbol{\theta}_0)$ and some $\beta = O(1)$, $\mathcal{L}(\boldsymbol{\theta}') \leq \mathcal{L}(\boldsymbol{\theta}) + \langle \boldsymbol{\theta}' - \boldsymbol{\theta}, \nabla_{\boldsymbol{\theta}} \mathcal{L}(\boldsymbol{\theta}) \rangle + \frac{\beta}{2} \|\boldsymbol{\theta}' - \boldsymbol{\theta}\|_2^2$.*

A form of smoothness is utilized for most existing analysis (Allen-Zhu et al., 2019; Banerjee et al., 2023). We work with smooth activation functions, which makes the smoothness condition easier to establish, but we note that similar conditions are usually established and used for ReLU networks as well Allen-Zhu et al. (2019). As long as the two conditions are satisfied at step $t$ of the GD update in (8), the loss is guaranteed to decrease with a suitable (constant) step-size choice.

**Theorem 1** (**Global Loss Reduction**). *Assume the loss $\mathcal{L}$ satisfies Conditions 1 and 2 with $\alpha_t \leq \beta$ at step $t$ of the GD update as in (8) with step-size $\eta_t = \frac{\omega_t}{\beta}$ for some $\omega_t \in (0, 2)$. Then, $\forall \bar{\boldsymbol{\theta}} \in \underset{\boldsymbol{\theta} \in B_{\rho}^{\text{Euc}}(\boldsymbol{\theta}_0)}{\text{arginf}} \; \mathcal{L}(\boldsymbol{\theta})$ and $\bar{\boldsymbol{\theta}}_{t+1} \in \underset{\boldsymbol{\theta} \in Q_{\kappa}^t \cap B_{\rho}^{\text{Euc}}(\boldsymbol{\theta}_0)}{\text{arginf}} \; \mathcal{L}(\boldsymbol{\theta})$ with $0 \leq \gamma_t := \frac{\mathcal{L}(\bar{\boldsymbol{\theta}}_{t+1}) - \mathcal{L}(\bar{\boldsymbol{\theta}})}{\mathcal{L}(\boldsymbol{\theta}_t) - \mathcal{L}(\bar{\boldsymbol{\theta}})} < 1$, we have*

$$\mathcal{L}(\boldsymbol{\theta}_{t+1}) - \mathcal{L}(\bar{\boldsymbol{\theta}}) \leq \left( 1 - \frac{\alpha_t \omega_t (1 - \gamma_t)}{\beta} (2 - \omega_t) \right) (\mathcal{L}(\boldsymbol{\theta}_t) - \mathcal{L}(\bar{\boldsymbol{\theta}})) . \quad (9)$$

Our analysis is inspired by recent related advances by Banerjee et al. (2023), where a related analysis was done for basic feedforward networks. We abstract out from that special case, and demonstrate that the analysis works for any losses satisfying Conditions 1 and 2. The heavy lifting for the optimization convergence analyses is then to establish the two conditions for specific models, viz. DONs and FNOs, which we respectively do in the next two sections. While NTK (neural tangent kernel) based analysis has been widely used for convergence analysis (Liu et al., 2021a; 2022b; Allen-Zhu et al., 2019), for wide networks NTK implies RSC, and they are both sufficient conditions for convergence (Banerjee et al., 2023). One can pursue a purely NTK based analysis of convergence–we do not take that route, instead establish convergence based on Theorem 1.

## 5 OPTIMIZATION ANALYSIS FOR DEEPONETS

In this section, we focus on DONs based on smooth activation functions. To build up to the optimization analysis, we first establish a bound on the spectral norm of the DON predictor, in particular showing that $\|\nabla^2 G_{\boldsymbol{\theta}}(\boldsymbol{u})(\boldsymbol{y})\|_2 = O(\frac{1}{\sqrt{m}})$ where, again, $m_f = m_g = m$. The spectral norm bound is then used to establish a form of Restricted Strong Convexity (RSC) of the DON loss (2), which in turn is used to establish geometric convergence of gradient descent (GD). For the analysis, analogous to Liu et al. (2021b), we consider a FNN for the branch net:

$$\alpha_f^{(0)} = \boldsymbol{u}, \quad \alpha_f^{(l)} = \phi_l\left(\frac{1}{\sqrt{m_{\mathrm{f}}}}W_f^{(l)}\alpha_f^{(l-1)}\right), \forall l \in [L-1], \quad f = \alpha_f^{(L)} = \frac{1}{\sqrt{m_{\mathrm{f}}}}W_f^{(L)}\alpha_f^{(L-1)}$$
(10)

where $m_f$ and $L$ denote the width and depth of the branch net respectively, $\phi_l$ is the activation function at layer $l$, $\alpha_f^{(l)}$ are the outputs at layer $l$, and $W_f^{(l)} \equiv w_{f_{ij}}^{(l)}$ denote the weight matrices at layer $l$. Similarly, we consider a fully connected feedforward network for the trunk net:

$$\alpha_g^{(0)} = \boldsymbol{y}, \quad \alpha_g^{(l)} = \phi_l\left(\frac{1}{\sqrt{m_g}}W_g^{(l)}\alpha_g^{(l-1)}\right), \forall l \in [L-1], \quad g = \alpha_g^{(L)} = \frac{1}{\sqrt{m_{\mathrm{g}}}}W_g^{(L)}\alpha_g^{(L-1)}$$
(11)

where, again, $m_g$ and $L$ denote the width and depth of the trunk net respectively and $W_g^{(l)} \equiv w_{g_{ij}}^{(l)}$ denote the weight matrices at layer $l$ of the trunk net. We consider we have $K$ outputs on each of the networks, and so $W_f^{(L)} \in \mathbf{R}^{K \times m_f}$ and $W_g^{(L)} \in \mathbf{R}^{K \times m_g}$.

We denote by $(w_{f,k}^{(L)})^\top$ and $(w_{g,k}^{(L)})^\top$ the $k$th row of the matrices $W_f^{(l)}$ and $W_g^{(l)}$ respectively. We let $\boldsymbol{\theta}_f^{\mathrm{hid}}$ and $\boldsymbol{\theta}_g^{\mathrm{hid}}$ be the vectors obtained by vectorizing all the weight matrices from the hidden layers $W_f^{(l)}$ and $W_g^{(l)}$, $l \in [L-1]$, and stacking them in a single vector respectively.

In order to aid our analysis, we make the following assumptions on the activations, the loss, and the weights:

**Assumption 1** (Activation functions). *The activation functions $\phi_l$ at each layer $l$ are 1-Lipschitz and $\beta_\phi$-smooth (i.e. $\phi'' \le \beta_\phi$) for some $\beta_\phi > 0$.*

**Assumption 2** (Initialization of Weights). *All weights of the branch and trunk nets are initialized independently as follows: (i) $w_{f_0, ij}^{(l)} \sim \mathcal{N}(0, \sigma_{f,0}^2)$ and $w_{g_0, ij}^{(l)} \sim \mathcal{N}(0, \sigma_{g,0}^2)$ for $l \in [L-1]$ where $\sigma_{f,0} = \frac{\sigma_0}{2(1+\frac{\sqrt{\log m_f}}{\sqrt{2m_f}})}$ and $\sigma_{g,0} = \frac{\sigma_0}{2(1+\frac{\sqrt{\log m_g}}{\sqrt{2m_g}})}$, $\sigma_0 > 0$; (ii) $w_{f_0, k}^{(L)}$ and $w_{g_0, k}^{(L)}$, $k \in [K]$, are random unit vectors with $\|w_{f_0, k}^{(L)}\|_2 = 1$ and $\|w_{g_0, k}^{(L)}\|_2 = 1$ respectively. Further, we assume the input data satisfies $\|\boldsymbol{u}\|_2 = \sqrt{d_u}$ and $\|\boldsymbol{y}\|_2 = \sqrt{d_y}$.*

We also introduce the neighborhood set $B_{\rho,\rho_1}^{\mathrm{Euc}}(\bar{\boldsymbol{\theta}}) = \{\boldsymbol{\theta} \in \mathbb{R}^{p_f + p_g} : \|\boldsymbol{\theta}_f^{\mathrm{hid}} - \bar{\boldsymbol{\theta}}_f^{\mathrm{hid}}\| \le \rho, \|\boldsymbol{\theta}_g^{\mathrm{hid}} - \bar{\boldsymbol{\theta}}_g^{\mathrm{hid}}\| \le \rho, \|w_{f,k}^{(L)} - \bar{w}_{f,k}^{(L)}\|_2 \le \rho_1, \|w_{g,k}^{(L)} - \bar{w}_{g,k}^{(L)}\|_2 \le \rho_1, k \in [K]\}$. We focus on showing that the two conditions needed for convergence of GD as discussed in Section 4 are satisfied by DONs. We start with the definition of the restricted set $Q_\kappa^t$ as in Condition 1 for $\alpha$-RSC, parameterized by some $\kappa \in (0, \frac{1}{2}]$. Due to the involvement and interaction of two neural networks, the branch and trunk

networks, the definition of $Q_\kappa^t$ looks seemingly involved. However, note that $Q_\kappa^t$ is only needed for establishing the $\alpha$-RSC condition for the analysis, and does not change the computational algorithm, which is simply GD run over all the branch and trunk network parameters.

**Definition 2** ($Q_\kappa^t$ **sets for DONs).** *For an iterate* $\boldsymbol{\theta}_t = [\boldsymbol{\theta}_{f,t}^\top \ \boldsymbol{\theta}_{g,t}^\top]^\top$, *consider the singular value decomposition* $\frac{1}{n}\sum_{i=1}^n \frac{1}{q_i}\sum_{j=1}^{q_i}\ell'_{i,j}\sum_{k=1}^K \nabla_{\boldsymbol{\theta}_f}f_k^{(i)}\nabla_{\boldsymbol{\theta}_g}g_{k,j}^{(i)}{}^\top = \sum_{h=1}^{\tilde{q}}\sigma_h \boldsymbol{a}_h \boldsymbol{b}_h^\top$, *where* $\tilde{q} \leq qk$ *with* $q = \sum_{i=1}^n q_i$, *and* $\sigma_h > 0, \boldsymbol{a}_h \in \mathbb{R}^{p_f}, \boldsymbol{b}_h \in \mathbb{R}^{p_g}$ *respectively denote the singular values, left singular vectors, and right singular vectors. Further, let* $\bar{G}_{\boldsymbol{\theta}} = \frac{1}{n}\sum_{i=1}^n\frac{1}{q_i}\sum_{j=1}^{q_i}G_{\boldsymbol{\theta}}(u^{(i)})(y_j^{(i)})$. *Then, for a suitable* $\kappa \in (0, \frac{1}{\sqrt{2}}]$, *we define the set:*

$$Q_\kappa^t := \left\{\boldsymbol{\theta}' = [\boldsymbol{\theta}'_f{}^\top \ \boldsymbol{\theta}'_g{}^\top]^\top \ : \ |\cos(\boldsymbol{\theta}'-\boldsymbol{\theta}_t, \nabla_{\boldsymbol{\theta}}\bar{G}_{\boldsymbol{\theta}_t})| \geq \kappa, \sum_{h=1}^{\tilde{q}}\sigma_h\langle\boldsymbol{\theta}'_f - \boldsymbol{\theta}_{f,t}, \boldsymbol{a}_h\rangle\langle\boldsymbol{\theta}'_g - \boldsymbol{\theta}_{g,t}, \boldsymbol{b}_h\rangle \geq 0\right\}. \quad (12)$$

We now show that $\alpha$-RSC (Condition 1) and smoothness (Condition 2) are satisfied by the DON loss with high probability, which implies that GD will lead to geometric decrease in the loss.

**Theorem 2** (**RSC**). *Under Assumptions 1 and 2 and* $Q_\kappa^t$ *as in Definition 2, (a)* $B_\kappa^t := Q_\kappa^t \cap B_{\rho,\rho_1}^{\mathrm{Euc}}(\boldsymbol{\theta}_0) \cap B_{\rho_2}^{\mathrm{Euc}}(\boldsymbol{\theta}_t)$ *is non-empty for suitable* $\rho, \rho_2 = O(1)$, *and (b) with probability at least* $1 - \frac{4L}{m}$, *at step* $t$ *of GD,* $\forall \boldsymbol{\theta}' \in B_\kappa^t$, *the DON loss* $\mathcal{L}$ *satisfies*

$$\alpha_t = c_1\|\nabla_{\boldsymbol{\theta}}\bar{G}_t\|_2^2 - \frac{c_2}{\sqrt{m}} \ , \quad \text{where} \quad \bar{G}_t = \frac{1}{n}\sum_{i=1}^n\frac{1}{q_i}\sum_{j=1}^{q_i}G_{\boldsymbol{\theta}_t}(u^{(i)})(y_j^{(i)}) \ . \quad (13)$$

*for some constants* $c_1, c_2 > 0$, *where* $c_2$ *depends on the depth* $L$ *and the radii* $\rho, \rho_1, \rho_2$. *Thus, the loss* $\mathcal{L}$ *satisfies RSC w.r.t* $(B_\kappa^t, \boldsymbol{\theta}_t)$ *whenever* $\|\nabla_{\boldsymbol{\theta}}\bar{G}_t\|_2^2 = \Omega(\frac{1}{\sqrt{m}})$.

**Theorem 3** (**Smoothness**). *Under the Assumptions 1 and 2, with probability at least* $1 - \frac{4L}{m}$, *for* $\boldsymbol{\theta} \in B_{\rho,\rho_1}^{\mathrm{Euc}}(\bar{\boldsymbol{\theta}})$, $\mathcal{L}$ *is* $\beta$-*smooth with* $\beta = 4(K\bar{\lambda}^2 + \tilde{c})(\frac{\bar{\lambda}c}{\sqrt{m}} + \varrho) + 2K^2\bar{\lambda}^2\varrho^2$ *with* $c = \max(c^{(f)}, c^{(g)}), \tilde{c} = \max_{ij}G^\dagger(u^{(i)})(y_j^{(i)}), \varrho = \max(\varrho^{(f)}, \varrho^{(g)}), \bar{\lambda} = \max(\lambda_1, \lambda_2)$ *with* $c^{(f)}, c^{(g)}, \varrho^{(f)}, \varrho^{(g)}, \lambda_1, \lambda_2$ *as in Lemma D.3.*

**Remark 1** (The benefit of over-parameterization for the RSC property). According to (13), $\|\nabla_{\boldsymbol{\theta}}\bar{G}_t\|_2^2 = \Omega(\frac{1}{\sqrt{m}})$ is needed to ensure that $\alpha_t > 0$, i.e., to ensure that the empirical loss satisfies the RSC property at time $t$. As the width $m$ increases (of both branch and trunk networks, since they both have the same width), the quantity $\|\nabla_{\boldsymbol{\theta}}\bar{G}_t\|_2^2$ will be able to attain the RSC property at a lower value. $\qquad\square$

**Remark 2** (Over-parameterization allows for a larger neighborhood around initialization). It can be shown that if choose $\rho < \sqrt{m}$, $\rho_1 = O(\mathrm{poly}(L))$, and the initialization parameter $\sigma_0 < 1 - \frac{\rho}{\sqrt{m}}$, then there is a polynomial dependence on the depth $L$ in all of our results. Moreover, since it is possible to make the radius $\rho$ larger as we increase the over-parameterization, it is possible to enlarge the neighborhood around the initialization point where our guarantees hold. $\qquad\square$

## 6 OPTIMIZATION ANALYSIS FOR FNOS

Complementary to Section 5, in this section establish the required conditions for the convergence of GD for FNOs. We again do so by bounding the spectral norm of the hessian of the FNO predictor, in particular showing that $\|\nabla_{\boldsymbol{\theta}}^2 G_{\boldsymbol{\theta}}(\boldsymbol{u})(\boldsymbol{x})\|_2 = O(\frac{1}{\sqrt{m}})$. This is then used to establish the $\alpha$-RSC of the FNO loss (7), which in turn can be used to establish the geometric convergence of gradient descent (GD). Our analysis is inspired by, and borrow ideas from, (Banerjee et al., 2023) and (Liu et al., 2021a). To this end, recall the FNO model

$$\alpha^{(l)} = \phi\left(\frac{1}{\sqrt{m}}W^{(l)}\alpha^{(l-1)} + \frac{1}{\sqrt{m}}F^*R^{(l)}F\alpha^{(l-1)}\right), \quad l \in [L+1]$$

$$f = \alpha^{(L+2)} := \frac{1}{\sqrt{m}}\mathbf{v}^T\alpha^{(L+1)} \ ,$$

where $W^{(l)}$ and $R^{(l)}$ denote the parameters at layer $l \in [L+1]$ with $w_{ij}^{(l)}$ and $r_{ij}^{(l)}$ denoting their respective entries. We denote entire set of trainable parameters of the FNO by $\boldsymbol{\theta}$, where $\boldsymbol{\theta} = [\boldsymbol{\theta}_w^\top \, \boldsymbol{\theta}_r^\top]^\top$ and $\boldsymbol{\theta}_w = [\text{vec}(W^{(1)})^\top, \ldots, \text{vec}(W^{(L+1)})^\top \, \mathbf{v}^\top]^\top$ and $\boldsymbol{\theta}_r = [\text{vec}(R^{(1)}), \ldots, \text{vec}(R^{(L+1)})^\top]^\top$. We denote the number of parameters by $p_w + p_r$, where $p_w = \dim(\boldsymbol{\theta}_w)$ and $p_r = \dim(\boldsymbol{\theta}_r)$. We let $\boldsymbol{\theta}^{\text{hid}}$ be the vector obtained by vectorizing all the weight matrices from the hidden layers $W^{(l)}$, $l \in [L+1]$, and stacking them in a single vector. Furthermore, let the parameters be initialized at $W_0^{(l)}$ and $R_0^{(l)}$, i.e. their entries are initialized at $w_{0,ij}^{(l)}$ and $r_{0,ij}^{(l)}$ respectively. Furthermore, let $\boldsymbol{\theta}_0$ denote the parameters at initialization and $\boldsymbol{\theta}_t$ denote the parameters at an intermediate step $t$.

**Assumption 3** (Activation functions). *The activation functions of the FNO ($\phi_l$) at each layer $l$ are 1-Lipschitz and $\beta_\phi$-smooth (i.e. $\phi'' \leq \beta_\phi$) for some $\beta_\phi > 0$.*

**Assumption 4** (Initialization of Weights). *All weights of the FNO are initialized independently as follows: (i) $w_{0,ij}^{(l)} \sim \mathcal{N}(0, \sigma_{0_w}^2)$ and $r_{0,ij}^{(l)} \sim \mathcal{N}(0, \sigma_{0_r}^2)$ for $l \in [L+1]$ where $\sigma_{0,w} = \frac{\sigma_{1,w}}{2(1+\frac{\sqrt{\log m}}{\sqrt{2m}})}$ and $\sigma_{0,r} = \frac{\sigma_{1,r}}{2(1+\frac{\sqrt{\log m}}{\sqrt{2m}})}$, where $\sigma_{1,w}, \sigma_{1,r} > 0$; (ii) $\mathbf{v}$ is a random unit vector with $\|\mathbf{v}\|_2 = 1$. Further, we assume the input to the network satisfies $\|\alpha^{(0)}\|_2 = \sqrt{d}$.*

We also introduce the neighborhood set $B_{\rho,\rho_1}^{\text{Euc}}(\bar{\boldsymbol{\theta}}) = \{\boldsymbol{\theta} \in \mathbb{R}^{p_w+p_r} \; : \; \|\boldsymbol{\theta}^{\text{hid}} - \bar{\boldsymbol{\theta}}^{\text{hid}}\|_2 \leq \rho, \|\mathbf{v} - \bar{\mathbf{v}}\|_2 \leq \rho_1\}$.

Next, we establish the two conditions required for the convergence of GD. Note that unlike DONs, the $Q_\kappa^t$ sets for FNOs are relatively simpler due to a single feedforward architecture.

**Definition 3** ($Q_\kappa^t$ sets for FNOs). *For iterate $\boldsymbol{\theta}_t \in \mathbb{R}^{p_w+p_r}$, let $\bar{\gamma}_t = \frac{1}{n}\sum_{i=1}^n \frac{1}{R}\sum_{j=1}^R \nabla_{\boldsymbol{\theta}} G_{\boldsymbol{\theta}_t}(u^{(i)})(x_j)$. For $\kappa \in (0,1]$, define $Q_\kappa^t := \{\boldsymbol{\theta} \in \mathbb{R}^{p_w+p_r} \mid |\cos(\boldsymbol{\theta} - \boldsymbol{\theta}_t, \gamma_t)| \geq \kappa\}$.*

**Theorem 4** (RSC). *Given that the activation functions satisfy the smoothness property (Assumption 3), the parameters are initialized as in Assumption 4 and the $Q_\kappa^t$ set chosen as in Definition 3, (a) $B_\kappa^t := Q_\kappa^t \cap B_\rho^{\text{Euc}}(\boldsymbol{\theta}_0) \cap B_{\rho_2}^{\text{Euc}}(\boldsymbol{\theta}_t)$ is non-empty for suitable $\rho, \rho_2 = O(1)$, and (b) with probability at least $(1 - \frac{2(L+2)}{m})$, at step $t$ of GD, the FNO loss $\mathcal{L}$ (7) satisfies $\alpha_t$-RSC w.r.t. $(B_\kappa^t, \theta_t)$ where, for constants $c_1, c_2 > 0$, we have*

$$\alpha_t = c_1 \|\nabla_{\boldsymbol{\theta}} \bar{G}_t\|_2^2 - \frac{c_2}{\sqrt{m}}, \quad where \quad \bar{G}_t = \frac{1}{n}\sum_{i=1}^n \frac{1}{R}\sum_{j=1}^R G_{\boldsymbol{\theta}_t}(u^{(i)})(x_j). \tag{14}$$

*Thus, the loss $\mathcal{L}(\boldsymbol{\theta})$ satisfies RSC w.r.t $(B_\kappa^t, \boldsymbol{\theta}_t)$ whenever $\|\nabla_{\boldsymbol{\theta}} \bar{G}_t\|_2^2 = \Omega(\frac{1}{\sqrt{m}})$.*

**Theorem 5** (Smoothness). *Under Assumptions 3 and 4 with probability at least $\left(1 - \frac{2(L+2)}{m}\right) \; \forall \boldsymbol{\theta}, \boldsymbol{\theta}' \in B_\rho^{\text{Euc}}(\boldsymbol{\theta}_0)$,*

$$\mathcal{L}(\theta') \leq \mathcal{L}(\theta) + \langle \theta' - \theta, \nabla_\theta \mathcal{L}(\theta) \rangle + \frac{\beta}{2}\|\theta' - \theta\|_2^2, \quad with \quad \beta = 2\varrho^2 + \frac{2c_H\sqrt{c_{\rho_1,\gamma}}}{\sqrt{m}} \tag{15}$$

**Lemma 6.1** (Predictor gradient bounds). *Under Assumptions 3 and 4 and for $\boldsymbol{\theta} \in B_\rho^{\text{Euc}}(\boldsymbol{\theta}_0)$ we have*

$$\|\nabla_{\boldsymbol{\theta}} G_{\boldsymbol{\theta}}(\boldsymbol{u})\|_2 \leq \varrho, \tag{16}$$

*where $\varrho^2 = h(L+2)^2$, $h(l) = \gamma^{l-1} + |\phi(0)| \sum_{i=1}^{l-1} \gamma^{i-1}$, and $\gamma = \sigma_1 + \frac{\rho}{\sqrt{m}}$ with $\sigma_1 = \sigma_{1,w} + \sigma_{1,r}$ and $\rho = \rho_w + \rho_r$.*

*Proof.* The proof follows directly from Lemma 4.1 in (Banerjee et al., 2023)

**Remark 3** (The effects of over-parameterization for FNOs). The same observations as in Remark 1 and Remark 2 that show how over-parameterization ensures (i) a better condition for ensuring the RSC property, (ii) a larger neighborhood around the initialization point over which our guarantees hold.

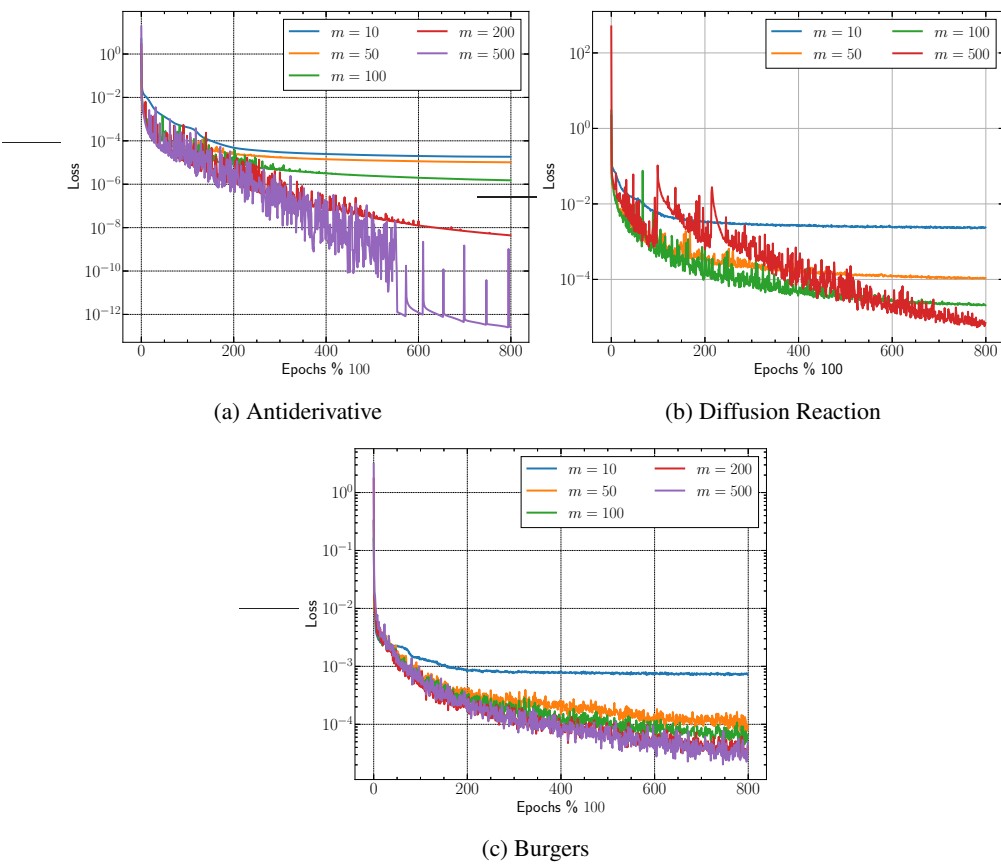

(a) Antiderivative

(b) Diffusion Reaction

(c) Burgers

Figure 1: Training progress of DONs with smooth activations as measured by the MSE loss (2) for (a) Antiderivative Operator, (b) Diffusion-Reaction Equation and (c) Burger's Equation. The y-axis is again plotted on a log-scale to clearly demarcate the effect of increasing width. Increasing the width $m$ again leads to lower training losses.

## 7 EXPERIMENTS

We now turn to a simple empirical evaluation of the effect of over-parameterization on the training performance of DONs and FNOs, as measured by the empirical risk over a mini-batch $B$ of the training dataset. We present empirical findings for three prototypical operator learning problems for DONs: (a) The Antiderivative (or integral) operator, (b) The Diffusion-Reaction Operator and (c) Burgers' Operator. Similarly, for the FNO model presented in Section 6, we present empirical evaluation on two prototypical operator learning problems: (a) Antiderivative Operator, and (b) Burger's equation. For definiteness, we set the width in each layer of the branch and trunk net to be the same (i.e. $m_f = m_g = m$) for the DON and then increase it uniformly from $m = 10$ to $m = 500$. We monitor the training process over $80,000$ training epochs and report the resulting average loss. Similarly, for the FNO, we adopt a similar strategy by increasing the width. Note that the objective of this section is to show the effect of overparameterization on the Neural Operator training and not to present any kind of comparison between the two Neural Operator.

**Remark 4** (Antiderivative Operator). The Antiderivative operator is a linear operator and hence is learned very accurately especially for wider DeepONets ($\mathcal{L}_{\mathcal{D}_B} \sim 10^{-12}$ at the end of training for a DON), and similarly $10^{-5}$ for FNO. □

**Remark 5** (Diffusion Reaction). The Diffusion reaction equation also demonstrates lower loss with increasing width, albeit less markedly than the antiderivative operator. This can be attributed in part to the fact that the operator is inherently nonlinear. □

**Remark 6** (Burger's equation). The operator corresponding to Burger's equation is more intricate with the added periodicity constraints on the solution. While the DON learns the mapping and

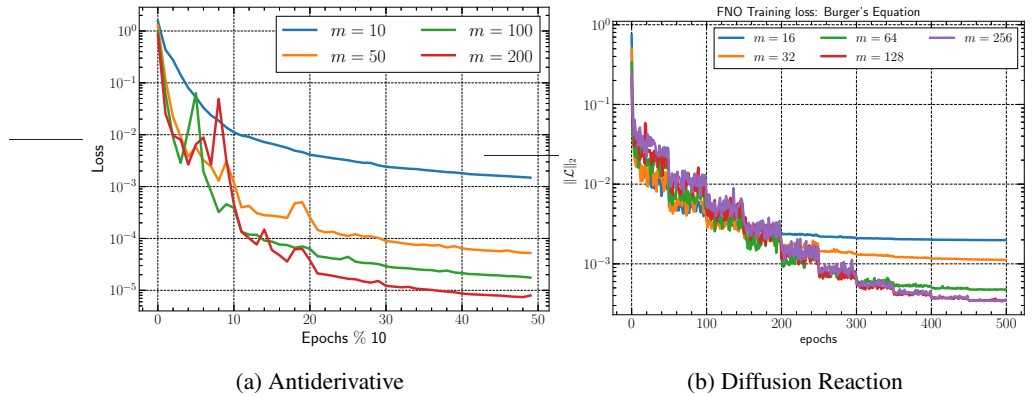

(a) Antiderivative  (b) Diffusion Reaction

Figure 2: Training progress of FNOs with smooth activations as measured by the MSE loss (7) for (a) Antiderivative Operator, (b) Burger's Equation. The y-axis is again plotted on a log-scale to clearly demarcate the effect of increasing width. Increasing the width ($m$) again leads to lower training losses.

outputs the solution over entire solution space $(x, t) \in [0, 1] \times [0, 1]$, the FNO in this case is only aimed at learning the mapping from the input (initial condition $t = 0$) to the final output $t = 1$ and not the entire solution space. $\qquad\square$

## 8    DISCUSSION AND CONCLUSION

We present novel optimization guarantees for the convergence of gradient descent (GD) for overparameterized Neural Operators. We focus on Neural Operators with smooth activations and analyze two popular classes of Neural Operators: (a) Deep Operator Networks (DONs) and (b) Fourier Neural Operators (FNOs), both in their simplest possible architectural configuration, i.e. feedforward networks. For each neural operator, we establish the conditions required for the convergence of GD based on restricted strong convexity (RSC) and smoothness of the loss. Our analysis is first of its kind and provides an encompassing framework to study neural operator optimization. We also present empirical evaluations on several prototypical operator learning problems that complement our theoretical underpinnings showing that wider neural operators lead to overall lower training losses across all the operator learning problems.

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

# A NEURAL OPERATOR INTRODUCTION

## A.1 LEARNING OPERATORS

Here we briefly outline the notion of learning for neural operators Li et al. (2021a; 2020b); Lu et al. (2021). The standard operator learning problem seeks to approximate a possibly nonlinear operator $G^{\dagger} : \mathcal{U} \mapsto \mathcal{V}$ by a parametric operator $G_{\boldsymbol{\theta} \in \Theta} : \mathcal{U} \mapsto \mathcal{V}$ that depends on the learnable parameters $\boldsymbol{\theta}$. The goal is to learn an optimal set of parameters $\boldsymbol{\theta}^{\dagger}$ such that $G_{\theta^{\dagger}} \approx G^{\dagger}$. Given observations $\{u^{(j)}\}_{j=1}^{n} \in \mathcal{U}$ and $\{G^{\dagger}(u^{(j)})\}_{j=1}^{n} \in \mathcal{V}$ where $u^{(j)} \sim \mu$ is an i.i.d sequence from the probability measure $\mu$ supported on $\mathcal{U}$ and $G(u^{(j)})$ is possibly corrupted with noise, the objective is to find $\theta^{\dagger}$ as the solution of the minimization problem

$$\boldsymbol{\theta}^{\dagger} = \operatorname{argmin}_{\boldsymbol{\theta} \in \Theta} \mathbb{E}_{\boldsymbol{u} \sim \mu} \left[ \mathcal{C} \left( G_{\boldsymbol{\theta}}(\boldsymbol{u}), G^{\dagger}(\boldsymbol{u}) \right) \right], \tag{17}$$

where $\mathcal{U}$ and $\mathcal{V}$ are separable Banach spaces and $\mathcal{C}$ a suitable cost functional. This is analogous to the notion of learning in finite dimensions, which is precisely the setup classical deep learning used for.

## A.2 DEEPONET ARCHITECTURE

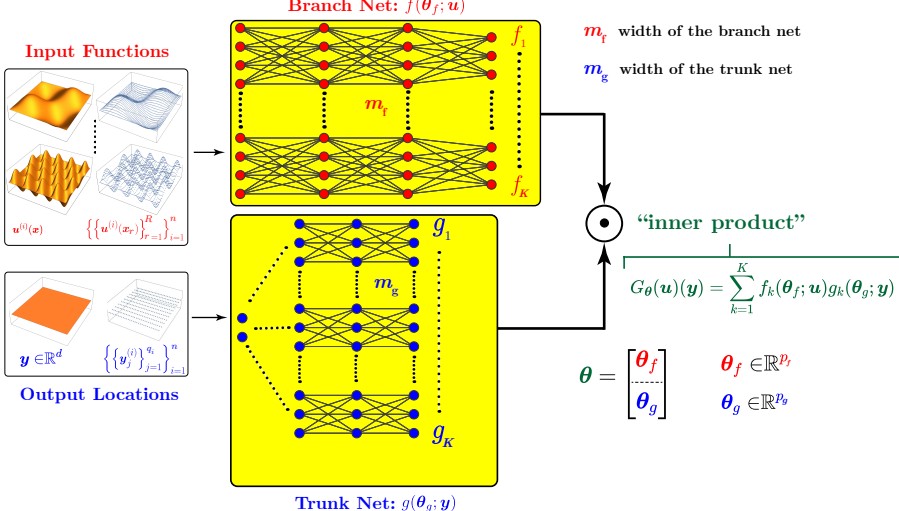

Figure 3: A schematic of the *unstacked* DeepONet architecture Lu et al. (2021) used in this study. Note that the input functions need not be sampled on a structured grid of points in general.

# B LEARNING NEURAL OPERATORS

## B.1 DON TRAINING TUPLE

Each DON training data comprises of the tuple $\mathcal{D}^{(i)} := \left( \{u^{(i)}(x_r)\}_{r=1}^{R}, \{y_j^{(i)}\}_{j=1}^{q_i}, \{G(u^{(i)})(y_j^{(i)})\}_{j=1}^{q_i} \right)$. The total training dataset comprises of all such training tuples $\mathcal{D} = \{\mathcal{D}^{(i)}\}_{i=1}^{n}$.

## B.2 MOTIVATION FOR FNOS

FNOs are closely related to the notion of fundamental solutions. This allows us to write $k_l := k\left(x, y, a(x), a(y); \boldsymbol{\theta}_{\mathcal{F}^{(l)}}\right) := k\left(x - y; \boldsymbol{\theta}_{\mathcal{F}^{(l)}}\right)$ (Li et al., 2021a). Taking the Fourier Transform $(\mathscr{F})$ and applying the convolution theorem gives $\left(\mathcal{K}^{(l)}(a; \boldsymbol{\theta}_{\mathcal{F}^{(l)}}) \alpha^{(l-1)}\right)(x) = \mathscr{F}^{-1}\left(\mathscr{F}(k_l) \cdot \mathscr{F}\left(\alpha^{(l-1)}\right)\right)(x), \forall x \in \mathcal{T}, l \in [L]$.

This helps in parameterizing the *kernel operator* $R^{(l)} = \mathscr{F}(k_l)$ directly in the Fourier space and in simplified notation obtain, $\left(\mathcal{K}^{(l)}\alpha^{(l-1)}\right)(x) = \mathscr{F}^{-1}\left(R^{(l)} \cdot \left(\mathscr{F}\alpha^{(l-1)}\right)\right)(x), \forall x \in \mathcal{T}, l \in [L]$.

Replacing this quantity back in (5), following (Li et al., 2021a), we define each Fourier block as follows

$$\alpha^{(l)}(x) = \phi\left(W^{(l)}\alpha^{(l-1)} + \mathscr{F}^{-1}\left(R^{(l)} \cdot \mathscr{F}\left(\alpha^{(l-1)}\right)\right)\right)(x), \quad x \in \mathcal{T}, \quad l \in [L], \quad (18)$$

where the Fourier transform of the input function $\alpha^{(l-1)}$ is $\mathscr{F}\alpha^{(l-1)}(\xi) := \int_{\mathcal{T}} e^{-2\pi i <\xi,y>} \alpha^{(l-1)}(y)dy, \ y \in \mathcal{T}$. Notice that $R^{(l)}$ is defined by the set of unknown parameters $\boldsymbol{\theta}_{\mathcal{F}^{(l)}}$, whereas $\boldsymbol{\theta}_{F^{(l)}}$ is defined by both the affine operator $W^{(l)}$ and parameters $\boldsymbol{\theta}_{\mathcal{F}^{(l)}}$. We now turn to the discrete version of (18) and the associated architecture.

Since we have a discrete domain, we employ the Discrete Fourier Transform (DFT). The entries of the $m \times m$ DFT kernel ($F$) can be written (up to a suitable scaling) as $F_{kj} := e^{\frac{-2\pi i}{m}(k-1)(j-1)}$, where $k, j \in [m]$ which allows us to write its action on an input vector $\boldsymbol{\alpha}^{(l-1)} \in \mathbb{R}^m$ as

$$v_k := \sum_{j=1}^{m} F_{kj}\alpha_j^{(l-1)} = \alpha_j^{(l-1)} e^{\frac{-2\pi i}{m}(k-1)(j-1)}, \quad (19)$$

where $i^2 = -1$.

## C   OPTIMIZATION CONVERGENCE ANALYSIS, FOR SECTION 4

In this appendix, we establish results for Section 4. In particular, we show that if Condition 1 ($\alpha$-RSC) and Condition 2 (smoothness) are satisfied, GD is guaranteed to geometrically decrease the loss as in Theorem 1. Our analysis follows the recent work of Banerjee et al. (2023), and we provide all proofs here for the sake of completeness.

We start with the following Lemma which shows that Condition 1 implies a form restricted PL condition

**Lemma C.1** (**Restricted PL**). *Assuming Condition 1 is satisfied, i.e., $B_t := Q_\kappa^t \cap \mathcal{N}(\boldsymbol{\theta}_0) \cap B_{\rho_2}^{\text{Euc}}(\boldsymbol{\theta}_t)$ is non-empty and the loss $\mathcal{L}$ satisfies $\alpha_t$-RSC w.r.t. $(B_t, \boldsymbol{\theta}_t)$, then $\mathcal{L}$ satisfies a restricted form of the Polyak-Łojasiewicz (PL) condition w.r.t. $(B_t, \boldsymbol{\theta}_t)$:*

$$\mathcal{L}(\boldsymbol{\theta}_t) - \inf_{\boldsymbol{\theta} \in B_t} \mathcal{L}(\boldsymbol{\theta}) \leq \frac{1}{2\alpha_t}\|\nabla_{\boldsymbol{\theta}}\mathcal{L}(\boldsymbol{\theta}_t)\|_2^2. \quad (20)$$

*Proof.* Define

$$\hat{\mathcal{L}}_{\boldsymbol{\theta}_t}(\boldsymbol{\theta}) := \mathcal{L}(\boldsymbol{\theta}_t) + \langle \boldsymbol{\theta} - \boldsymbol{\theta}_t, \nabla_{\boldsymbol{\theta}}\mathcal{L}(\boldsymbol{\theta}_t) \rangle + \frac{\alpha_t}{2}\|\boldsymbol{\theta} - \boldsymbol{\theta}_t\|_2^2.$$

By Theorem D.4, $\forall \boldsymbol{\theta}' \in B_t$, we have

$$\mathcal{L}(\boldsymbol{\theta}') \geq \hat{\mathcal{L}}_{\boldsymbol{\theta}_t}(\boldsymbol{\theta}'). \quad (21)$$

Further, note that $\hat{\mathcal{L}}_{\boldsymbol{\theta}_t}(\boldsymbol{\theta})$ is minimized at $\hat{\boldsymbol{\theta}}_{t+1} := \boldsymbol{\theta}_t - \nabla_{\boldsymbol{\theta}}\mathcal{L}(\boldsymbol{\theta}_t)/\alpha_t$ and the minimum value is:

$$\inf_{\boldsymbol{\theta}} \hat{\mathcal{L}}_{\boldsymbol{\theta}_t}(\boldsymbol{\theta}) = \hat{\mathcal{L}}_{\boldsymbol{\theta}_t}(\hat{\boldsymbol{\theta}}_{t+1}) = \mathcal{L}(\boldsymbol{\theta}_t) - \frac{1}{2\alpha_t}\|\nabla_{\boldsymbol{\theta}}\mathcal{L}(\boldsymbol{\theta}_t)\|_2^2.$$

Then, we have

$$\inf_{\boldsymbol{\theta} \in B_t} \mathcal{L}(\boldsymbol{\theta}) \overset{(a)}{\geq} \inf_{\boldsymbol{\theta} \in B_t} \hat{\mathcal{L}}_{\boldsymbol{\theta}_t}(\boldsymbol{\theta}) \geq \inf_{\boldsymbol{\theta}} \hat{\mathcal{L}}_{\boldsymbol{\theta}_t}(\boldsymbol{\theta}) = \mathcal{L}(\boldsymbol{\theta}_t) - \frac{1}{2\alpha_t}\|\nabla_{\boldsymbol{\theta}}\mathcal{L}(\boldsymbol{\theta}_t)\|_2^2,$$

where (a) follows from (21). Rearranging terms completes the proof. $\square$

Next, we show that the restricted PL condition on $B_t$ in Lemma C along with smoothness (Condition 2) can be used to show geometric loss reduction on $B_t$.

**Lemma C.2 (Local Loss Reduction).** *Assume the loss $\mathcal{L}$ satisfies Conditions 1 and 2 with $\alpha_t \leq \beta$ at step $t$ of the GD update as in* (8) *with step-size $\eta_t = \frac{\omega_t}{\beta}$ for some $\omega_t \in (0, 2)$. Then, for any $\bar{\boldsymbol{\theta}}_{t+1} \in \text{arginf}_{\boldsymbol{\theta} \in Q_\kappa^t \cap \mathcal{N}(\boldsymbol{\theta}_0)} \mathcal{L}(\boldsymbol{\theta})$, we have*

$$\mathcal{L}(\boldsymbol{\theta}_{t+1}) - \mathcal{L}(\bar{\boldsymbol{\theta}}) \leq \left(1 - \frac{\alpha_t \omega_t}{\beta}(2 - \omega_t)\right)(\mathcal{L}(\boldsymbol{\theta}_t) - \mathcal{L}(\bar{\boldsymbol{\theta}})) . \tag{22}$$

*Proof.* Since $\mathcal{L}$ is $\beta$-smooth by Theorem D.4, we have

$$\begin{aligned}
\mathcal{L}(\boldsymbol{\theta}_{t+1}) &\leq \mathcal{L}(\boldsymbol{\theta}_t) + \langle \boldsymbol{\theta}_{t+1} - \boldsymbol{\theta}_t, \nabla_{\boldsymbol{\theta}}\mathcal{L}(\boldsymbol{\theta}_t)\rangle + \frac{\beta}{2}\|\boldsymbol{\theta}_{t+1} - \boldsymbol{\theta}_t\|_2^2 \\
&= \mathcal{L}(\boldsymbol{\theta}_t) - \eta_t\|\nabla_{\boldsymbol{\theta}}\mathcal{L}(\boldsymbol{\theta}_t)\|_2^2 + \frac{\beta\eta_t^2}{2}\|\nabla_{\boldsymbol{\theta}}\mathcal{L}(\boldsymbol{\theta}_t)\|_2^2 \\
&= \mathcal{L}(\boldsymbol{\theta}_t) - \eta_t\left(1 - \frac{\beta\eta_t}{2}\right)\|\nabla_{\boldsymbol{\theta}}\mathcal{L}(\boldsymbol{\theta}_t)\|_2^2
\end{aligned} \tag{23}$$

Since $\bar{\boldsymbol{\theta}}_{t+1} \in \text{arginf}_{\boldsymbol{\theta} \in B_t} \mathcal{L}(\boldsymbol{\theta})$ and $\alpha_t > 0$ by assumption, from Lemma C we obtain

$$-\|\nabla_{\boldsymbol{\theta}}\mathcal{L}(\boldsymbol{\theta}_t)\|_2^2 \leq -2\alpha_t(\mathcal{L}(\boldsymbol{\theta}_t) - \mathcal{L}(\bar{\boldsymbol{\theta}}_{t+1})) .$$

Hence

$$\begin{aligned}
\mathcal{L}(\boldsymbol{\theta}_{t+1}) - \mathcal{L}(\bar{\boldsymbol{\theta}}_{t+1}) &\leq \mathcal{L}(\boldsymbol{\theta}_t) - \mathcal{L}(\bar{\boldsymbol{\theta}}_{t+1}) - \eta_t\left(1 - \frac{\beta\eta_t}{2}\right)\|\nabla_{\boldsymbol{\theta}}\mathcal{L}(\boldsymbol{\theta}_t)\|_2^2 \\
&\overset{(a)}{\leq} \mathcal{L}(\boldsymbol{\theta}_t) - \mathcal{L}(\bar{\boldsymbol{\theta}}_{t+1}) - \eta_t\left(1 - \frac{\beta\eta_t}{2}\right)2\alpha_t(\mathcal{L}(\boldsymbol{\theta}_t) - \mathcal{L}(\bar{\boldsymbol{\theta}}_{t+1})) \\
&= \left(1 - 2\alpha_t\eta_t\left(1 - \frac{\beta\eta_t}{2}\right)\right)(\mathcal{L}(\boldsymbol{\theta}_t) - \mathcal{L}(\bar{\boldsymbol{\theta}}_{t+1}))
\end{aligned}$$

where (a) follows for any $\eta_t \leq \frac{2}{\beta}$ because this implies $1 - \frac{\beta\eta_t}{2} \geq 0$. Choosing $\eta_t = \frac{\omega_T}{\beta}, \omega_t \in (0, 2)$,

$$\mathcal{L}(\boldsymbol{\theta}_{t+1}) - \mathcal{L}(\bar{\boldsymbol{\theta}}_{t+1}) \leq \left(1 - \frac{\alpha_t \omega_t}{\beta}(2 - \omega_t)\right)(\mathcal{L}(\boldsymbol{\theta}_t) - \mathcal{L}(\bar{\boldsymbol{\theta}}_{t+1})) .$$

This completes the proof. $\qquad\square$

Finally, we show that the local geometric loss reduction result in $B_t$ (Lemma C.2) can be extended to show geometric loss reduction, which is the main optimization result.

**Theorem 1 (Global Loss Reduction).** *Assume the loss $\mathcal{L}$ satisfies Conditions 1 and 2 with $\alpha_t \leq \beta$ at step $t$ of the GD update as in* (8) *with step-size $\eta_t = \frac{\omega_t}{\beta}$ for some $\omega_t \in (0, 2)$. Then, $\forall \bar{\bar{\boldsymbol{\theta}}} \in$*

$$\underset{\boldsymbol{\theta} \in B_\rho^{\text{Euc}}(\boldsymbol{\theta}_0)}{\text{arginf}} \mathcal{L}(\boldsymbol{\theta}) \text{ and } \bar{\boldsymbol{\theta}}_{t+1} \in \underset{\boldsymbol{\theta} \in Q_\kappa^t \cap B_\rho^{\text{Euc}}(\boldsymbol{\theta}_0)}{\text{arginf}} \mathcal{L}(\boldsymbol{\theta}) \text{ with } 0 \leq \gamma_t := \frac{\mathcal{L}(\bar{\boldsymbol{\theta}}_{t+1}) - \mathcal{L}(\bar{\boldsymbol{\theta}})}{\mathcal{L}(\boldsymbol{\theta}_t) - \mathcal{L}(\bar{\boldsymbol{\theta}})} < 1, \text{ we have}$$

$$\mathcal{L}(\boldsymbol{\theta}_{t+1}) - \mathcal{L}(\bar{\boldsymbol{\theta}}) \leq \left(1 - \frac{\alpha_t \omega_t(1 - \gamma_t)}{\beta}(2 - \omega_t)\right)(\mathcal{L}(\boldsymbol{\theta}_t) - \mathcal{L}(\bar{\boldsymbol{\theta}})) . \tag{9}$$

*Proof.* We start by showing $\gamma_t = \frac{\mathcal{L}(\bar{\boldsymbol{\theta}}_{t+1}) - \mathcal{L}(\boldsymbol{\theta}^*)}{\mathcal{L}(\boldsymbol{\theta}_t) - \mathcal{L}(\boldsymbol{\theta}^*)}$ satisfies $0 \leq \gamma_t < 1$. Since $\boldsymbol{\theta}^* \in \underset{\boldsymbol{\theta} \in \mathcal{N}(\boldsymbol{\theta}_0)}{\text{arginf}} \mathcal{L}(\boldsymbol{\theta})$,

$\bar{\boldsymbol{\theta}}_{t+1} \in \underset{\boldsymbol{\theta} \in B_t}{\text{arginf}} \mathcal{L}(\boldsymbol{\theta})$, and $\boldsymbol{\theta}_{t+1} \in Q_\kappa^t \cap \mathcal{N}(\boldsymbol{\theta}_0)$ by the definition of gradient descent, we have

$$\mathcal{L}(\boldsymbol{\theta}^*) \leq \mathcal{L}(\bar{\boldsymbol{\theta}}_{t+1}) \leq \mathcal{L}(\boldsymbol{\theta}_{t+1}) \overset{(a)}{\leq} \mathcal{L}(\boldsymbol{\theta}_t) - \frac{1}{2\beta}\|\nabla_{\boldsymbol{\theta}}\mathcal{L}(\boldsymbol{\theta}_t)\|_2^2 < \mathcal{L}(\boldsymbol{\theta}_t) ,$$

where (a) follows from (23). Since $\mathcal{L}(\bar{\boldsymbol{\theta}}_{t+1}) \geq \mathcal{L}(\boldsymbol{\theta}^*)$ and $\mathcal{L}(\boldsymbol{\theta}_t) > \mathcal{L}(\boldsymbol{\theta}^*)$, we have $\gamma_t \geq 0$. Further, since $\mathcal{L}(\bar{\boldsymbol{\theta}}_{t+1}) < \mathcal{L}(\boldsymbol{\theta}_t)$, we have $\gamma_t < 1$.

Now, with $\omega_t \in (0, 2)$, we have

$$
\begin{aligned}
\mathcal{L}(\boldsymbol{\theta}_{t+1}) - \mathcal{L}(\boldsymbol{\theta}^*) &= \mathcal{L}(\boldsymbol{\theta}_{t+1}) - \mathcal{L}(\bar{\boldsymbol{\theta}}_{t+1}) + \mathcal{L}(\bar{\boldsymbol{\theta}}_{t+1}) - \mathcal{L}(\boldsymbol{\theta}^*) \\
&\leq \left(1 - \frac{\alpha_t \omega_t}{\beta}(2 - \omega_t)\right)(\mathcal{L}(\boldsymbol{\theta}_t) - \mathcal{L}(\bar{\boldsymbol{\theta}}_{t+1})) + \left(1 - \frac{\alpha_t \omega_t}{\beta}(2 - \omega_t)\right)(\mathcal{L}(\bar{\boldsymbol{\theta}}_{t+1}) - \mathcal{L}(\boldsymbol{\theta}^*)) \\
&\quad + \left(\mathcal{L}(\bar{\boldsymbol{\theta}}_{t+1}) - \left(1 - \frac{\alpha_t \omega_t}{\beta}(2 - \omega_t)\right)\mathcal{L}(\bar{\boldsymbol{\theta}}_{t+1})\right) - \left(\mathcal{L}(\boldsymbol{\theta}^*) - \left(1 - \frac{\alpha_t \omega_t}{\beta}(2 - \omega_t)\right)\mathcal{L}(\boldsymbol{\theta}^*)\right) \\
&= \left(1 - \frac{\alpha_t \omega_t}{\beta}(2 - \omega_t)\right)(\mathcal{L}(\boldsymbol{\theta}_t) - \mathcal{L}(\boldsymbol{\theta}^*)) + \frac{\alpha_t \omega_t}{\beta}(2 - \omega_t)(\mathcal{L}(\bar{\boldsymbol{\theta}}_{t+1}) - \mathcal{L}(\boldsymbol{\theta}^*)) \\
&= \left(1 - \frac{\alpha_t \omega_t}{\beta}(2 - \omega_t)\right)(\mathcal{L}(\boldsymbol{\theta}_t) - \mathcal{L}(\boldsymbol{\theta}^*)) + \frac{\alpha_t \omega_t}{\beta}(2 - \omega_t)\gamma_t(\mathcal{L}(\boldsymbol{\theta}_t) - \mathcal{L}(\boldsymbol{\theta}^*)) \\
&= \left(1 - \frac{\alpha_t \omega_t}{\beta}(1 - \gamma_t)(2 - \omega_t)\right)(\mathcal{L}(\boldsymbol{\theta}_t) - \mathcal{L}(\boldsymbol{\theta}^*)) .
\end{aligned}
$$

That completes the proof. $\qquad \square$

## D   ANALYSIS FOR DEEPONETS, FOR SECTION 5

### D.1   NON-EMPTY RESTRICTED SET $Q_\kappa^t$ FOR DEEPONETS, AS IN DEFINITION 2

First, recall our definition of the $Q_\kappa^t$ set for DeepONets:

**Definition 2** ($Q_\kappa^t$ **sets for DONs**). *For an iterate $\boldsymbol{\theta}_t = [\boldsymbol{\theta}_{f,t}^\top \ \boldsymbol{\theta}_{g,t}^\top]^\top$, consider the singular value decomposition $\frac{1}{n}\sum_{i=1}^n \frac{1}{q_i}\sum_{j=1}^{q_i} \ell'_{i,j} \sum_{k=1}^K \nabla_{\boldsymbol{\theta}_f} f_k^{(i)} \nabla_{\boldsymbol{\theta}_g} g_{k,j}^{(i)}{}^\top = \sum_{h=1}^{\tilde{q}} \sigma_h \boldsymbol{a}_h \boldsymbol{b}_h^\top$, where $\tilde{q} \leq qk$ with $q = \sum_{i=1}^n q_i$, and $\sigma_h > 0, \boldsymbol{a}_h \in \mathbb{R}^{p_f}, \boldsymbol{b}_h \in \mathbb{R}^{p_g}$ respectively denote the singular values, left singular vectors, and right singular vectors. Further, let $\bar{G}_{\boldsymbol{\theta}} = \frac{1}{n}\sum_{i=1}^n \frac{1}{q_i}\sum_{j=1}^{q_i} G_{\boldsymbol{\theta}}(u^{(i)})(y_j^{(i)})$. Then, for a suitable $\kappa \in (0, \frac{1}{\sqrt{2}}]$, we define the set:*

$$
Q_\kappa^t := \left\{ \boldsymbol{\theta}' = [\boldsymbol{\theta}_f'^\top \ \boldsymbol{\theta}_g'^\top]^\top \ : \ |\cos(\boldsymbol{\theta}' - \boldsymbol{\theta}_t, \nabla_{\boldsymbol{\theta}}\bar{G}_{\boldsymbol{\theta}_t})| \geq \kappa, \sum_{h=1}^{\tilde{q}} \sigma_h \langle \boldsymbol{\theta}_f' - \boldsymbol{\theta}_{f,t}, \boldsymbol{a}_h \rangle \langle \boldsymbol{\theta}_g' - \boldsymbol{\theta}_{g,t}, \boldsymbol{b}_h \rangle \geq 0 \right\} . \quad (12)
$$

We now show that these restricted sets $Q_\kappa^t$ are non-empty.

**Proposition 1** ($Q_\kappa^t$ **is non-empty**). *For over-parameterized branch and trunk nets with $p_f, p_g > qk$ where $q = \sum_{i=1}^n q_i$, the restricted set $Q_\kappa^t$ as defined in 2 is non-empty.*

*Proof.* From the definition of $Q_\kappa^t$, $\boldsymbol{\theta}'$ needs to satisfy two conditions, which we refer to respectively as the cosine similarity condition and svd condition for convenience:

$$
|\cos(\boldsymbol{\theta}' - \boldsymbol{\theta}_t, \nabla_{\boldsymbol{\theta}}\bar{G}_{\boldsymbol{\theta}_t})| \geq \kappa \qquad \text{(cosine similarity condition)} ,
$$

$$
\sum_{h=1}^{\tilde{q}} \sigma_h \langle \boldsymbol{\theta}_f' - \boldsymbol{\theta}_{f,t}, \boldsymbol{a}_h \rangle \langle \boldsymbol{\theta}_g' - \boldsymbol{\theta}_{g,t}, \boldsymbol{b}_h \rangle \geq 0 \qquad \text{(svd condition)} .
$$

We simply construct a $\boldsymbol{\theta}' = [\boldsymbol{\theta}_f'^\top \ \boldsymbol{\theta}_g'^\top]^\top \in Q_\kappa^t$ along with the value of $\kappa$. Without loss of generality, we make $\boldsymbol{\theta}_t$ the origin of the coordinate system and work with the unit vector $\bar{\mathbf{g}} = [\bar{\mathbf{g}}_f^\top \ \bar{\mathbf{g}}_g^\top]^\top = \frac{\nabla_{\boldsymbol{\theta}}\bar{G}_{\boldsymbol{\theta}_t}}{\|\nabla_{\boldsymbol{\theta}}\bar{G}_{\boldsymbol{\theta}_t}\|_2}$, since the cosine similarity condition does not care for magnitudes Further, we assume $\boldsymbol{\theta}'$ also to be a unit vector. Then, our problem reduces to feasibility of the following system of two quadratic equations over $\boldsymbol{\theta}_f' \in \mathbb{R}^{p_f}, \boldsymbol{\theta}_g' \in \mathbb{R}^{p_g}$:

$$
\left(\langle \boldsymbol{\theta}_f', \bar{\mathbf{g}}_f \rangle + \langle \boldsymbol{\theta}_g', \bar{\mathbf{g}}_g \rangle\right)^2 \geq \kappa^2 \qquad \text{(cosine similarity condition)} ,
$$

$$
\boldsymbol{\theta}_f'^\top \left(\sum_{h=1}^{\tilde{q}} \sigma_h \boldsymbol{a}_h \boldsymbol{b}_h^\top\right) \boldsymbol{\theta}_g' \geq 0 \qquad \text{(svd condition)} ,
$$

where singular values $\sigma_h > 0$, $\boldsymbol{a}_h \in \mathbb{R}^{p_f}$ are orthogonal unit vectors, $\boldsymbol{b}_h \in \mathbb{R}^{p_g}$ are orthogonal unit vectors, $\bar{\mathbf{g}} = [\bar{\mathbf{g}}_f^\top \ \bar{\mathbf{g}}_g^\top]^\top \in \mathbb{R}^{p_f + p_g}$ and $\boldsymbol{\theta}' = [\boldsymbol{\theta}_f'; \boldsymbol{\theta}_g'] \in \mathbb{R}^{p_f + p_g}$ are unit vectors, and we can choose a suitable $\kappa \in (0, \frac{1}{2})$. Without loss of generality, assume $\|\bar{\mathbf{g}}_f\|_2 \geq \|\bar{\mathbf{g}}_g\|_2$ so that $\|\bar{\mathbf{g}}_f\|_2 \geq \frac{1}{\sqrt{2}}$. Then, set $\boldsymbol{\theta}_g' = \mathbf{0}$ so that our feasibility condition reduces to $\langle \boldsymbol{\theta}_f', \bar{\mathbf{g}}_f \rangle^2 \geq \kappa^2$ for some suitably chosen $\kappa \in (0, 1]$. Finally, set $\boldsymbol{\theta}_f' = \frac{\bar{\mathbf{g}}_f}{\|\bar{\mathbf{g}}_f\|_2}$ so that

$$\langle \boldsymbol{\theta}_f', \bar{\mathbf{g}}_f \rangle^2 = \left( \frac{\bar{\mathbf{g}}_f}{\|\bar{\mathbf{g}}_f\|_2} \bar{\mathbf{g}}_f \right)^2 = \|\bar{\mathbf{g}}_f\|_2^2 \geq \frac{1}{2} \,,$$

so that the feasibility condition is satisfied for $\kappa \in (0, \frac{1}{\sqrt{2}}]$ That completes the proof. $\qquad\square$

### D.2 Spectral Norm of the Hessian of branch and trunk Nets

The convergence analysis makes use of the gradients and Hessians of the total loss and the predictor with respect to the parameters $\boldsymbol{\theta}$, namely,

$$\nabla_{\boldsymbol{\theta}} \mathcal{L}(\boldsymbol{\theta}) = \left[ \nabla_{\boldsymbol{\theta}_f} \mathcal{L}; \nabla_{\boldsymbol{\theta}_g} \mathcal{L} \right], \quad \text{and} \quad \nabla_{\boldsymbol{\theta}}^2 \mathcal{L} = \boldsymbol{H}(\boldsymbol{\theta}) = \begin{bmatrix} H_{ff} & H_{fg} \\ H_{gf} & H_{gg} \end{bmatrix}, \qquad (24)$$

where $\nabla_{\boldsymbol{\theta}_f} \mathcal{L}(\boldsymbol{\theta}) \in \mathbb{R}^{p_f} = \partial \mathcal{L}(\boldsymbol{\theta})/\partial \boldsymbol{\theta}_f$ and $\nabla_{\boldsymbol{\theta}_g} \mathcal{L}(\boldsymbol{\theta}) = \partial \mathcal{L}(\boldsymbol{\theta})/\partial \boldsymbol{\theta}_g \in \mathbb{R}^{p_g}$. Note that we make use of the notation $\nabla_{\boldsymbol{\theta}_f}(\cdot)$ to denote the derivative wrt the parameters $\boldsymbol{\theta}_f$ and this *is not* a functional gradient. Similarly, the individual blocks in the $2 \times 2$ block Hessian $\boldsymbol{H}(\boldsymbol{\theta})$ are given by

$$H_{ff} = \nabla_{\boldsymbol{\theta}_f}^2 \mathcal{L} = \frac{\partial^2 \mathcal{L}}{\partial \boldsymbol{\theta}_f{}^2}, \quad H_{fg} = \frac{\partial^2 \mathcal{L}}{\partial \boldsymbol{\theta}_f \partial \boldsymbol{\theta}_g}, \quad H_{gf} = H_{fg}^\top = \frac{\partial^2 \mathcal{L}}{\partial \boldsymbol{\theta}_g \partial \boldsymbol{\theta}_f}, \quad H_{gg} = \nabla_{\boldsymbol{\theta}_g}^2 \mathcal{L} = \frac{\partial^2 \mathcal{L}}{\partial \boldsymbol{\theta}_g^2}, \tag{25}$$

where $H_{ff} \in \mathbb{R}^{p_f \times p_f}$, $H_{gg} \in \mathbb{R}^{p_g \times p_g}$, $H_{fg} \in \mathbb{R}^{p_f \times p_g}$, $H_{gf} \in \mathbb{R}^{p_g \times p_f}$ and the argument $\boldsymbol{\theta}$ is ignored for clarity of exposition. Using (2) and rewriting the derivatives in (24) and (25), we get

$$\frac{\partial \mathcal{L}}{\partial \boldsymbol{\theta}_f} = \frac{1}{n} \sum_{i=1}^{n} \frac{1}{q_i} \sum_{j=1}^{q_i} \ell'_{i,j} \sum_{k=1}^{K} g_{k,j}^{(i)} \nabla_{\boldsymbol{\theta}_f} f_k^{(i)} \quad \text{and} \quad \frac{\partial \mathcal{L}}{\partial \boldsymbol{\theta}_g} = \frac{1}{n} \sum_{i=1}^{n} \frac{1}{q_i} \sum_{j=1}^{q_i} \ell'_{ij} \sum_{k=1}^{K} f_k^{(i)} \nabla_{\boldsymbol{\theta}_g} g_{k,j}^{(i)}, \quad (26)$$

for the gradients, and

$$\frac{\partial^2 \mathcal{L}}{\partial \boldsymbol{\theta}_f{}^2} = \frac{1}{n} \sum_{i=1}^{n} \frac{1}{q_i} \sum_{j=1}^{q_i} \ell'_{i,j} \sum_{k=1}^{K} g_{k,j}^{(i)} \nabla_{\boldsymbol{\theta}_f}^2 f_k^{(i)} + \frac{1}{n} \sum_{i=1}^{n} \frac{1}{q_i} \sum_{j=1}^{q_i} \ell''_{i,j} \left( \sum_{k,\hat{k}=1}^{K} g_{k,j}^{(i)} g_{\hat{k},j}^{(i)} \nabla_{\boldsymbol{\theta}_f} f_k^{(i)} \nabla_{\boldsymbol{\theta}_f} f_{\hat{k}}^{(i)\top} \right),$$

$$\frac{\partial^2 \mathcal{L}}{\partial \boldsymbol{\theta}_g{}^2} = \frac{1}{n} \sum_{i=1}^{n} \frac{1}{q_i} \sum_{j=1}^{q_i} \ell'_{i,j} \sum_{k=1}^{k} f_k^{(i)} \nabla_{\boldsymbol{\theta}_g}^2 g_{k,j}^{(i)} + \frac{1}{n} \sum_{i=1}^{n} \frac{1}{q_i} \sum_{j=1}^{q_i} \ell''_{i,j} \left( \sum_{k,\hat{k}=1}^{K} f_k^{(i)} f_{\hat{k}}^{(i)} \nabla_{\boldsymbol{\theta}_g} g_{k,j}^{(i)} \nabla_{\boldsymbol{\theta}_g} g_{\hat{k},j}^{(i)\top} \right),$$

$$\frac{\partial^2 \mathcal{L}}{\partial \boldsymbol{\theta}_f \partial \boldsymbol{\theta}_g} = \underbrace{\frac{1}{n} \sum_{i=1}^{n} \frac{1}{q_i} \sum_{j=1}^{q_i} \ell'_{i,j} \sum_{k=1}^{K} \nabla_{\boldsymbol{\theta}_f} f_k^{(i)} \nabla_{\boldsymbol{\theta}_g} g_{k,j}^{(i)\top}}_{=H_{fg}^{(1)}} + \underbrace{\frac{1}{n} \sum_{i=1}^{n} \frac{1}{q_i} \sum_{j=1}^{q_i} \ell''_{i,j} \left( \sum_{k,\hat{k}=1}^{K} g_{k,j}^{(i)} f_{\hat{k}}^{(i)} \nabla_{\boldsymbol{\theta}_f} f_k^{(i)} \nabla_{\boldsymbol{\theta}_g} g_{\hat{k},j}^{(i)\top} \right)}_{=H_{fg}^{(2)}},$$

$$(27)$$

for the individual blocks of the hessian (24) where, we make use of the notation $g_{k,j}^{(i)} = g_k(\boldsymbol{\theta}_g; y_j^{(i)})$ and $f_k^{(i)} = f_k(\boldsymbol{\theta}_f; u^{(i)})$.

In order to prove the RSC and smoothness properties of the empirical loss $\mathcal{L}$ in the next section, we will need to upper bound the spectral norm of its Hessian. As can be seen above, the gradient and Hessians of the predictors (i.e., the branch and trunk networks) appear in the Hessian of $\mathcal{L}$, and thus, we will eventually need the upper bound of their norms. For this, we will make use of the next lemma.

**Lemma D.3** (Bounds on the Predictor). *Under Assumptions 1 and 2, and for $\boldsymbol{\theta} \in B_\rho^{\mathrm{Euc}}(\boldsymbol{\theta}_0)$, with probability at least $1 - 2L\left(\frac{1}{m_f} + \frac{1}{m_g}\right)$, we have for every $k \in [K]$, $i \in [n]$, $j \in [q_i]$,*

$$\left\|\nabla_{\boldsymbol{\theta}_f}^2 f_k^{(i)}\right\| \leq \frac{c^{(f)}}{\sqrt{m_f}}, \quad \text{and} \quad \left\|\nabla_{\boldsymbol{\theta}_g}^2 g_{k,j}^{(i)}\right\| \leq \frac{c^{(g)}}{\sqrt{m_g}} \tag{28}$$

$$\left\|\nabla_{\boldsymbol{\theta}_f} f_k^{(i)}\right\|_2 \leq \varrho^{(f)}, \quad \text{and} \quad \left\|\nabla_{\boldsymbol{\theta}_g} g_{k,j}^{(i)}\right\|_2 \leq \varrho^{(g)}, \tag{29}$$

$$|f_k^{(i)}| \leq \lambda_1, \quad \text{and} \quad |g_{k,j}^{(i)}| \leq \lambda_2, \tag{30}$$

*where $c^{(f)}, c^{(g)}, \varrho^{(f)}, \varrho^{(g)}, \lambda_1 \lambda_2$ are suitable constants that depend on the depth $L$ and the radius $\rho$, $\rho_1$.*

*Proof.* The proof follows from a direct adaptation of Theorem 4.1 and Lemma 4.1 in (Banerjee et al., 2023). $\qquad \square$

### D.4 Optimization Guarantees for DeepONets

**Theorem 2** (RSC). *Under Assumptions 1 and 2 and $Q_\kappa^t$ as in Definition 2, (a) $B_\kappa^t := Q_\kappa^t \cap B_{\rho,\rho_1}^{\mathrm{Euc}}(\boldsymbol{\theta}_0) \cap B_{\rho_2}^{\mathrm{Euc}}(\boldsymbol{\theta}_t)$ is non-empty for suitable $\rho, \rho_2 = O(1)$, and (b) with probability at least $1 - \frac{4L}{m}$, at step $t$ of GD, $\forall \boldsymbol{\theta}' \in B_\kappa^t$, the DON loss $\mathcal{L}$ satisfies*

$$\alpha_t = c_1 \|\nabla_{\boldsymbol{\theta}} \bar{G}_t\|_2^2 - \frac{c_2}{\sqrt{m}}, \quad \text{where} \quad \bar{G}_t = \frac{1}{n} \sum_{i=1}^n \frac{1}{q_i} \sum_{j=1}^{q_i} G_{\boldsymbol{\theta}_t}(u^{(i)})(y_j^{(i)}). \tag{13}$$

*for some constants $c_1, c_2 > 0$, where $c_2$ depends on the depth $L$ and the radii $\rho, \rho_1, \rho_2$. Thus, the loss $\mathcal{L}$ satisfies RSC w.r.t $(B_\kappa^t, \boldsymbol{\theta}_t)$ whenever $\|\nabla_{\boldsymbol{\theta}} \bar{G}_t\|_2^2 = \Omega(\frac{1}{\sqrt{m}})$.*

For (b), for any $\boldsymbol{\theta}' \in B_\kappa^t$, by the second order Taylor expansion of the DeepONet loss w.r.t. iterate $\boldsymbol{\theta}_t \in B_\kappa^t$, we have

$$\mathcal{L}(\boldsymbol{\theta}') = \mathcal{L}(\boldsymbol{\theta}_t) + \langle \boldsymbol{\theta}' - \boldsymbol{\theta}_t, \nabla_{\boldsymbol{\theta}} \mathcal{L}(\boldsymbol{\theta}_t) \rangle + \frac{1}{2}(\boldsymbol{\theta}' - \boldsymbol{\theta}_t)^\top \frac{\partial^2 \mathcal{L}(\tilde{\boldsymbol{\theta}})}{\partial \boldsymbol{\theta}^2} (\boldsymbol{\theta}' - \boldsymbol{\theta}_t),$$

where $\tilde{\boldsymbol{\theta}} = \xi \boldsymbol{\theta}' + (1 - \xi)\boldsymbol{\theta}_t$ for some $\xi \in [0, 1]$. To establish $\alpha_t$-RSC of the loss with $\alpha_t$ as in (13), it suffices to focus on the quadratic form of the Hessian for $\boldsymbol{\theta}' \in B_\kappa^t$ and show

$$(\boldsymbol{\theta}' - \boldsymbol{\theta}_t)^\top \boldsymbol{H}(\tilde{\boldsymbol{\theta}})(\boldsymbol{\theta} - \boldsymbol{\theta}_t) \geq \alpha_t \|\boldsymbol{\theta}' - \boldsymbol{\theta}_t\|_2^2. \tag{31}$$

Note that the Hessian, by chain rule, is given by

$$\boldsymbol{H}(\tilde{\boldsymbol{\theta}}) = \frac{\partial^2 \mathcal{L}(\tilde{\boldsymbol{\theta}})}{\partial \boldsymbol{\theta}^2} = \frac{1}{n} \sum_{i=1}^n \frac{1}{q_i} \sum_{j=1}^{q_i} \left( \ell_{i,j}'' \nabla G_{\tilde{\boldsymbol{\theta}}}(u^{(i)})(y_j^{(i)}) \nabla G_{\tilde{\boldsymbol{\theta}}}(u^{(i)})(y_j^{(i)})^\top + \ell_{i,j}' \nabla^2 G_{\tilde{\boldsymbol{\theta}}}(u^{(i)})(y_j^{(i)}) \right).$$

Given the $2 \times 2$ block structure of the Hessian as in (24), denoting $\delta\boldsymbol{\theta} := \boldsymbol{\theta}' - \boldsymbol{\theta}_t$ for compactness, the quadratic form on the Hessian is given by

$$\delta\boldsymbol{\theta}^\top \boldsymbol{H}(\tilde{\boldsymbol{\theta}})\delta\boldsymbol{\theta} = \underbrace{\delta\boldsymbol{\theta}_f^\top H_{ff}(\tilde{\boldsymbol{\theta}})\delta\boldsymbol{\theta}_f}_{T_1} + \underbrace{2\delta\boldsymbol{\theta}_f^\top H_{fg}(\tilde{\boldsymbol{\theta}})\delta\boldsymbol{\theta}_g}_{T_2} + \underbrace{\delta\boldsymbol{\theta}_g^\top H_{gg}(\tilde{\boldsymbol{\theta}})\delta\boldsymbol{\theta}_g}_{T_3}. \tag{32}$$

One of the aspects of the analysis for each of the terms $T_1, T_2, T_3$ is that we have to change the dependencies of the gradient terms from $\tilde{\boldsymbol{\theta}}$ to $\boldsymbol{\theta}_t$ and then suitably use properties of $Q_\kappa^t$. Focusing

on $T_1$ and using the exact form of $H_{ff}(\tilde{\boldsymbol{\theta}})$ as in (27), we have

$$
\begin{aligned}
T_1 &= \frac{1}{n}\sum_{i=1}^{n}\frac{1}{q_i}\sum_{j=1}^{q_i}\ell_{i,j}''\left\langle \delta\boldsymbol{\theta}_f, \sum_{k=1}^{K} g_{k,j}^{(i)}\nabla_{\boldsymbol{\theta}_f}f_k^{(i)}(\tilde{\boldsymbol{\theta}}_f)\right\rangle^2 + \frac{1}{n}\sum_{i=1}^{n}\frac{1}{q_i}\sum_{j=1}^{q_i}\ell_{ij}'\sum_{k=1}^{K}g_{k,j}^{(i)}\delta\boldsymbol{\theta}_f^\top\nabla_{\boldsymbol{\theta}_f}^2 f_k^{(i)}(\tilde{\boldsymbol{\theta}}_f)\delta\boldsymbol{\theta}_f \\
&\overset{(a)}{\geq} \frac{2}{n}\sum_{i=1}^{n}\frac{1}{q_i}\sum_{j=1}^{q_i}\left\langle \delta\boldsymbol{\theta}_f, \nabla_{\boldsymbol{\theta}_f}G_{\tilde{\boldsymbol{\theta}}}(u^{(i)})(y_j^{(i)})\right\rangle^2 - \frac{(2K\lambda_1\lambda_2+\tilde{c})\lambda_2 c^{(f)}}{\sqrt{m_f}}\|\delta\boldsymbol{\theta}_f\|_2^2 \\
&= \underbrace{\frac{2}{n}\sum_{i=1}^{n}\frac{1}{q_i}\sum_{j=1}^{q_i}\left\langle \delta\boldsymbol{\theta}_f, \nabla_{\boldsymbol{\theta}_f}G_{\boldsymbol{\theta}_t}(u^{(i)})(y_j^{(i)}) + \left(\nabla_{\boldsymbol{\theta}_f}G_{\tilde{\boldsymbol{\theta}}}(u^{(i)})(y_j^{(i)}) - \nabla_{\boldsymbol{\theta}_f}G_{\boldsymbol{\theta}_t}(u^{(i)})(y_j^{(i)})\right)\right\rangle^2}_{I_1} \\
&\quad - \frac{(2K\lambda_1\lambda_2+\tilde{c})\lambda_2 c^{(f)}}{\sqrt{m_f}}\|\delta\boldsymbol{\theta}_f\|_2^2 ,
\end{aligned}
$$

where (a) follows from having a square loss and the different bounds in Lemma D.3, so that $\ell_{ij}'' = 2$ and $|\ell_{ij}'| \leq 2K\lambda_1\lambda_2 + \tilde{c}$ with $\tilde{c} = 2\max_{ij}G^\dagger(u^{(i)})(y_j^{(i)})$. Now, note that

$$
\begin{aligned}
I_1 &= \frac{1}{n}\sum_{i=1}^{n}\frac{1}{q_i}\sum_{j=1}^{q_i}\left\langle \delta\boldsymbol{\theta}_f, \nabla_{\boldsymbol{\theta}_f}G_{\boldsymbol{\theta}_t}(u^{(i)})(y_j^{(i)})\right\rangle^2 + \sum_{i=1}^{n}\frac{1}{q_i}\sum_{j=1}^{q_i}\left\langle \delta\boldsymbol{\theta}_f, \nabla_{\boldsymbol{\theta}_f}G_{\tilde{\boldsymbol{\theta}}}(u^{(i)})(y_j^{(i)}) - \nabla_{\boldsymbol{\theta}_f}G_{\boldsymbol{\theta}_t}(u^{(i)})(y_j^{(i)})\right\rangle^2 \\
&\quad + \frac{2}{n}\sum_{i=1}^{n}\frac{1}{q_i}\sum_{j=1}^{q_i}\left\langle \delta\boldsymbol{\theta}_f, \nabla_{\boldsymbol{\theta}_f}G_{\boldsymbol{\theta}_t}(u^{(i)})(y_j^{(i)})\right\rangle\left\langle \delta\boldsymbol{\theta}_f, \nabla_{\boldsymbol{\theta}_f}G_{\tilde{\boldsymbol{\theta}}}(u^{(i)})(y_j^{(i)}) - \nabla_{\boldsymbol{\theta}_f}G_{\boldsymbol{\theta}_t}(u^{(i)})(y_j^{(i)})\right\rangle \\
&\geq \frac{1}{n}\sum_{i=1}^{n}\frac{1}{q_i}\sum_{j=1}^{q_i}\left\langle \delta\boldsymbol{\theta}_f, \nabla_{\boldsymbol{\theta}_f}G_{\boldsymbol{\theta}_t}(u^{(i)})(y_j^{(i)})\right\rangle^2 \\
&\quad - \frac{2}{n}\sum_{i=1}^{n}\frac{1}{q_i}\sum_{j=1}^{q_i}\left\|\nabla_{\boldsymbol{\theta}_f}G_{\boldsymbol{\theta}_t}(u^{(i)})(y_j^{(i)})\right\|_2 \left\|\nabla_{\boldsymbol{\theta}_f}G_{\tilde{\boldsymbol{\theta}}}(u^{(i)})(y_j^{(i)}) - \nabla_{\boldsymbol{\theta}_f}G_{\boldsymbol{\theta}_t}(u^{(i)})(y_j^{(i)})\right\|_2 \|\delta\boldsymbol{\theta}_f\|_2^2 \\
&\geq \frac{1}{n}\sum_{i=1}^{n}\frac{1}{q_i}\sum_{j=1}^{q_i}\left\langle \delta\boldsymbol{\theta}_f, \nabla_{\boldsymbol{\theta}_f}G_{\boldsymbol{\theta}_t}(u^{(i)})(y_j^{(i)})\right\rangle^2 - \varrho^{(f)}\frac{c^{(f)}}{\sqrt{m_f}}\|\delta\boldsymbol{\theta}_f\|_2^3 ,
\end{aligned}
$$

where we have used Lemma D.3 and the fact that $\|\tilde{\boldsymbol{\theta}}_f - \boldsymbol{\theta}_{t,f}\|_2 \leq \|\delta\boldsymbol{\theta}_f\|_2$ and $\|\tilde{\boldsymbol{\theta}}_g - \boldsymbol{\theta}_{t,g}\|_2 \leq \|\delta\boldsymbol{\theta}_g\|_2$. As a result

$$
T_1 \geq \frac{1}{n}\sum_{i=1}^{n}\frac{1}{q_i}\sum_{j=1}^{q_i}\left\langle \delta\boldsymbol{\theta}_f, \nabla_{\boldsymbol{\theta}_f}G_{\boldsymbol{\theta}_t}(u^{(i)})(y_j^{(i)})\right\rangle^2 - \frac{(\rho c^{(f)} + \lambda c_0)\rho^{(f)}}{\sqrt{m_f}}\|\delta\boldsymbol{\theta}_f\|_2^2 , \tag{33}
$$

where we have used $\|\delta\boldsymbol{\theta}_f\|_2 \leq \rho$. The analysis for $T_3$ is similar, and we get

$$
T_3 \geq \frac{1}{n}\sum_{i=1}^{n}\frac{1}{q_i}\sum_{j=1}^{q_i}\left\langle \delta\boldsymbol{\theta}_g, \nabla_{\boldsymbol{\theta}_g}G_{\boldsymbol{\theta}_t}(u^{(i)})(y_j^{(i)})\right\rangle^2 - \frac{(\rho c^{(g)} + \lambda c_0)\rho^{(g)}}{\sqrt{m_g}}\|\delta\boldsymbol{\theta}_g\|_2^2 . \tag{34}
$$

Focusing on $T_2$ and using the exact forms in terms of $H_{fg}^{(1)}(\tilde{\boldsymbol{\theta}})$ and $H_{fg}^{(2)}(\tilde{\boldsymbol{\theta}})$ as in (27), we have

$$
\begin{aligned}
\frac{1}{2}T_2 &= \underbrace{\delta\boldsymbol{\theta}_f^\top\left(\frac{1}{n}\sum_{i=1}^{n}\frac{1}{q_i}\sum_{j=1}^{q_i}\ell_{ij}'\sum_{k=1}^{K}\nabla_{\boldsymbol{\theta}_f}f_k^{(i)}(\tilde{\boldsymbol{\theta}}_f)\nabla_{\boldsymbol{\theta}_g}g_{k,j}^{(i)}(\tilde{\boldsymbol{\theta}}_g)^\top\right)\delta\boldsymbol{\theta}_g}_{I_2} \\
&\quad + \underbrace{\delta\boldsymbol{\theta}_f^\top\left(\frac{1}{n}\sum_{i=1}^{n}\frac{1}{q_i}\sum_{j=1}^{q_i}\ell_{i,j}''\left(\sum_{k=1}^{K}g_{k,j}^{(i)}\nabla_{\boldsymbol{\theta}_f}f_k^{(i)}(\tilde{\boldsymbol{\theta}}_f)\right)\left(\sum_{k'=1}^{K}f_{k'}^{(i)}\nabla_{\boldsymbol{\theta}_g}g_{k',j}^{(i)}(\tilde{\boldsymbol{\theta}}_g)^\top\right)\right)\delta\boldsymbol{\theta}_g}_{I_3} .
\end{aligned}
$$

For both $I_2$ and $I_3$, our goal is to first transfer the dependence of the gradient terms on $\tilde{\boldsymbol{\theta}}$ to $\boldsymbol{\theta}_t$, so that we can use properties of the restricted set $Q_\kappa^t$ which is based on $\boldsymbol{\theta}_t$ to simplify the analysis. Towards that end, note that

$$
\begin{aligned}
I_2 &= \delta\boldsymbol{\theta}_f^\top \left( \frac{1}{n}\sum_{i=1}^n \frac{1}{q_i}\sum_{j=1}^{q_i} \ell'_{ij} \sum_{k=1}^K \nabla_{\boldsymbol{\theta}_f} f_k^{(i)}(\boldsymbol{\theta}_{t,f}) \nabla_{\boldsymbol{\theta}_g} g_{k,j}^{(i)}(\boldsymbol{\theta}_{t,g})^\top \right) \delta\boldsymbol{\theta}_g \\
&+ \delta\boldsymbol{\theta}_f^\top \left( \frac{1}{n}\sum_{i=1}^n \frac{1}{q_i}\sum_{j=1}^{q_i} \ell'_{ij} \sum_{k=1}^K \left( \nabla_{\boldsymbol{\theta}_f} f_k^{(i)}(\tilde{\boldsymbol{\theta}}_f) - \nabla_{\boldsymbol{\theta}_f} f_k^{(i)}(\boldsymbol{\theta}_{t,f}) \right) \nabla_{\boldsymbol{\theta}_g} g_{k,j}^{(i)}(\tilde{\boldsymbol{\theta}}_g)^\top \right) \delta\boldsymbol{\theta}_g \\
&+ \delta\boldsymbol{\theta}_f^\top \left( \frac{1}{n}\sum_{i=1}^n \frac{1}{q_i}\sum_{j=1}^{q_i} \ell'_{ij} \sum_{k=1}^K \nabla_{\boldsymbol{\theta}_f} f_k^{(i)}(\boldsymbol{\theta}_{t,f}) \left( \nabla_{\boldsymbol{\theta}_g} g_{k,j}^{(i)}(\tilde{\boldsymbol{\theta}}_g) - \nabla_{\boldsymbol{\theta}_g} g_{k,j}^{(i)}(\boldsymbol{\theta}_{t,g}) \right)^\top \right) \delta\boldsymbol{\theta}_g \\
&\overset{(a)}{\geq} \delta\boldsymbol{\theta}_f^\top \left( \sum_{h=1}^{\tilde{q}} \sigma_h \boldsymbol{a}_h \boldsymbol{b}_h^\top \right) \delta\boldsymbol{\theta}_g \\
&- \frac{\lambda}{n}\sum_{i=1}^n \frac{1}{q_i}\sum_{j=1}^{q_i} \left\| \nabla_{\boldsymbol{\theta}_f} f_k^{(i)}(\tilde{\boldsymbol{\theta}}_f) - \nabla_{\boldsymbol{\theta}_f} f_k^{(i)}(\boldsymbol{\theta}_{t,f}) \right\|_2 \left\| \nabla_{\boldsymbol{\theta}_g} g_{k,j}^{(i)}(\tilde{\boldsymbol{\theta}}_g)^\top \right\|_2 \|\delta\boldsymbol{\theta}_f\|_2 \|\delta\boldsymbol{\theta}_g\|_2 \\
&- \frac{\lambda}{n}\sum_{i=1}^n \frac{1}{q_i}\sum_{j=1}^{q_i}\sum_{k=1}^K \left\| \nabla_{\boldsymbol{\theta}_f} f_k^{(i)}(\boldsymbol{\theta}_{t,f}) \right\|_2 \left\| \nabla_{\boldsymbol{\theta}_g} g_{k,j}^{(i)}(\tilde{\boldsymbol{\theta}}_g) - \nabla_{\boldsymbol{\theta}_g} g_{k,j}^{(i)}(\boldsymbol{\theta}_{t,g}) \right\|_2 \delta\boldsymbol{\theta}_f^\top \|_2 \|\delta\boldsymbol{\theta}_g\|_2 \\
&\overset{(b)}{\geq} -\frac{\lambda}{n}\sum_{i=1}^n \frac{1}{q_i}\sum_{j=1}^{q_i} \left\| \nabla_{\boldsymbol{\theta}_f} f_k^{(i)}(\tilde{\boldsymbol{\theta}}_f) - \nabla_{\boldsymbol{\theta}_f} f_k^{(i)}(\boldsymbol{\theta}_{t,f}) \right\|_2 \left\| \nabla_{\boldsymbol{\theta}_g} g_{k,j}^{(i)}(\tilde{\boldsymbol{\theta}}_g)^\top \right\|_2 \|\delta\boldsymbol{\theta}_f\|_2 \|\delta\boldsymbol{\theta}_g\|_2 \\
&- \frac{\lambda}{n}\sum_{i=1}^n \frac{1}{q_i}\sum_{j=1}^{q_i}\sum_{k=1}^K \left\| \nabla_{\boldsymbol{\theta}_f} f_k^{(i)}(\boldsymbol{\theta}_{t,f}) \right\|_2 \left\| \nabla_{\boldsymbol{\theta}_g} g_{k,j}^{(i)}(\tilde{\boldsymbol{\theta}}_g) - \nabla_{\boldsymbol{\theta}_g} g_{k,j}^{(i)}(\boldsymbol{\theta}_{t,g}) \right\|_2 \|\delta\boldsymbol{\theta}_f\|_2 \|\delta\boldsymbol{\theta}_g\|_2 \\
&\overset{(c)}{\geq} -\left( \frac{\lambda(c^{(g)}\varrho^{(f)})}{\sqrt{m_f}} + \frac{\lambda(c^{(g)}\varrho^{(f)})}{\sqrt{m_f}} \right) \|\delta\boldsymbol{\theta}_f\|_2 \|\delta\boldsymbol{\theta}_g\|_2 \\
&\geq -\frac{1}{2}\left( \frac{\lambda(c^{(g)}\varrho^{(f)})}{\sqrt{m_f}} + \frac{\lambda(c^{(g)}\varrho^{(f)})}{\sqrt{m_f}} \right) \|\delta\boldsymbol{\theta}\|_2^2 \,,
\end{aligned}
$$

where (a) follows from the SVD in Definition 2, (b) follows since $\boldsymbol{\theta}' \in B^t$ and $\delta\boldsymbol{\theta} = \boldsymbol{\theta}' - \boldsymbol{\theta}_t$, by the properties of $Q_\kappa^t \subset B^t$, we have $\sum_h \sigma_h \langle \delta\boldsymbol{\theta}_f, \boldsymbol{a}_h \rangle \langle \delta\boldsymbol{\theta}_g, \boldsymbol{b}_h \rangle \geq 0$, and (c) follows Lemma D.3 and the fact that $\|\tilde{\boldsymbol{\theta}}_f - \boldsymbol{\theta}_{t,f}\|_2 \leq \|\delta\boldsymbol{\theta}_f\|_2$ and $\|\tilde{\boldsymbol{\theta}}_g - \boldsymbol{\theta}_{t,g}\|_2 \leq \|\delta\boldsymbol{\theta}_g\|_2$.

Next focusing on $I_3$, since we are using square loss, we have

$$I_3 = \delta\boldsymbol{\theta}_f^\top \left( \frac{1}{n} \sum_{i=1}^n \frac{1}{q_i} \sum_{j=1}^{q_i} \nabla_{\boldsymbol{\theta}_f} G_{\tilde{\boldsymbol{\theta}}}(u^{(i)})(y_j^{(i)}) \nabla_{\boldsymbol{\theta}_g} G_{\tilde{\boldsymbol{\theta}}}(u^{(i)})(y_j^{(i)})^\top \right) \delta\boldsymbol{\theta}_g$$

$$\overset{(b)}{\geq} \frac{1}{n} \sum_{i=1}^n \frac{1}{q_i} \sum_{j=1}^{q_i} \left\langle \delta\boldsymbol{\theta}_f, \nabla_{\boldsymbol{\theta}_f} G_{\tilde{\boldsymbol{\theta}}}(u^{(i)})(y_j^{(i)}) \right\rangle \left\langle \delta\boldsymbol{\theta}_g, \nabla_{\boldsymbol{\theta}_g} G_{\tilde{\boldsymbol{\theta}}}(u^{(i)})(y_j^{(i)}) \right\rangle$$

$$= \frac{1}{n} \sum_{i=1}^n \frac{1}{q_i} \sum_{j=1}^{q_i} \left\langle \delta\boldsymbol{\theta}_f, \nabla_{\boldsymbol{\theta}_f} G_{\boldsymbol{\theta}_t}(u^{(i)})(y_j^{(i)}) \right\rangle \left\langle \delta\boldsymbol{\theta}_g, \nabla_{\boldsymbol{\theta}_g} G_{\boldsymbol{\theta}_t}(u^{(i)})(y_j^{(i)}) \right\rangle$$

$$+ \frac{1}{n} \sum_{i=1}^n \frac{1}{q_i} \sum_{j=1}^{q_i} \left\langle \delta\boldsymbol{\theta}_f, \left( \nabla_{\boldsymbol{\theta}_f} G_{\tilde{\boldsymbol{\theta}}}(u^{(i)})(y_j^{(i)}) - \nabla_{\boldsymbol{\theta}_f} G_{\boldsymbol{\theta}_t}(u^{(i)})(y_j^{(i)}) \right) \right\rangle \left\langle \delta\boldsymbol{\theta}_g, \nabla_{\boldsymbol{\theta}_g} G_{\tilde{\boldsymbol{\theta}}}(u^{(i)})(y_j^{(i)}) \right\rangle$$

$$+ \frac{1}{n} \sum_{i=1}^n \frac{1}{q_i} \sum_{j=1}^{q_i} \left\langle \delta\boldsymbol{\theta}_f, \nabla_{\boldsymbol{\theta}_f} G_{\boldsymbol{\theta}_t}(u^{(i)})(y_j^{(i)}) \right\rangle \left\langle \delta\boldsymbol{\theta}_g, \left( \nabla_{\boldsymbol{\theta}_g} G_{\tilde{\boldsymbol{\theta}}}(u^{(i)})(y_j^{(i)}) - \nabla_{\boldsymbol{\theta}_g} G_{\boldsymbol{\theta}_t}(u^{(i)})(y_j^{(i)}) \right) \right\rangle$$

$$\geq \frac{1}{n} \sum_{i=1}^n \frac{1}{q_i} \sum_{j=1}^{q_i} \left\langle \delta\boldsymbol{\theta}_f, \nabla_{\boldsymbol{\theta}_f} G_{\boldsymbol{\theta}_t}(u^{(i)})(y_j^{(i)}) \right\rangle \left\langle \delta\boldsymbol{\theta}_g, \nabla_{\boldsymbol{\theta}_g} G_{\boldsymbol{\theta}_t}(u^{(i)})(y_j^{(i)}) \right\rangle$$

$$- \frac{1}{n} \sum_{i=1}^n \frac{1}{q_i} \sum_{j=1}^{q_i} \left\| \nabla_{\boldsymbol{\theta}_f} G_{\tilde{\boldsymbol{\theta}}}(u^{(i)})(y_j^{(i)}) - \nabla_{\boldsymbol{\theta}_f} G_{\boldsymbol{\theta}_t}(u^{(i)})(y_j^{(i)}) \right\|_2 \left\| \nabla_{\boldsymbol{\theta}_g} G_{\tilde{\boldsymbol{\theta}}}(u^{(i)})(y_j^{(i)}) \right\|_2 \|\delta\boldsymbol{\theta}_f\|_2 \langle \delta\boldsymbol{\theta}_g \|_2$$

$$- \frac{1}{n} \sum_{i=1}^n \frac{1}{q_i} \sum_{j=1}^{q_i} \left\| \nabla_{\boldsymbol{\theta}_f} G_{\boldsymbol{\theta}_t}(u^{(i)})(y_j^{(i)}) \right\|_2 \left\| \nabla_{\boldsymbol{\theta}_g} G_{\tilde{\boldsymbol{\theta}}}(u^{(i)})(y_j^{(i)}) - \nabla_{\boldsymbol{\theta}_g} G_{\boldsymbol{\theta}_t}(u^{(i)})(y_j^{(i)}) \right\|_2 \|\delta\boldsymbol{\theta}_f\|_2 \|\delta\boldsymbol{\theta}_g\|_2$$

$$\geq \frac{1}{n} \sum_{i=1}^n \frac{1}{q_i} \sum_{j=1}^{q_i} \left\langle \delta\boldsymbol{\theta}_f, \nabla_{\boldsymbol{\theta}_f} G_{\boldsymbol{\theta}_t}(u^{(i)})(y_j^{(i)}) \right\rangle \left\langle \delta\boldsymbol{\theta}_g, \nabla_{\boldsymbol{\theta}_g} G_{\boldsymbol{\theta}_t}(u^{(i)})(y_j^{(i)}) \right\rangle - \left( \frac{c^{(f)} \varrho^{(g)}}{\sqrt{m_f}} + \frac{c^{(g)} \varrho^{(f)}}{\sqrt{m_f}} \right) \|\delta\boldsymbol{\theta}_f\|_2 \|\delta\boldsymbol{\theta}_g\|_2$$

$$\geq \frac{1}{n} \sum_{i=1}^n \frac{1}{q_i} \sum_{j=1}^{q_i} \left\langle \delta\boldsymbol{\theta}_f, \nabla_{\boldsymbol{\theta}_f} G_{\boldsymbol{\theta}_t}(u^{(i)})(y_j^{(i)}) \right\rangle \left\langle \delta\boldsymbol{\theta}_g, \nabla_{\boldsymbol{\theta}_g} G_{\boldsymbol{\theta}_t}(u^{(i)})(y_j^{(i)}) \right\rangle - \frac{1}{2} \left( \frac{c^{(f)} \varrho^{(g)}}{\sqrt{m_f}} + \frac{c^{(g)} \varrho^{(f)}}{\sqrt{m_f}} \right) \|\delta\boldsymbol{\theta}\|_2^2 .$$

Combining the bounds on $T_1, T_2, T_3$ and using $m = m_g = m_f$, for a suitable constant $c_2$ based on $c^{(f)}, c^{(g)}, \varrho^{(f)}, \varrho^{(g)}, \lambda, \rho$, we have

$$\delta\boldsymbol{\theta}^\top \boldsymbol{H}(\tilde{\boldsymbol{\theta}})\delta\boldsymbol{\theta} \geq \frac{1}{n} \sum_{i=1}^n \frac{1}{q_i} \sum_{j=1}^{q_i} \left( \left\langle \delta\boldsymbol{\theta}_f, \nabla_{\boldsymbol{\theta}_f} G_{\boldsymbol{\theta}_t}(u^{(i)})(y_j^{(i)}) \right\rangle + \left\langle \delta\boldsymbol{\theta}_g, \nabla_{\boldsymbol{\theta}_g} G_{\boldsymbol{\theta}_t}(u^{(i)})(y_j^{(i)}) \right\rangle \right)^2 - \frac{c_2}{\sqrt{m}} \|\delta\boldsymbol{\theta}\|_2^2$$

$$\overset{(a)}{\geq} \left( \left\langle \delta\boldsymbol{\theta}_f, \nabla_{\boldsymbol{\theta}_f} \bar{G}_{\boldsymbol{\theta}_t}(u^{(i)})(y_j^{(i)}) \right\rangle + \left\langle \delta\boldsymbol{\theta}_g, \nabla_{\boldsymbol{\theta}_g} \bar{G}_{\boldsymbol{\theta}_t}(u^{(i)})(y_j^{(i)}) \right\rangle \right)^2 - \frac{c_2}{\sqrt{m}} \|\delta\boldsymbol{\theta}\|_2^2$$

$$= \left\langle \delta\boldsymbol{\theta}, \nabla_{\boldsymbol{\theta}} \bar{G}_{\boldsymbol{\theta}_t} \right\rangle^2 - \frac{c_2}{\sqrt{m}} \|\delta\boldsymbol{\theta}\|_2^2$$

$$\overset{(b)}{\geq} \kappa^2 \|\nabla_{\boldsymbol{\theta}} \bar{G}_{\boldsymbol{\theta}_t}\|_2^2 \|\delta\boldsymbol{\theta}\|_2^2 - \frac{c_2}{\sqrt{m}} \|\delta\boldsymbol{\theta}\|_2^2$$

$$= \alpha_t \|\delta\boldsymbol{\theta}\|_2^2 ,$$

where (a) follows from Jensen's inequality and with $\bar{G}_{\boldsymbol{\theta}} = \frac{1}{n} \sum_{i=1}^n \frac{1}{q_i} \sum_{j=1}^{q_i} G_{\boldsymbol{\theta}}(u^{(i)})(y_j^{(i)})$ as in Definition 2 and (b) follows from Definition 2, and $\alpha_t = \kappa^2 \|\nabla_{\boldsymbol{\theta}} \bar{G}_{\boldsymbol{\theta}}\|_2^2 - \frac{c_2}{\sqrt{m}}$. That completes the proof. $\qquad\square$

**Theorem 3 (Smoothness).** *Under the Assumptions 1 and 2, with probability at least $1 - \frac{4L}{m}$, for $\boldsymbol{\theta} \in B_{\rho, \rho_1}^{\mathrm{Euc}}(\bar{\boldsymbol{\theta}})$, $\mathcal{L}$ is $\beta$-smooth with $\beta = 4(K\bar{\lambda}^2 + \tilde{c})(\frac{\bar{\lambda} c}{\sqrt{m}} + \varrho) + 2K^2\bar{\lambda}^2 \varrho^2$ with $c = \max(c^{(f)}, c^{(g)}), \tilde{c} = \max_{ij} G^\dagger(u^{(i)})(y_j^{(i)}), \varrho = \max(\varrho^{(f)}, \varrho^{(g)}), \bar{\lambda} = \max(\lambda_1, \lambda_2)$ with $c^{(f)}, c^{(g)}, \varrho^{(f)}, \varrho^{(g)}, \lambda_1, \lambda_2$ as in Lemma D.3.*

*Proof.* By the second order Taylor expansion about $\bar{\boldsymbol{\theta}}$, we have $\mathcal{L}(\boldsymbol{\theta}') = \mathcal{L}(\bar{\boldsymbol{\theta}}) + \langle \boldsymbol{\theta}' - \bar{\boldsymbol{\theta}}, \nabla_{\boldsymbol{\theta}} \mathcal{L}(\bar{\boldsymbol{\theta}}) \rangle + \frac{1}{2}(\boldsymbol{\theta}' - \bar{\boldsymbol{\theta}})^{\top} \frac{\partial^2 \mathcal{L}(\tilde{\boldsymbol{\theta}})}{\partial \boldsymbol{\theta}^2}(\boldsymbol{\theta}' - \bar{\boldsymbol{\theta}})$, where $\tilde{\boldsymbol{\theta}} = \xi \boldsymbol{\theta}' + (1 - \xi)\bar{\boldsymbol{\theta}}$ for some $\xi \in [0, 1]$. Then,

$$(\boldsymbol{\theta}' - \bar{\boldsymbol{\theta}})^{\top} \frac{\partial^2 \mathcal{L}(\tilde{\boldsymbol{\theta}})}{\partial \boldsymbol{\theta}^2}(\boldsymbol{\theta}' - \bar{\boldsymbol{\theta}}) = (\boldsymbol{\theta}' - \bar{\boldsymbol{\theta}})^{\top} \bigg( \frac{1}{n} \sum_{i=1}^{n} \frac{1}{q_i} \sum_{j=1}^{q_i} \ell_{i,j}'' \nabla G_{\tilde{\boldsymbol{\theta}}}(u^{(i)})(y_j^{(i)}) \nabla G_{\tilde{\boldsymbol{\theta}}}(u^{(i)})(y_j^{(i)})^{\top}$$

$$+ \ell_{i,j}' \nabla^2 G_{\tilde{\boldsymbol{\theta}}}(u^{(i)})(y_j^{(i)}) \bigg)(\boldsymbol{\theta}' - \bar{\boldsymbol{\theta}})$$

$$= \underbrace{\frac{1}{n} \sum_{i=1}^{n} \frac{1}{q_i} \sum_{j=1}^{q_i} \ell_{i,j}'' \left\langle \boldsymbol{\theta}' - \bar{\boldsymbol{\theta}}, \nabla G_{\tilde{\boldsymbol{\theta}}}(u^{(i)})(y_j^{(i)}) \right\rangle^2}_{I_1}$$

$$+ \underbrace{\frac{1}{n} \sum_{i=1}^{n} \frac{1}{q_i} \sum_{j=1}^{q_i} \ell_{i,j}'(\boldsymbol{\theta}' - \bar{\boldsymbol{\theta}})^{\top} \nabla^2 G_{\tilde{\boldsymbol{\theta}}}(u^{(i)})(y_j^{(i)})(\boldsymbol{\theta}' - \bar{\boldsymbol{\theta}})}_{I_2} .$$

Now, note that

$$I_1 = \frac{1}{n} \sum_{i=1}^{n} \frac{1}{q_i} \sum_{j=1}^{q_i} \ell_{i,j}'' \left\langle \boldsymbol{\theta}' - \bar{\boldsymbol{\theta}}, \nabla G_{\tilde{\boldsymbol{\theta}}}(u^{(i)})(y_j^{(i)}) \right\rangle^2$$

$$\overset{(a)}{\leq} \frac{2}{n} \sum_{i=1}^{n} \frac{1}{q_i} \sum_{j=1}^{q_i} \left\| \nabla G_{\tilde{\boldsymbol{\theta}}}(u^{(i)})(y_j^{(i)}) \right\|_2^2 \|\boldsymbol{\theta}' - \bar{\boldsymbol{\theta}}\|_2^2$$

$$\overset{(b)}{\leq} 2K^2 \bar{\lambda}^2 \varrho^2 \|\boldsymbol{\theta}' - \bar{\boldsymbol{\theta}}\|_2^2 ,$$

where (a) follows by the Cauchy-Schwartz inequality and (b) from Lemma D.3 as follows:

$$\|G_{\tilde{\boldsymbol{\theta}}}(u^{(i)})(y_j^{(i)})\|_2 \leq \sum_{k=1}^{K} \left\| \begin{bmatrix} g_{k,j}^{(i)}(\tilde{\boldsymbol{\theta}}_g) \nabla_{\tilde{\boldsymbol{\theta}}_f} f_k^{(i)} \\ f_k^{(i)}(\tilde{\boldsymbol{\theta}}_f) \nabla_{\tilde{\boldsymbol{\theta}}_g} g_{k,j}^{(i)} \end{bmatrix} \right\|_2$$

$$\leq \sum_{k=1}^{K} \|\nabla_{\tilde{\boldsymbol{\theta}}_f} f_k^{(i)}\|_2 |g_{k,j}^{(i)}| + \|\nabla_{\tilde{\boldsymbol{\theta}}_g} g_{k,j}^{(i)}\|_2 |f_k^{(i)}| \leq K(\varrho^{(g)}\lambda_1 + \varrho^{(f)}\lambda_2) \leq K\bar{\lambda}\varrho.$$

For $I_2$, with $Q_{t,(i,j)} = (\boldsymbol{\theta}' - \bar{\boldsymbol{\theta}})^{\top} \nabla^2 G_{\tilde{\boldsymbol{\theta}}}(u^{(i)})(y_j^{(i)})(\boldsymbol{\theta}' - \bar{\boldsymbol{\theta}})$, we have

$$|Q_{t,(i,j)}| \leq \|\boldsymbol{\theta}' - \bar{\boldsymbol{\theta}}\|_2^2 \left\| \nabla^2 G_{\tilde{\boldsymbol{\theta}}}(u^{(i)})(y_j^{(i)}) \right\|_2 \leq 2(\bar{\lambda}\frac{c}{\sqrt{m}} + \varrho)\|\boldsymbol{\theta}' - \bar{\boldsymbol{\theta}}\|_2^2 ,$$

where the last inequality follows from

$$\left\| \nabla^2 G_{\tilde{\boldsymbol{\theta}}}(u^{(i)})(y_j^{(i)}) \right\|_2 \leq \left\| \begin{bmatrix} |g_{k,j}^{(i)}| \|\nabla_{\tilde{\boldsymbol{\theta}}_f}^2 f_k^{(i)}\|_2 & \nabla_{\tilde{\boldsymbol{\theta}}_f} f_k^{(i)} \nabla_{\tilde{\boldsymbol{\theta}}_g} g_{k,j}^{(i)}{}^{\top} \\ \nabla_{\tilde{\boldsymbol{\theta}}_g} g_{k,j}^{(i)} \nabla_{\tilde{\boldsymbol{\theta}}_f} f_k^{(i)}{}^{\top} & |f_k^{(i)}| \|\nabla_{\tilde{\boldsymbol{\theta}}_f}^2 g_{k,j}^{(i)}\|_2 \end{bmatrix} \right\|_2$$

$$\leq |g_{k,j}^{(i)}| \|\nabla_{\tilde{\boldsymbol{\theta}}_f}^2 f_k^{(i)}\|_2 + 2\|\nabla 2_{\tilde{\boldsymbol{\theta}}_f} f_k^{(i)}\|_2 \|\nabla 2_{\tilde{\boldsymbol{\theta}}_g} g_{k,j}^{(i)}\|_2 + |f_k^{(i)}| \|\nabla_{\tilde{\boldsymbol{\theta}}_g}^2 g_{k,j}^{(i)}\|_2 \leq 2(\bar{\lambda}\frac{c}{\sqrt{m}} + \varrho).$$

Then, we have

$$I_2 = \frac{1}{n} \sum_{i=1}^{n} \frac{1}{q_i} \sum_{j=1}^{q_i} \ell_{i,j}'(\boldsymbol{\theta}' - \bar{\boldsymbol{\theta}})^{\top} \nabla^2 G_{\tilde{\boldsymbol{\theta}}}(u^{(i)})(y_j^{(i)})(\boldsymbol{\theta}' - \bar{\boldsymbol{\theta}})$$

$$\leq 4(K\bar{\lambda}^2 + \tilde{c})(\frac{\bar{\lambda}c}{\sqrt{m}} + \varrho)\|\boldsymbol{\theta}' - \bar{\boldsymbol{\theta}}\|_2^2 ,$$

with $\tilde{c} = \max_{ij} G^{\dagger}(u^{(i)})(y_j^{(i)})$.

Putting the upper bounds on $I_1$ and $I_2$ back, we have

$$(\boldsymbol{\theta}' - \bar{\boldsymbol{\theta}})^\top \frac{\partial^2 \mathcal{L}(\tilde{\boldsymbol{\theta}})}{\partial \boldsymbol{\theta}^2}(\boldsymbol{\theta}' - \bar{\boldsymbol{\theta}}) \leq \left[ 4(K\bar{\lambda}^2 + \tilde{c})(\frac{\bar{\lambda}c}{\sqrt{m}} + \varrho) + 2K^2\bar{\lambda}^2\varrho^2 \right] \|\boldsymbol{\theta}' - \bar{\boldsymbol{\theta}}\|_2^2 .$$

This completes the proof. $\qquad\qquad\square$

# E   ANALYSIS FOR FOURIER NEURAL OPERATORS

**Theorem 6** (**Hessian Spectral Norm Bound**). *Under Assumptions 3 and 4 and for $\theta \in B_\rho^{Euc}(\theta_0)$, with probability at least $1 - \frac{2(L+2)}{m}$, for any $\boldsymbol{u}_i, i \in [n]$, we have*

$$\|\nabla_{\boldsymbol{\theta}}^2 G(\boldsymbol{\theta}; \boldsymbol{u}_i)\|_2 \leq \frac{c_H}{\sqrt{m}} \tag{35}$$

where $G_\theta(\cdot)$ is corresponds to the FNO predictor and $c_H = \ldots$. For definiteness, we recall the FNO model

$$\alpha^{(l)} = \phi\left( \frac{1}{\sqrt{m}}W^{(l)}\alpha^{(l-1)} + \frac{1}{\sqrt{m}}F^*R^{(l)}F\alpha^{(l-1)} \right), \quad l \in [L+1]$$

$$f := \alpha^{(L+2)} := \frac{1}{\sqrt{m}}\mathbf{v}^T\alpha^{(L+1)} , \tag{36}$$

where $W^{(l)}, R^{(l)} \in \mathbb{R}^{m \times m}$ for $l \in \{2, \ldots, L+1\}$, $W^{(1)} \in \mathbb{R}^{m \times d}$, $R^{(1)} = 0$ and $\alpha^{(0)} = \mathbf{x} \in \mathbb{R}^d$.

*Proof.* The proof follows as a direct result of Lemma E.1, E.2, E.3, E.4,

**Lemma E.1** (**Initialization of the Parameters**). *Consider the initialization of the parameters $w_{ij}^{(l)}$ by $w_{0,ij}^{(l)}$ and $R_{ij}^{(l)}$ by $R_{0,ij}^{(l)}$ respectively where $w_{0,ij}^{(l)} \sim \mathcal{N}\left(0, \sigma_{0_w}^2\right)$ and similarly $R_{0,ij}^{(l)} \sim \mathcal{N}\left(0, \sigma_{0_R}^2\right)$ with $\sigma_{0_w} = \dfrac{\sigma_{1_w}}{2(1 + \sqrt{\frac{\log m}{2m}})}$ and $\sigma_{0_R} = \dfrac{\sigma_{1_R}}{2(1 + \sqrt{\frac{\log m}{2m}})}$. Then, we have,*

$$\|W_0^{(l)}\|_2 \leq \sigma_{1_w}\sqrt{m}, \quad and \quad \|R_0^{(l)}\|_2 \leq \sigma_{1_R}\sqrt{m}. \tag{37}$$

*Proof.* The proof follows directly from Lemma A.1 in Banerjee et al. (2023). We reproduce it here for the sake of completeness. For $(m_l \times m_{l-1})$ random matrices $W_0^{(l)}$ and $R_0^{(l)}$ with i.i.d entries $w_{0,ij}^{(l)} \in \mathcal{N}(0, \sigma_{0,w}^2)$ and $r_{0,ij}^{(l)} \in \mathcal{N}(0, \sigma_{0,r}^2)$, the largest singular values are bounded from above with probabilities $(1 - 2\exp(-t^2/2\sigma_{0,w}^2))$ and $(1 - 2\exp(-t^2/2\sigma_{0,r}^2))$ respectively, namely,

$$\sigma_{\max}(W_0^{(l)}) \leq \sigma_{0,w}(\sqrt{m_l} + \sqrt{m_{l-1}}) + t, \quad and \quad \sigma_{\max}(R_0^{(l)}) \leq \sigma_{0,r}(\sqrt{m_l} + \sqrt{m_{l-1}}) + t. \tag{38}$$

In order to derive the above concentration result note that $W_0^{(l)} = \sigma_{0,w}\bar{W}_0^{(l)}$ and $R_0^{(l)} = \sigma_{0,r}\bar{R}_0^{(l)}$, where the entries $\bar{w}_{0,ij}^{(l)} \in \mathcal{N}(0, 1)$ and $\bar{r}_{0,ij}^{(l)} \in \mathcal{N}(0, 1)$. We can then write

$$\mathbb{E}[\|W_0^{(l)}\|_2] = \sigma_{0,w}\mathbb{E}[\|\bar{W}_0^{(l)}\|_2] = \sigma_{0,w}(\sqrt{m_l} + \sqrt{m_{l-1}}),$$

$$\mathbb{E}[\|R_0^{(l)}\|_2] = \sigma_{0,r}\mathbb{E}[\|\bar{R}_0^{(l)}\|_2] = \sigma_{0,r}(\sqrt{m_l} + \sqrt{m_{l-1}})$$

from Gordon's Theorem for Gaussian random matrices (see Theorem 5.32, Proposition 3.4 in (Vershynin, 2010)) where the function $f : B \to \|\sigma_0 B\|_2$ is a $\sigma_0$-Lipchitz function (where the matrix $B$ can be treated as a vector). Finally, choosing $t_w = \sigma_{0,w}\sqrt{2\log m}$ so that (38) holds with probability at least $\left(1 - \frac{2}{m}\right)$. In order to obtain the result in (37) we consider the following cases:

- **Case 1:** $l = 1$. With $m_0 = d_u$ and $m_1 = m$.

$$\left\|W_0^{(1)}\right\|_2 \leq \sigma_0(\sqrt{d} + \sqrt{m} + \sqrt{2\log m}) \leq \sigma_0(2\sqrt{m} + \sqrt{2\log m}),$$

$$R_0^{(1)} = 0.$$

- **Case 2:** $2 \leq l \leq L$. With $m_l = m_{l-1} = m$

$$\|W_0^{(l)}\|_2 \leq \sigma_{0,w} \left(2\sqrt{m} + \sqrt{2\log m}\right), \qquad \|R_0^{(l)}\|_2 \leq \sigma_{0,r} \left(2\sqrt{m} + \sqrt{2\log m}\right).$$

Now, using $\sigma_{0,w} = \dfrac{\sigma_{1,w}}{2(1 + \sqrt{\frac{\log m}{2m}})}$ and $\sigma_{0,r} = \dfrac{\sigma_{1,r}}{2(1 + \sqrt{\frac{\log m}{2m}})}$ completes the proof. $\quad\square$

**Proposition 2 (Layer-wise matrices).** *Under Assumptions 4 for $\theta \in B_\rho^{\mathrm{Euc}}(\theta_0)$, with probability at least $(1 - 2/m)$ we have*

$$\left\|W^{(l)}\right\|_2 \leq \left(\sigma_{1_w} + \frac{\rho_w}{\sqrt{m}}\right)\sqrt{m}, \quad \text{and} \quad \left\|R^{(l)}\right\|_2 \leq \left(\sigma_{1_R} + \frac{\rho_r}{\sqrt{m}}\right)\sqrt{m}, \ l \in [L+1] \quad (39)$$

*Proof.* By the virtue of triangle inequality, we have for $l \in [L+1]$

$$\|W^{(l)}\|_2 \leq \|W_0^{(l)}\|_2 + \|W^{(l)} - W_0^{(l)}\|_2 \overset{(a)}{\leq} \sigma_{1,w}\sqrt{m} + \rho_w,$$

$$\|R^{(l)}\|_2 \leq \|R_0^{(l)}\|_2 + \|R^{(l)} - R_0^{(l)}\|_2 \overset{(a)}{\leq} \sigma_{1,r}\sqrt{m} + \rho_r,$$

where (a) follows from Lemma E.1. $\quad\square$

**Remark 7.** Note that $R^{(1)} = 0$, so that Proposition 2 is trivially satisfied for it. $\quad\square$

We now show that the $L_2$ norm of the output at the layer $l$ of the FNO, i.e. $\alpha^{(l)}$, is bounded by $O(\sqrt{m})$.

**Lemma E.2 ($L_2$-norm of the output at $l$-th layer).** *Consider any $l \in [L+1]$. Under Assumptions 3 and 4 for $\theta \in B_\rho^{\mathrm{Euc}}$, with probability at least $(1 - \frac{2l}{m})$, we have*

$$\left\|\boldsymbol{\alpha}^{(l)}\right\|_2 \leq \sqrt{m}\left(\sigma_1 + \frac{\rho}{\sqrt{m}}\right)^l + \sqrt{m}\sum_{i=1}^{l}\left(\sigma_1 + \frac{\rho}{\sqrt{m}}\right)^{i-1}|\phi(0)| = \left(\gamma^l + |\phi(0)|\sum_{i=1}^{l}\gamma^{i-1}\right)\sqrt{m},$$

$$(40)$$

*where,*

$$\sigma_1 = \sigma_{1,w} + \sigma_{1,r}, \quad \text{and,} \quad \rho = \rho_w + \rho_r.$$

*Proof.* We prove the result using induction (see Lemma A.2 in Banerjee et al. (2023)). First, note that the input is normalized to have $\|\boldsymbol{u}\|_2 = \sqrt{d}$, which implies that $\|\alpha^{(0)}\|_2 = \sqrt{d}$ (note that $R^{(1)} = 0$). Then, using the fact that $m_0 = d_u$ and $\phi$ is 1-Lipchitz,

$$\left\|\phi\left(\frac{1}{\sqrt{d}}W^{(1)}\alpha^{(0)}\right)\right\|_2 - \|\phi(\mathbf{0})\|_2 \leq \left\|\phi\left(\frac{1}{\sqrt{d}}W^{(1)}\alpha^{(0)}\right) - \phi(\mathbf{0})\right\|_2 \leq \left\|\frac{1}{\sqrt{d}}W^{(1)}\alpha^{(0)}\right\|_2, \quad (41)$$

which in turn gives,

$$\begin{aligned}
\left\|\boldsymbol{\alpha}^{(1)}\right\|_2 &= \left\|\phi\left(\frac{1}{\sqrt{d_u}}W^{(1)}\boldsymbol{\alpha}^{(0)}\right)\right\|_2 \leq \left\|\frac{1}{\sqrt{d_u}}W^{(1)}\boldsymbol{\alpha}^{(0)}\right\|_2 + \|\phi(\mathbf{0})\|_2 \\
&\leq \frac{1}{\sqrt{d_u}}\left\|W^{(1)}\right\|_2\left\|\alpha^{(0)}\right\|_2 + |\phi(0)|\sqrt{m} \\
&\leq \left(\sigma_{1,w} + \frac{\rho_w}{\sqrt{m}}\right)\sqrt{m} + |\phi(0)|\sqrt{m} \\
&\leq \left(\sigma_{1,w} + \sigma_{1,r} + \frac{\rho_w + \rho_r}{\sqrt{m}}\right)\sqrt{m} + |\phi(0)|\sqrt{m}
\end{aligned}$$

where we use $m$ instead of $d_u$ to aid clarity in the subsequent steps below. Now, for completeness, consider also the output at layer 2, namely,

$$\|\boldsymbol{\alpha}^{(2)}\|_2 = \left\|\phi\left(\frac{1}{\sqrt{m}}W^{(2)}\alpha^{(1)} + \frac{1}{\sqrt{m}}F^*R^{(2)}F\alpha^{(1)}\right)\right\|_2,$$

which gives,

$$\left\| \phi\left( \frac{1}{\sqrt{m}} W^{(2)} \boldsymbol{\alpha}^{(1)} + \frac{1}{\sqrt{m}} F^* R^{(2)} F \boldsymbol{\alpha}^{(1)} \right) \right\|_2 - \|\phi(\mathbf{0})\|_2$$

$$\leq \left\| \phi\left( \frac{1}{\sqrt{m}} W^{(2)} \boldsymbol{\alpha}^{(1)} + \frac{1}{\sqrt{m}} F^* R^{(2)} F \boldsymbol{\alpha}^{(1)} \right) - \phi(\mathbf{0}) \right\|_2 \leq \left\| \frac{1}{\sqrt{m}} W^{(2)} \boldsymbol{\alpha}^{(1)} + \frac{1}{\sqrt{m}} F^* R^{(2)} F \boldsymbol{\alpha}^{(1)} \right\|_2,$$

and, in turn,

$$\|\boldsymbol{\alpha}^{(2)}\|_2 \leq \left\| \frac{1}{\sqrt{m}} W^{(2)} \boldsymbol{\alpha}^{(1)} + \frac{1}{\sqrt{m}} F^* R^{(2)} F \boldsymbol{\alpha}^{(1)} \right\|_2 + \|\phi(\mathbf{0})\|_2$$

$$\leq \left\| \frac{1}{\sqrt{m}} W^{(2)} \boldsymbol{\alpha}^{(1)} \right\|_2 + \left\| \frac{1}{\sqrt{m}} F^* R^{(2)} F \boldsymbol{\alpha}^{(1)} \right\|_2 + |\phi(0)|\sqrt{m}$$

$$\leq \frac{1}{\sqrt{m}} \|W^{(2)}\|_2 \|\boldsymbol{\alpha}^{(1)}\|_2 + \frac{1}{\sqrt{m}} \|R^{(2)}\|_2 \|\boldsymbol{\alpha}^{(1)}\|_2 + \sqrt{m}|\phi(0)|$$

$$\leq \left( \sigma_{1,w} + \frac{\rho_w}{\sqrt{m}} + \sigma_{1,r} + \frac{\rho_r}{\sqrt{m}} \right) \|\boldsymbol{\alpha}^{(1)}\|_2 + \sqrt{m}|\phi(0)|$$

$$\leq \sqrt{m} \left( \sigma_1 + \frac{\rho}{\sqrt{m}} \right)^2 + \left( 1 + \left( \sigma_1 + \frac{\rho}{\sqrt{m}} \right) \right) \sqrt{m}|\phi(0)|.$$

Now, for the inductive step, consider that the output at layer $l - 1$ satisfies

$$\left\| \boldsymbol{\alpha}^{(l-1)} \right\|_2 \leq \sqrt{m} \left( \sigma_{1,w} + \sigma_{1,r} + \frac{\rho_w + \rho_r}{\sqrt{m}} \right)^{l-1} + \sqrt{m} \sum_{i=1}^{l-1} \left( \sigma_{1,w} + \sigma_{1,r} + \frac{\rho_w + \rho_r}{\sqrt{m}} \right)^{i-1} |\phi(0)|.$$

Finally, at layer $l$, we have

$$\left\| \boldsymbol{\alpha}^{(l)} \right\|_2 \leq \frac{1}{\sqrt{m}} \left( \underbrace{\left\| W^{(l)} \right\|_2 + \left\| F^* R^{(l)} F \right\|_2} \right) \|\boldsymbol{\alpha}^{(l-1)}\|_2 + \sqrt{m}|\phi(0)| \tag{42}$$

$$\leq \left( \sigma_{1,w} + \sigma_{1,r} + \frac{\rho_w + \rho_r}{\sqrt{m}} \right) \|\boldsymbol{\alpha}^{(l-1)}\|_2 + \sqrt{m}|\phi(0)| \tag{43}$$

$$\leq \sqrt{m} \left( \sigma_{1,w} + \sigma_{1,r} + \frac{\rho_w + \rho_r}{\sqrt{m}} \right)^l + \sqrt{m} \sum_{i=1}^{l} \left( \sigma_1 + \frac{\rho}{\sqrt{m}} \right)^{i-1} |\phi(0)|. \tag{44}$$

Introducing $\gamma = \sigma_1 + \frac{\rho}{\sqrt{m}}$, we can write

$$\|\boldsymbol{\alpha}^{(l)}\|_2 \leq \sqrt{m} \left( \gamma^l + |\phi(0)| \sum_{i=1}^{l} \gamma^{i-1} \right). \tag{45}$$

This completes the proof. $\qquad\square$

**Lemma E.3.** *For $l \in \{2, \ldots, L+1\}$, under Assumptions 3 and 4 for $\theta \in B_\rho^{\mathrm{Euc}}(\theta_0)$, with probability at least $(1 - \frac{2}{m}])$, we have,*

$$\left\| \frac{\partial \boldsymbol{\alpha}^{(l)}}{\partial \boldsymbol{\alpha}^{(l-1)}} \right\|_2^2 \leq \left( \sigma_{1,w} + \frac{\rho_w}{\sqrt{m}} \right)^2 + \leq \left( \sigma_{1,r} + \frac{\rho_r}{\sqrt{m}} \right)^2 = \gamma_w^2 + \gamma_r^2. \tag{46}$$

*Proof.* We have

$$\left[ \frac{\partial \boldsymbol{\alpha}^{(l)}}{\partial \boldsymbol{\alpha}^{(l-1)}} \right]_{ij} = \frac{1}{\sqrt{m}} \phi'(\widetilde{\boldsymbol{\alpha}}^{(l-1)}) \left[ W_{ij}^{(l)} + [F^* R^{(l)} F]_{ij} \right].$$

Now, $\|A\|_2 = \sup_{\|v\|_2=1} \|Av\|_2$ we have,

$$
\begin{aligned}
\left\|\frac{\partial \boldsymbol{\alpha}^{(l)}}{\partial \boldsymbol{\alpha}^{(l-1)}}\right\|_2^2 &= \sup_{\|\boldsymbol{v}\|_2=1} \frac{1}{m} \left(\phi'^2 \left\|\left(W^{(l)} + F^* R^{(l)} F\right) \boldsymbol{v}\right\|_2^2\right) \\
&\overset{(a)}{\leq} \sup_{\|\boldsymbol{v}\|_2=1} \frac{1}{m} \left(\|W^{(l)}\boldsymbol{v}\|_2^2 + \|F^* R^{(l)} F\boldsymbol{v}\|_2^2\right) \overset{(b)}{=} \sup_{\|\boldsymbol{v}\|_2=1} \frac{1}{m} \left(\|W^{(l)}\boldsymbol{v}\|_2^2 + \|R^{(l)}\boldsymbol{v}\|_2^2\right),
\end{aligned}
\tag{47}
$$

where $(a)$ follows from the fact that $\phi$ is 1-Lipchitz and using the triangle inequality, and $(b)$ follows from the fact that $F^*$ and $F$ are isometries wrt the $L_2$ norm, i.e. $\|F\boldsymbol{v}\|_2^2 = \|v\|_2^2$ and $\|F^*\boldsymbol{v}\|_2^2 = \|v\|_2^2$. This finally gives

$$
\begin{aligned}
\left\|\frac{\partial \boldsymbol{\alpha}^{(l)}}{\partial \boldsymbol{\alpha}^{(l-1)}}\right\|_2^2 &\leq \frac{1}{m} \left(\|W^{(l)}\|_2^2 + \|R^{(l)}\|_2^2\right) = \left(\sigma_{1,w} + \frac{\rho_w}{\sqrt{m}}\right)^2 + \left(\sigma_{1,r} + \frac{\rho_r}{\sqrt{m}}\right)^2 \\
&= \gamma_w^2 + \gamma_r^2.
\end{aligned}
$$

This completes the proof. $\qquad\square$

**Lemma E.4.** *Consider an arbitrary layer $l \in [L+1]$ and the gradient of the output at layer $l$ with respect to the parameters, i.e.,*

$$
\boldsymbol{g}^{(l)} = \begin{bmatrix} \dfrac{\partial \boldsymbol{\alpha}^{(l)}}{\partial \boldsymbol{w}^{(l)}} \\[2mm] \dfrac{\partial \boldsymbol{\alpha}^{(l)}}{\partial \boldsymbol{r}^{(l)}} \end{bmatrix},
$$

*where, $\boldsymbol{w}^{(l)} = vec(W^{(l)})$ and $\boldsymbol{r}^{(l)} = vec(R^{(l)})$. Under Assumptions 3 and 4 and for $\theta \in B_\rho^{Euc}(\theta_0)$, with probability at least $\left(1 - \frac{2l}{m}\right)$,*

$$
\|\boldsymbol{g}^{(l)}\|_2^2 \leq 2 \left(\gamma^{(l-1)} + |\phi(0)| \sum_{i=1}^{l-1} \gamma^{(i-1)}\right)^2
$$

*Proof.* We can index the vectors $\boldsymbol{w}^{(l)}$ and $\boldsymbol{r}^{(l)}$ using $j \in [m]$ and $j' \in [d_u]$ for $l = 1$ and $j' \in [m]$ for $l \in \{2, \dots, L+1\}$. Therefore,

$$
\left[\frac{\partial \alpha^{(l)}}{\partial w^{(l)}}\right]_{i,jj'} = \frac{1}{\sqrt{m}} \phi'(\widetilde{\alpha}_i^{(l)}) \delta_{ij} \alpha_{j'}^{(l-1)}, \quad \delta_{ij} = \begin{cases} 1 & i = j \\ 0 & \text{otherwise} \end{cases}.
$$

Now, for $l \in \{2, \dots, L+1\}$, we can write the 2-norm of the matrices as follows

$$
\begin{aligned}
\left\|\frac{\partial \boldsymbol{\alpha}^{(l)}}{\partial \boldsymbol{w}^{(l)}}\right\|_2^2 &= \sup_{\|V\|_F=1} \frac{1}{m} \sum_{i=1}^{m} \left(\phi'\left(\widetilde{\alpha}_i^{(l)}\right) \sum_{j,j'=1}^{m} \alpha_{j'}^{(l-1)} \delta_{ij} V_{jj'}\right)^2 \\
&\leq \sup_{\|V\|_F=1} \frac{1}{m} \|V\boldsymbol{\alpha}^{(l-1)}\|_2^2 \\
&\leq \sup_{\|V\|_F=1} \frac{1}{m} \|V\|_2^2 \|\boldsymbol{\alpha}^{(l-1)}\|_2^2 \\
&\overset{(a)}{\leq} \sup_{\|V\|_F=1} \frac{1}{m} \|V\|_F^2 \|\boldsymbol{\alpha}^{(l-1)}\|_2^2 \\
&\leq \frac{1}{m} \|\boldsymbol{\alpha}^{(l-1)}\|_2^2 \\
&\overset{(b)}{\leq} \frac{1}{m} \left[\sqrt{m} \left(\gamma^{l-1} + |\phi(0)| \sum_{i=1}^{l-1} \gamma^{i-1}\right)\right]^2 = \left(\gamma^{l-1} + |\phi(0)| \sum_{i=1}^{l-1} \gamma^{i-1}\right)^2,
\end{aligned}
$$

where, $(a)$ follows from the fact that $\|V\|_2^2 \leq \|V\|_F^2$ and $(b)$ from (45). The $l = 1$ case follows in a similar fashion:

$$\left\|\frac{\partial \alpha^{(1)}}{\partial \mathbf{w}^{(1)}}\right\|_2^2 \leq \frac{1}{d_u}\|\boldsymbol{\alpha}^{(0)}\|_2^2 = \frac{1}{d_u}\|\boldsymbol{u}\|_2^2 = 1, \quad \left\|\frac{\partial \alpha^{(1)}}{\partial \mathbf{r}^{(1)}}\right\|_2^2 = 0.$$

Similarly,

$$
\begin{aligned}
\left\|\frac{\partial \alpha^{(l)}}{\partial \mathbf{r}^{(l)}}\right\|_2^2 &= \sup_{\|V\|_F=1} \frac{1}{m}\sum_{i=1}^m \left(\phi'\left(\widetilde{\alpha}_i^{(l)}\right) F_{ij}^* F_{j'p}\alpha_p^{(l-1)}V_{jj'}\right)^2 \\
&\leq \sup_{\|V\|_F=1} \frac{1}{m}\|(F^*VF)\boldsymbol{\alpha}^{(l-1)}\|_2^2 \\
&\leq \sup_{\|V\|_F=1} \frac{1}{m}\|F^*VF\|_2^2\|\boldsymbol{\alpha}^{(l-1)}\|_2^2 \\
&\leq \sup_{\|V\|_F=1} \frac{1}{m}\|F^*\|_2^2\|V\|_2^2\|F\|_2^2\|\boldsymbol{\alpha}^{(l-1)}\|_2^2 \\
&\overset{(a)}{\leq} \sup_{\|V\|_F=1} \frac{1}{m}\|V\|_F^2\|\boldsymbol{\alpha}^{(l-1)}\|_2^2 \\
&\leq \frac{1}{m}\|\boldsymbol{\alpha}^{(l-1)}\|_2^2 \\
&\overset{(b)}{\leq} \frac{1}{m}\left[\sqrt{m}\left(\gamma^{l-1} + |\phi(0)|\sum_{i=1}^{l-1}\gamma^{i-1}\right)\right]^2 = \left(\gamma^{l-1} + |\phi(0)|\sum_{i=1}^{l-1}\gamma^{i-1}\right)^2,
\end{aligned}
$$

where $(a)$ follows again by $\|V\|_2^2 \leq \|V\|_F^2$ and the fact that $F^*$ and $F$ are unitary matrices, and $(b)$ from (45). Now, finally

$$\|\boldsymbol{g}\|_2^2 = \left\|\frac{\partial \alpha^{(l)}}{\partial \mathbf{w}^{(l)}}\right\|_2^2 + \left\|\frac{\partial \alpha^{(l)}}{\partial \mathbf{r}^{(l)}}\right\|_2^2 \leq 2\left(\gamma^{l-1} + |\phi(0)|\sum_{i=1}^{l-1}\gamma^{i-1}\right)^2. \tag{48}$$

This completes the proof $\qquad\square$

We now focus on bounding the hessian of the predictor (36). Note that the FNO model can be considered having $(L+1)$ layers, with Layer 1 being the feedforward single layer encoder, the $L$ layers from 2 to $L+1$ being FNO layers, and Layer $(L+2)$ being the output of the linear decoder. Furthermore, we make use of Einstein summation convention, i.e. repeated indices imply summation, unless explicitly stated.

The Hessian matrix $H$ for the $L$ FNO layers can be viewed as $2 \times 2$ block matrix, namely,

$$H = \begin{bmatrix} H_w^{(l_1,l_2)} & H_{w,r}^{(l_1,l_2)} \\ H_{r,w}^{(l_1,l_2)} & H_r^{(l_1,l_2)} \end{bmatrix}$$

where

- the $(1,1)$ block has $L \times L$ sub-blocks corresponding to $H_w^{((l_1,l_2)} := \frac{\partial^2 f}{\partial \mathbf{w}^{(l_1)}\partial \mathbf{w}^{(l_2)}}$ for $l_1, l_2 \in \{2, \ldots, L+1\}$,

- the $(2,2)$ block has $L \times L$ sub-blocks corresponding to $H_r^{((l_1,l_2)} := \frac{\partial^2 f}{\partial \mathbf{r}^{(l_1)}\partial \mathbf{r}^{(l_2)}}$ for $l_1, l_2 \in \{2, \ldots, L+1\}$, and

- the $(1,2)$ and $(2,1)$ cross blocks have terms of the form $H_{w,r}^{((l_1,l_2)} := \frac{\partial^2 f}{\partial \mathbf{w}^{(l_1)}\partial \mathbf{r}^{(l_2)}}$ for $l_1, l_2 \in \{2, \ldots, L+1\}$.

There are additional blocks corresponding to $W^{(1)}$, including

- diagonal block $H_w^{((1,1))} := \frac{\partial^2 f}{\partial \mathbf{w}^{(1)2}}$,

- off-diagonal blocks $H_w^{((1,l_1))} := \frac{\partial^2 f}{\partial \mathbf{w}^{(1)} \partial \mathbf{w}^{(l_1)}}$ for $l_1 \in \{2, \ldots, L+1\}$, as well as $H_w^{((l_1,1))}$, and

- off-diagonal blocks $H_{w,r}^{(1,l_2)} := \frac{\partial^2 f}{\partial \mathbf{w}^{(1)} \partial \mathbf{r}^{(l_2)}}$ for $l_2 \in \{2, \ldots, L+1\}$, as well as $H_{r,w}^{((l_2,1))}$.

Further, there are additional blocks corresponding to $v$, including

- diagonal block $H_v := \frac{\partial^2 f}{\partial \mathbf{v}^2}$, which is 0,

- off-diagonal block $H_{v,w}^{((l_1))} := \frac{\partial^2 f}{\partial \mathbf{v} \partial \mathbf{w}^{(l_1)}}$ for $l_1 \in \{1, \ldots, L+1\}$, as well as $H_{w,v}^{((l_1))}$, and

- off-diagonal block $H_{v,r}^{(l_2)} := \frac{\partial^2 f}{\partial \mathbf{v} \partial \mathbf{r}^{(l_2)}}$ for $l_2 \in \{2, \ldots, L+1\}$, as well as $H_{v,w}^{((l_1))}$.

**Gradients.** The gradient of $f$ wrt any $\mathbf{w}^{(l_1)}$ and any $\mathbf{r}^{(l_1)}$, $l_1 \in [L+1]$ and for any $l_1, l_2 \leq l \leq L$, is given by

$$\frac{\partial f}{\partial \mathbf{w}^{(l_1)}} = \left( \frac{\partial \alpha^{(l_1)}}{\partial \mathbf{w}^{(l_1)}} \prod_{l'=l_1+1}^{l} \frac{\partial \alpha^{(l')}}{\partial \alpha^{(l'-1)}} \right) \frac{\partial f}{\partial \alpha^{(l)}} , \tag{49}$$

$$\frac{\partial f}{\partial \mathbf{r}^{(l_2)}} = \left( \frac{\partial \alpha^{(l_2)}}{\partial \mathbf{r}^{(l_2)}} \prod_{l'=l_2+1}^{l} \frac{\partial \alpha^{(l')}}{\partial \alpha^{(l'-1)}} \right) \frac{\partial f}{\partial \alpha^{(l)}} . \tag{50}$$

Note that the choice of $l$ with $l_1 \leq l \leq L$ gives different forms of the recurive decomposition. Further, as typical, $\prod_a^b$ with $a > b$ is simply a 1, i.e., the term does not affect the analysis.

For the analysis, for convenience, let $l_1, l_2$ respectively be the index for $w, r$ when both $w, r$ are being considered, and we will consider both $l_1 \leq l_2$ and $l_1 \geq l_2$, which suffices due to the symmetry of $H$.

First, note that

$$\|H\|_2 \leq \sum_{l_1,l_2=1}^{L+1} \|H_w^{((l_1,l_2))}\|_2 + \sum_{l_1,l_2=2}^{L+1} \|H_r^{((l_1,l_2))}\|_2 + 2 \sum_{l_1=1}^{L+1} \sum_{l_2=2}^{L+1} \|H_{w,r}^{((l_1,l_2))}\|_2 + 2 \sum_{l_1=1}^{L+1} \|H_{v,w}^{(l_1)}\|_2 + 2 \sum_{l_2=2}^{L+1} \|H_{v,r}^{(l_2)}\|_2 . \tag{51}$$

**Diagonal blocks.** Note that the analysis for $\|H_w^{((l_1,l_2))}\|_2$ and $\|H_r^{((l_1,l_2))}\|_2$ terms follow exactly from the (Liu et al., 2021a).

**Off-Diagonal blocks.** For the off-diagonal blocks, we focus on bounding $\|H_{w,r}^{((l_1,l_2))}\|_2$ for (Case 1.A) $l_1 \leq l_2$, (Case 1.B) $l_2 \leq l_1$. Further, we bound (Case 2.A) $\|H_{v,w}^{(l_1)}\|_2$ and (Case 2.B) $\|H_{v,r}^{(l_2)}\|_2$.

We define

$$\mathcal{Q}_\infty^{(w,r)}(f) := \max_{l \in [L+1]} \left\{ \left\| \frac{\partial f}{\partial \alpha^{(l)}} \right\|_\infty \right\} ,$$

$$\mathcal{Q}_2^{(w,r)}(f) := \max_{l \in [L+1]} \left\{ \left\| \frac{\partial \alpha^{(l)}}{\partial \mathbf{w}^{(l)}} \right\|_2 , \left\| \frac{\partial \alpha^{(l)}}{\partial \mathbf{r}^{(l)}} \right\|_2 \right\} ,$$

$$\mathcal{Q}_{2,2,1}^{(w,r)}(f) := \max_{1 \leq l_1 \leq l_2 \leq l_3 \leq L} \left\{ \left\| \frac{\partial^2 \alpha^{(l_1)}}{\partial \mathbf{w}^{(l_1)} \partial \mathbf{r}^{(l_1)}} \right\|_{2,2,1} , \left\| \frac{\partial \alpha^{(l_1)}}{\partial \mathbf{w}^{(l_1)}} \right\|_2 \left\| \frac{\partial^2 \alpha^{(l_2)}}{\partial \alpha^{(l_2-1)} \partial \mathbf{r}^{(l_2)}} \right\|_{2,2,1} , \left\| \frac{\partial \alpha^{(l_1)}}{\partial \mathbf{r}^{(l_1)}} \right\|_2 \left\| \frac{\partial^2 \alpha^{(l_2)}}{\partial \alpha^{(l_2-1)} \partial \mathbf{w}^{(l_2)}} \right\|_{2,2,1} , \right.$$

$$\left. \left\| \frac{\partial \alpha^{(l_1)}}{\partial \mathbf{w}^{(l_1)}} \right\|_2 \left\| \frac{\partial \alpha^{(l_2)}}{\partial \mathbf{w}^{(l_2)}} \right\|_2 \left\| \frac{\partial^2 \alpha^{(l_3)}}{(\partial \alpha^{(l_3-1)})^2} \right\|_{2,2,1} \right\} . \tag{52}$$

Let $L_\phi$ denote the layerwise Lipschitz constant, i.e., $\max_{l\in[L-1]} \|\frac{\partial \alpha^{(l+1)}}{\partial \alpha^{(l)}}\|_2 \le L_\phi$.

**Case 1.A:** $1 \le l_1 \le l_2 \le L$. By building on the form of the gradient, we have

$$
H_{w,r}^{(l_1,l_2)} = \frac{\partial^2 \alpha^{(l_1)}}{\partial \mathbf{w}^{(l_1)} \partial \mathbf{r}^{(l_1)}} \frac{\partial f}{\partial \alpha^{(l_1)}} \mathbf{1}_{[l_1=l_2]} + \left( \frac{\partial \alpha^{(l_1)}}{\partial \mathbf{w}^{(l_1)}} \prod_{l'=l_1+1}^{l_2-1} \frac{\partial \alpha^{(l')}}{\partial \alpha^{(l'-1)}} \right) \frac{\partial^2 \alpha^{(l_2)}}{\partial \alpha^{(l_2-1)} \partial \mathbf{r}^{(l_2)}} \left( \frac{\partial f}{\partial \alpha^{(l_2)}} \right)
$$
$$
+ \sum_{l=l_2+1}^{L} \left( \frac{\partial \alpha^{(l_1)}}{\partial \mathbf{w}^{(l_1)}} \prod_{l'=l_1+1}^{l-1} \frac{\partial \alpha^{(l')}}{\partial \alpha^{(l'-1)}} \right) \left( \frac{\partial \alpha^{(l_2)}}{\partial \mathbf{r}^{(l_2)}} \prod_{l'=l_2+1}^{l-1} \frac{\partial \alpha^{(l')}}{\partial \alpha^{(l'-1)}} \right) \frac{\partial^2 \alpha^{(l)}}{(\partial \alpha^{(l-1)})^2} \left( \frac{\partial f}{\partial \alpha^{(l)}} \right) .
$$

Then,

$$
\|H_{w,r}^{(l_1,l_2)}\|_2 \le \left\| \frac{\partial^2 \alpha^{(l_1)}}{\partial \mathbf{w}^{(l_1)} \partial \mathbf{r}^{(l_1)}} \right\|_{2,2,1} \left\| \frac{\partial f}{\partial \alpha^{(l_1)}} \right\|_\infty + \left\| \frac{\partial \alpha^{(l_1)}}{\partial \mathbf{w}^{(l_1)}} \right\|_2 \prod_{l'=l_1+1}^{l_2-1} \left\| \frac{\partial \alpha^{(l')}}{\partial \alpha^{(l'-1)}} \right\|_2 \left\| \frac{\partial^2 \alpha^{(l_2)}}{\partial \alpha^{(l_2-1)} \partial \mathbf{r}^{(l_2)}} \right\|_{2,2,1} \left\| \frac{\partial f}{\partial \alpha^{(l_2)}} \right\|_\infty
$$
$$
+ \sum_{l=l_2+1}^{L} \left( \left\| \frac{\partial \alpha^{(l_1)}}{\partial \mathbf{w}^{(l_1)}} \right\|_2 \prod_{l'=l_1+1}^{l-1} \left\| \frac{\partial \alpha^{(l')}}{\partial \alpha^{(l'-1)}} \right\|_2 \right) \left( \left\| \frac{\partial \alpha^{(l_2)}}{\partial \mathbf{r}^{(l_2)}} \right\|_2 \prod_{l'=l_2+1}^{l-1} \left\| \frac{\partial \alpha^{(l')}}{\partial \alpha^{(l'-1)}} \right\|_2 \right) \left\| \frac{\partial^2 \alpha^{(l)}}{(\partial \alpha^{(l-1)})^2} \right\|_{2,2,1} \left\| \frac{\partial f}{\partial \alpha^{(l)}} \right\|_\infty
$$
$$
\le \left\| \frac{\partial^2 \alpha^{(l_1)}}{\partial \mathbf{w}^{(l_1)} \partial \mathbf{r}^{(l_1)}} \right\|_{2,2,1} \left\| \frac{\partial f}{\partial \alpha^{(l_1)}} \right\|_\infty + L_\phi^{l_2-l_1-1} \left\| \frac{\partial \alpha^{(l_1)}}{\partial \mathbf{w}^{(l_1)}} \right\|_2 \left\| \frac{\partial^2 \alpha^{(l_2)}}{\partial \alpha^{(l_2-1)} \partial \mathbf{r}^{(l_2)}} \right\|_{2,2,1} \left\| \frac{\partial f}{\partial \alpha^{(l_2)}} \right\|_\infty
$$
$$
+ \sum_{l=l_2+1}^{L} L_\phi^{2l-l_2-l_1} \left\| \frac{\partial \alpha^{(l_1)}}{\partial \mathbf{w}^{(l_1)}} \right\|_2 \left\| \frac{\partial \alpha^{(l_2)}}{\partial \mathbf{r}^{(l_2)}} \right\|_2 \left\| \frac{\partial^2 \alpha^{(l)}}{(\partial \alpha^{(l-1)})^2} \right\|_{2,2,1} \left\| \frac{\partial f}{\partial \alpha^{(l)}} \right\|_\infty .
$$

Then, based on the definitions in (52), we have

$$
\|H_{w,r}^{(l_1,l_2)}\|_2 \le C_1' \mathcal{Q}_{2,2,1}^{w,r}(f) \mathcal{Q}_\infty^{w,r}(f) ,
$$

where $C_1'$ is a suitable constant.

**Lemma E.5.** *Under Assumptions 3 and 4 for $\boldsymbol{\theta} \in B_\rho^{\mathrm{Euc}}(\boldsymbol{\theta}_0)$, the following inequalities hold with probability at least $(1 - \frac{2(L+2)}{m}])$.*

$$
\left\| \frac{\partial^2 \boldsymbol{\alpha}^{(l_1)}}{\partial \mathbf{w}^{(l_1)} \partial \boldsymbol{r}^{(l_1)}} \right\|_{2,2,1}^2 \le \beta_\phi \left( \gamma^{l_1-1} + |\phi(0)| \sum_{i=1}^{l_1-1} \gamma^{i-1} \right)^2 \tag{53}
$$

$$
\left\| \frac{\partial^2 \boldsymbol{\alpha}^{(l)}}{\partial \boldsymbol{\alpha}^{(l-1)2}} \right\|_{2,2,1}^2 \le \dots \tag{54}
$$

$$
\left\| \frac{\partial^2 \boldsymbol{\alpha}^{(l_2)}}{\partial \boldsymbol{\alpha}^{(l_2-1)} \partial \boldsymbol{r}^{(l_2)}} \right\|_{2,2,1}^2 \le \beta_\phi \left( \gamma^2 + (\gamma^{l_2-1} + |\phi(0)| \sum_{i=1}^{l_2-1} \gamma^{i-1}) \right) + 1 \tag{55}
$$

*Proof.* We first begin with proving (53). Note that from (36) we have

$$
\frac{\partial^2 \alpha_i^{(l_l)}}{\partial w_{jj'}^{(l_1)} \partial r_{kk'}^{(l_1)}} = \frac{1}{m} \phi'' \left( \tilde{\alpha}^{(l_1)} \right) \cdot \alpha_{j'}^{(l_1-1)} \delta_{ij} F_{ik}^* F_{k'q} \alpha_q^{(l_1-1)},
$$

where we make use of Einstein notation and there is no summation on the index $i$. Now,

$$
\begin{aligned}
\left\| \frac{\partial^2 \alpha_i^{(l_l)}}{\partial w_{jj'}^{(l_1)} \partial r_{kk'}^{(l_1)}} \right\|_{2,2,1}
&= \sup_{\|V_1\|_F=1, \|V_2\|_F=1} \sum_{i=1}^m \left| \frac{1}{m} \phi''(\alpha_i^{(\tilde{l}_1)}) \alpha_{j'}^{(l_1-1)} \delta_{ij} F_{ik}^* F_{k'q} \alpha_q^{(l_i-1)} V_{1_{jj'}} V_{2_{kk'}} \right| \\
&= \sup_{\|V_1\|_F=1, \|V_2\|_F=1} \sum_{i=1}^m \left| \frac{\phi''}{m} \left( V_{1_{ij'}} \alpha_{j'}^{(l_1-1)} \right) \left( F_{ik}^* V_{2_{kk'}} F_{k'q} \alpha_q^{(l_1-1)} \right) \right| \\
&= \sup_{\|V_1\|_F=1, \|V_2\|_F=1} \left| \left\langle V_1 \boldsymbol{\alpha}^{(l_1-1)}, (F^* V_2 F) \boldsymbol{\alpha}^{(l_1-1)} \right\rangle \right| \\
&\overset{(a)}{\leq} \sup_{\|V_1\|_F=1, \|V_2\|_F=1} \frac{\beta_\phi}{2m} \left( \left\| V_1 \boldsymbol{\alpha}^{(l_1-1)} \right\|_2^2 + \left\| F^H V_2 F \boldsymbol{\alpha}^{(l-1)} \right\|_2^2 \right) \\
&\overset{(b)}{\leq} \frac{\beta_\phi}{2m} \left( \left\| \alpha^{(l_1-1)} \right\|_2^2 + \left\| \alpha^{(l_1-1)} \right\|_2^2 \right) = \beta_\phi \left( \gamma^{l_1-1} + |\phi(0)| \sum_{i=1}^{l_1-1} \gamma^{i-1} \right)^2,
\end{aligned}
$$

where $(a)$ follows from: $|V_1 \boldsymbol{\alpha}^{(l_1-1)}|_1^2 \leq \|V_1 \boldsymbol{\alpha}\|_2^2 \leq \|V_1\|_2^2 \|\boldsymbol{\alpha}\|_2^2$ and $\|V_1\|_2^2 \leq \|V_1\|_F^2$ and $(b)$ follows from (45). This completes the proof for (53). $\qquad\square$

For proving (54), again note from (36) that

$$
\begin{aligned}
\left[ \frac{\partial^2 \alpha^{(l)}}{\partial \alpha^{(l-1)^2}} \right]_{i,j,k}
&= \frac{1}{m} \phi''(\tilde{\alpha}^{(l)}) \left( W_{ij}^{(l)} + F_{ip}^* R_{pq}^{(l)} F_{qj} \right) \cdot \left( W_{ik}^{(l)} + F_{iu}^* R_{uv}^{(l)} F_{vk} \right) \\
&= \frac{\phi''}{m} \left[ \underbrace{W_{ij}^{(l)} W_{ik}^{(l)}}_{=T_1} + \underbrace{W_{ij}^{(l)} F_{iu}^* R_{uv}^{(l)} F_{vk}}_{=T_2} + \underbrace{F_{ip}^* R_{pq}^{(l)} F_{qj} W^{(ik)}}_{=T_3} + \underbrace{F_{ip}^* R_{pq}^{(l)} F_{qj} F_{iu}^* R_{uv}^{(l)} F_{vk}}_{=T_4} \right].
\end{aligned}
$$

$$(57)$$

Then, we can write

$$
\left\| \left[ \frac{\partial^2 \boldsymbol{\alpha}^{(l)}}{\partial \boldsymbol{\alpha}^{(l-1)^2}} \right] \right\|_{2,2,1} = \sup_{\|\boldsymbol{v}_1\|_2=1, \|\boldsymbol{v}_2\|_2=1} \sum_{i=1}^m \left| \left( \frac{\partial^2 \alpha^{(l_1)}}{\partial \alpha^{(l_1-1)^2}} \right)_{i,j,k} v_{1_j} v_{2_k} \right|.
$$

Let us now handle each of the terms separately:

$$
\begin{aligned}
\sup_{\|\boldsymbol{v}_1\|_2=1, \|\boldsymbol{v}_2\|_2=1} \sum_{i=1}^m \frac{\phi''}{m} T_{1_{ij}} v_{1_j} v_{2_k} &= \frac{\phi''}{m} \sup_{\|\boldsymbol{v}_1\|_2=1, \|\boldsymbol{v}_2\|_2=1} \sum_{i=1}^m \left| \left( W_{ij}^{(l)} v_{1_j} \right) \cdot \left( W_{ik}^{(l)} v_{1_k} \right) \right| \\
&= \frac{\phi''}{m} \sup_{\|\boldsymbol{v}_1\|_2=1, \|\boldsymbol{v}_2\|_2=1} \left| \left\langle W^{(l)} \boldsymbol{v}_1, W^{(l)} \boldsymbol{v}_2 \right\rangle \right| \\
&\leq \frac{\beta_\phi}{2m} \cdot \left( \|W^{(l)}\|_2^2 \|\boldsymbol{v}_1\|_2^2 + \|W^{(l)}\|_2^2 \|\boldsymbol{v}_2\|_2^2 \right) \\
&\leq \beta_\phi \left( \sigma_{1,w} + \frac{\rho_w}{\sqrt{m}} \right)^2 = \beta_\phi \gamma_w^2.
\end{aligned}
$$

$$(58)$$

$$
\begin{aligned}
\sup_{\|\boldsymbol{v}_1\|_2=1,\|\boldsymbol{v}_2\|_2=1} \sum_{i=1}^{m} \frac{\phi''}{m} T_{4_{ij}} v_{1_j} v_{2_k} &= \frac{\phi''}{m} \sup_{\|\boldsymbol{v}_1\|_2=1,\|\boldsymbol{v}_2\|_2=1} \sum_{i=1}^{m} \left| \left( (F^*R^{(l)}F)_{ij} v_{1_j} \right) \cdot \left( (F^*R^{(l)}F)_{ip} v_{2_p} \right) \right| \\
&= \frac{\phi''}{m} \sup_{\|\boldsymbol{v}_1\|_2=1,\|\boldsymbol{v}_2\|_2=1} \left| \left\langle (F^*R^{(l)}F)\boldsymbol{v}_1, (F^*R^{(l)}F)\boldsymbol{v}_2 \right\rangle \right| \\
&\leq \frac{\beta_\phi}{2m} \sup_{\|\boldsymbol{v}_1\|_2=1,\|\boldsymbol{v}_2\|_2=1} \left( \|F^*R^{(l)}F\|_2^2 \|\boldsymbol{v}_1\|_2^2 + \|F^*R^{(l)}F\|_2^2 \|\boldsymbol{v}_2\|_2^2 \right) \\
&\leq \frac{\beta_\phi}{m} \|F^*R^{(l)}F\|_2^2 \\
&= \frac{\beta_\phi}{m} \|R^l\|_2^2 = \beta_\phi \gamma_w^2.
\end{aligned}
\tag{59}
$$

$$
\begin{aligned}
\sup_{\|\boldsymbol{v}_1\|_2=1,\|\boldsymbol{v}_2\|_2=1} \sum_{i=1}^{m} \frac{\phi''}{m} T_{2_{ij}} v_{1_j} v_{2_k} &= \frac{\phi''}{m} \sup_{\|\boldsymbol{v}_1\|_2=1,\|\boldsymbol{v}_2\|_2=1} \sum_{i=1}^{m} \left| (W_{ij}^{(l)} v_{1_j}) \cdot (F^*R^{(l)}F)_{ip} v_{2_p} \right| \\
&= \frac{\phi''}{m} \sup_{\|\boldsymbol{v}_1\|_2=1,\|\boldsymbol{v}_2\|_2=1} \left| \left\langle W^{(l)} \boldsymbol{v}_1, F^H R^{(l)} F \boldsymbol{v}_2 \right\rangle \right| \\
&\leq \frac{\beta_\phi}{2m} \sup_{\|\boldsymbol{v}_1\|_2=1,\|\boldsymbol{v}_2\|_2=1} \left( \|W^{(l)}\|_2^2 \|\boldsymbol{v}_1\|_2^2 + \|F^*R^{(l)}F\|_2^2 \|\boldsymbol{v}_2\|_2^2 \right) \\
&\leq \frac{\beta_\phi}{2m} \left( \|W^{(l)}\|_2^2 + \|R^{(l)}\|_2^2 \right) = \frac{\beta_\phi}{2} \left( \gamma_w^2 + \gamma_r^2 \right).
\end{aligned}
\tag{60}
$$

Similarly, for the term corresponding to $T_3$ we obtain

$$
\sup_{\|\boldsymbol{v}_1\|_2=1,\|\boldsymbol{v}_2\|_2=1} \sum_{i=1}^{m} \frac{\phi''}{m} T_{3_{ij}} v_{1_j} v_{2_k} \leq \frac{\beta_\phi}{2} \left( \gamma_w^2 + \gamma_r^2 \right)
\tag{61}
$$

Putting together (58), (59), (60) and (61), we get

$$
\left\| \frac{\partial^2 \boldsymbol{\alpha}^{(l)}}{\partial \boldsymbol{\alpha}^{(l-1)^2}} \right\|_{2,2,1}^2 \leq 2\beta_\phi(\gamma_w^2 + \gamma_r^2) \leq 2(\gamma_w^2 + \gamma_r^2 + \sqrt{2}\gamma_w\gamma_r) = 2\gamma^2.
\tag{62}
$$

This completes the proof for (54). We now look at the proof for (55). First note that

$$
\begin{aligned}
\left[ \frac{\partial^2 \boldsymbol{\alpha}^{(l_2)}}{\partial \boldsymbol{\alpha}^{(l_2-1)} \partial \boldsymbol{r}_{jj'}^{(l_2)}} \right]_{i,jj'k} &= \frac{1}{m} \phi''(\tilde{\alpha}_i) \left( W_{ik}^{(l_2)} + F_{ip}^* R_{pq}^{(l_2)} F_{qk} \right) F_{ij}^* F_{j'q} \alpha_q^{(l_2-1)} + \frac{1}{\sqrt{m}} \phi'(\tilde{\alpha}_i^{(l_2)}), F_{ij}^* F_{j'k} \\
&= \underbrace{\frac{\phi''}{m} \left( W_{ik}^{(l_2)} F_{ij}^* F_{j'q} \alpha_q^{(l_2-1)} \right)}_{=T_1} + \underbrace{\frac{\phi''}{m} \left( F_{ip}^* R_{pq}^{(l_2)} F_{qk} F_{ij}^* F_{j'q} \alpha_q^{(l_2-1)} \right)}_{=T_2} + \underbrace{\frac{1}{\sqrt{m}} \phi'(\tilde{\alpha}_i^{(l_2)}), F_{ij}^* F_{j'k}}_{=T_3}.
\end{aligned}
$$

Again, we analyze each of the terms separately

$$
\begin{aligned}
\left\| T_{1_{i,jj'k}} \right\|_{2,2,1} &= \sup_{\|\boldsymbol{v}_1\|_2=1,\|V_2\|_F=1} \sum_{i=1}^{m} \left| \frac{\phi''}{m} \left( W_{ik} v_{1_k}^{(l_2)} F_{ij}^* V_{2_{jj'}}, F_{j'q} \alpha_q^{(l_2-1)} \right) \right| \\
&= \sup_{\|\boldsymbol{v}_1\|_2=1,\|V_2\|_F=1} \left| \frac{\phi''}{m} \left\langle W^{(l)} \boldsymbol{v}_1, F^* V_2 F \boldsymbol{\alpha}^{(l_2-1)} \right\rangle \right| \\
&\leq \frac{\beta_\phi}{2m} \left( \|W^{(l_2)} \boldsymbol{v}_1\|_2^2 + \|F^* V_2 F \boldsymbol{\alpha}^{(l_2-1)}\|_2^2 \right) \\
&\leq \frac{\beta_\phi}{2} \left( \gamma_w^2 + \left( \gamma^{(l_2-1)} + |\phi(0)| \sum_{i=1}^{l_2-1} \gamma^{i-1} \right)^2 \right)
\end{aligned}
\tag{63}
$$

$$\left\|T_{2_{i,jj'k}}\right\|_{2,2,1} = \sup_{\|\boldsymbol{v}_1\|_2=1,\|V_2\|_F=1} \sum_{i=1}^{m} \left| \frac{\phi''}{m} \left( F_{ip}^* R_{pq}^{(l_2)} F_{qk} v_{1_k} F_{ij}^* V_{2_{jj'}}, F_{j'q} \alpha_q^{(l_2-1)} \right) \right|$$

$$= \sup_{\|\boldsymbol{v}_1\|_2=1,\|V_2\|_F=1} \left| \frac{\phi''}{m} \left\langle F^* R^{(l)} F \boldsymbol{v}_1, F^* V_2 F \boldsymbol{\alpha}^{(l_2-1)} \right\rangle \right|$$

$$\leq \sup_{\|\boldsymbol{v}_1\|_2=1,\|V_2\|_F=1} \frac{\beta_\phi}{m} \left( \|F^* R^{(l)} F \boldsymbol{v}_1\|_2^2 + \|F^* V_2 F \boldsymbol{\alpha}^{(l_2-1)}\|_2^2 \right)$$

$$\leq \sup_{\|\boldsymbol{v}_1\|_2=1,\|V_2\|_F=1} \frac{\beta_\phi}{m} \left( \|F^* R^{(l)} F\|_2^2 \|\boldsymbol{v}_1\|_2^2 + \|F^* V_2 F\|_2^2 \|\boldsymbol{\alpha}^{(l_2-1)}\|_2^2 \right)$$

$$\overset{(a)}{\leq} \frac{\beta_\phi}{m} \left( \|R^{(l)}\|_2^2 + \|\boldsymbol{\alpha}^{(l_2-1)}\|_2^2 \right) = \frac{\beta_\phi}{2} \left( \gamma_r^2 + \left( \gamma^{(l_2-1)} + |\phi(0)| \sum_{i=1}^{l_2-1} \gamma^{i-1} \right)^2 \right)$$

$$\tag{64}$$

where $(a)$ follows, again, by exploiting the isometry of $F^*$ and $F$ wrt the $L_2$ norm, and using $\|V_2\|_2 \leq \|V_2\|_F$. Finally,

$$\left\|T_{3_{i,jj'k}}\right\|_{2,2,1} = \sup_{\|\boldsymbol{v}_1\|_2=1,\|V_2\|_F=1} \sum_{i=1}^{m} \left| \frac{\phi'}{\sqrt{m}} F_{ij}^* V_{2_{jj'}}, F_{j'k} v_{1_k} \right|$$

$$= \sup_{\|\boldsymbol{v}_1\|_2=1,\|V_2\|_F=1} \left| \frac{\phi'}{\sqrt{m}} F^* V_2 F \boldsymbol{v}_1 \right|$$

$$\leq \sup_{\|\boldsymbol{v}_1\|_2=1,\|V_2\|_F=1} \frac{1}{\sqrt{m}} \sum_{i=1}^{m} \|(F^* V_2 F)_{i,:}\|_2 \|\boldsymbol{v}_1\|$$

$$\leq \sup_{\|\boldsymbol{v}_1\|_2=1,\|V_2\|_F=1} \frac{1}{\sqrt{m}} \sum_{i=1}^{m} \|V_{2,i,:}\|_2 \leq \sup_{\|\boldsymbol{v}_1\|_2=1,\|V_2\|_F=1} \|V_2\|_F = 1,$$

$$\tag{65}$$

where we make use of the fact that $\sum_{i=1}^{m} \|\mathbf{V}_{2,i,:}\|_2 \leq \sqrt{m} \sqrt{\sum_{i=1}^{m} \|\mathbf{V}_{2,i,:}\|_2^2}$ and the isometry of $F^*$ and $F$ wrt the $L_2$ norm. Combining (63), (64) and (65), we get

$$\left\| \frac{\partial^2 \boldsymbol{\alpha}^{(l_2)}}{\partial \boldsymbol{\alpha}^{(l_2-1)} \partial \boldsymbol{r}_{jj'}^{(l_2)}} \right\|_{2,2,1} \leq \frac{\beta_\phi}{2} \left( \gamma_w^2 + \gamma_r^2 \right) + \beta_\phi \left( \gamma^{(l_2-1)} + |\phi(0)| \sum_{i=1}^{l_2-1} \gamma^{i-1} \right)^2 + 1$$

$$\leq \beta_\phi \left( \gamma^2 + (\gamma^{l_2-1} + |\phi(0)| \sum_{i=1}^{l_2-1} \gamma^{i-1}) \right) + 1.$$

$$\tag{66}$$

This completes the proof. $\qquad\qquad\square$

**Case 1.B:** $1 \leq l_2 \leq l_1 \leq L$. By building on the form of the gradient, we have

$$H_{w,r}^{(l_1,l_2)} = \frac{\partial^2 \alpha^{(l_2)}}{\partial \mathbf{w}^{(l_2)} \partial \mathbf{r}^{(l_2)}} \frac{\partial f}{\partial \alpha^{(l_2)}} \mathbf{1}_{[l_1=l_2]} + \left( \frac{\partial \alpha^{(l_2)}}{\partial \mathbf{r}^{(l_2)}} \prod_{l'=l_2+1}^{l_1-1} \frac{\partial \alpha^{(l')}}{\partial \alpha^{(l'-1)}} \right) \frac{\partial^2 \alpha^{(l_1)}}{\partial \alpha^{(l_1-1)} \partial \mathbf{w}^{(l_1)}} \left( \frac{\partial f}{\partial \alpha^{(l_1)}} \right)$$

$$+ \sum_{l=l_1+1}^{L} \left( \frac{\partial \alpha^{(l_2)}}{\partial \mathbf{r}^{(l_2)}} \prod_{l'=l_2+1}^{l-1} \frac{\partial \alpha^{(l')}}{\partial \alpha^{(l'-1)}} \right) \left( \frac{\partial \alpha^{(l_1)}}{\partial \mathbf{w}^{(l_1)}} \prod_{l'=l_1+1}^{l-1} \frac{\partial \alpha^{(l')}}{\partial \alpha^{(l'-1)}} \right) \frac{\partial^2 \alpha^{(l)}}{(\partial \alpha^{(l-1)})^2} \left( \frac{\partial f}{\partial \alpha^{(l)}} \right).$$

Then,

$$
\begin{aligned}
\|H_{w,r}^{(l_1,l_2)}\|_2 &\leq \left\|\frac{\partial^2 \alpha^{(l_2)}}{\partial \mathbf{w}^{(l_2)} \partial \mathbf{r}^{(l_2)}}\right\|_{2,2,1} \left\|\frac{\partial f}{\partial \alpha^{(l_2)}}\right\|_\infty + \left\|\frac{\partial \alpha^{(l_2)}}{\partial \mathbf{r}^{(l_2)}}\right\|_2 \prod_{l'=l_2+1}^{l_1-1} \left\|\frac{\partial \alpha^{(l')}}{\partial \alpha^{(l'-1)}}\right\|_2 \left\|\frac{\partial^2 \alpha^{(l_1)}}{\partial \alpha^{(l_1-1)} \partial \mathbf{w}^{(l_1)}}\right\|_{2,2,1} \left\|\frac{\partial f}{\partial \alpha^{(l_1)}}\right\|_\infty \\
&\quad + \sum_{l=l_1+1}^{L} \left(\left\|\frac{\partial \alpha^{(l_2)}}{\partial \mathbf{r}^{(l_2)}}\right\|_2 \prod_{l'=l_2+1}^{l-1} \left\|\frac{\partial \alpha^{(l')}}{\partial \alpha^{(l'-1)}}\right\|_2\right) \left(\left\|\frac{\partial \alpha^{(l_1)}}{\partial \mathbf{w}^{(l_1)}}\right\|_2 \prod_{l'=l_1+1}^{l-1} \left\|\frac{\partial \alpha^{(l')}}{\partial \alpha^{(l'-1)}}\right\|_2\right) \left\|\frac{\partial^2 \alpha^{(l)}}{(\partial \alpha^{(l-1)})^2}\right\|_{2,2,1} \left\|\frac{\partial f}{\partial \alpha^{(l)}}\right\|_\infty \\
&\leq \left\|\frac{\partial^2 \alpha^{(l_2)}}{\partial \mathbf{w}^{(l_2)} \partial \mathbf{r}^{(l_2)}}\right\|_{2,2,1} \left\|\frac{\partial f}{\partial \alpha^{(l_2)}}\right\|_\infty + L_\phi^{l_2-l_1-1} \left\|\frac{\partial \alpha^{(l_2)}}{\partial \mathbf{r}^{(l_2)}}\right\|_2 \left\|\frac{\partial^2 \alpha^{(l_1)}}{\partial \alpha^{(l_1-1)} \partial \mathbf{w}^{(l_1)}}\right\|_{2,2,1} \left\|\frac{\partial f}{\partial \alpha^{(l_1)}}\right\|_\infty \\
&\quad + \sum_{l=l_2+1}^{L} L_\phi^{2l-l_2-l_1} \left\|\frac{\partial \alpha^{(l_2)}}{\partial \mathbf{r}^{(l_2)}}\right\|_2 \left\|\frac{\partial \alpha^{(l_1)}}{\partial \mathbf{r}^{(l_1)}}\right\|_2 \left\|\frac{\partial^2 \alpha^{(l)}}{(\partial \alpha^{(l-1)})^2}\right\|_{2,2,1} \left\|\frac{\partial f}{\partial \alpha^{(l)}}\right\|_\infty .
\end{aligned}
$$

**Lemma E.6.** *Under Assumptions 3 and 4, with $\boldsymbol{\theta} \in B_\rho^{\mathrm{Euc}}(\boldsymbol{\theta}_0)$, we have with high probability*

$$
\left\|\frac{\partial^2 \boldsymbol{\alpha}^{(l_2)}}{\partial \mathbf{w}^{(l_2)} \partial \boldsymbol{r}^{(l_2)}}\right\|_{2,2,1}^2 \leq \beta_\phi \left(\gamma^{l_2-1} + |\phi(0)| \sum_{i=1}^{l_2-1} \gamma^{i-1}\right)^2 \tag{67}
$$

$$
\left\|\frac{\partial^2 \boldsymbol{\alpha}^{(l_1)}}{\partial \boldsymbol{\alpha}^{(l_1-1)} \partial \mathbf{w}^{(l_1)}}\right\|_{2,2,1}^2 \leq \beta_\phi \left(\gamma^2 + (\gamma^{l_1-1} + |\phi(0)| \sum_{i=1}^{l_1-1} \gamma^{i-1})\right) + 1. \tag{68}
$$

*Proof.* The proof of (67) follows in a manner similar to the proof of (53). Next, for proving (68) consider the following

$$
\left[\frac{\partial \boldsymbol{\alpha}^{(l_1)}}{\partial \boldsymbol{\alpha}^{(l_1-1)} \partial \boldsymbol{w}^{(l_1)}}\right]_{i,jj'k} = \left(\underbrace{\frac{\phi''(\widetilde{\alpha}_i^{(l_1)})}{m} W_{ik}^{(l_1)} \alpha_{j'}^{(l_1)} \delta_{ij}}_{=T_1} + \underbrace{\frac{\phi''(\widetilde{\alpha}_i^{(l_1)})}{m} F_{ip}^* R_{pq}^{(l_1)} F_{qk} \alpha_{j'}^{(l_1)} \delta_{ij}}_{=T_2}\right) + \underbrace{\frac{1}{\sqrt{m}} \phi'(\alpha^{(l_1)}) \delta_{ij} \delta_{kj'}}_{=T_3}
$$

Then analyzing each term separately, we get

$$
\begin{aligned}
\left\|T_{1_{i,jj'k}}\right\|_{2,2,1} &= \sup_{\|\boldsymbol{v}_1\|_2=1, \|V_2\|_F=1} \sum_{i=1}^{m} \left|\frac{\phi''}{m} W_{ik}^{(l_1)} v_{1_k} V_{2_{ij}} \alpha_j^{(l_1-1)}\right| \\
&= \sup_{\|\boldsymbol{v}_1\|_2=1, \|V_2\|_F=1} \left|\frac{\phi''}{m} \left\langle W^{(l_1)} \boldsymbol{\alpha}^{(l_1)}, V_2 \boldsymbol{\alpha}^{(l_1-1)}\right\rangle\right| \\
&\leq \sup_{\|\boldsymbol{v}_1\|_2=1, \|V_2\|_F=1} \frac{\beta_\phi}{2m} \left(\|W^{(l_1)}\|_2^2 \|\boldsymbol{v}_1\|_2^2 + \|V_2\|_2^2 \|\boldsymbol{\alpha}^{(l_1-1)}\|_2^2\right) \\
&\leq \frac{\beta_\phi}{2m} \left(\|W^{(l_1)}\|_2^2 + \|\boldsymbol{\alpha}^{(l_1-1)}\|_2^2\right) = \frac{\beta_\phi}{2} \left(\gamma_w^2 + \left(\gamma^{l_1-1} + |\phi(0)| \sum_{i=1}^{l_1-1} \gamma^{i-1}\right)^2\right),
\end{aligned} \tag{69}
$$

$$
\begin{aligned}
\left\| T_{2_{i,jj'k}} \right\|_{2,2,1} &= \sup_{\|\boldsymbol{v}_1\|_2=1, \|V_2\|_F=1} \sum_{i=1}^{m} \left| \frac{\phi''}{m} F_{ip}^* R_{pq}^{(l_1)} F_{qk} v_{1_k} V_{2_{ij'}} \alpha_{j'}^{(l_1-1)} \right| \\
&= \sup_{\|\boldsymbol{v}_1\|_2=1, \|V_2\|_F=1} \left| \frac{\phi''}{m} \left\langle F^* R^{(l_1)} F \boldsymbol{v}_1, V_2 \boldsymbol{\alpha}^{(l_1-1)} \right\rangle \right| \\
&\le \sup_{\|\boldsymbol{v}_1\|_2=1, \|V_2\|_F=1} \frac{\beta_\phi}{2m} \left( \|F^* R^{(l_1)} F \boldsymbol{v}_1\|_2^2 + \|V_2 \boldsymbol{\alpha}^{(l_1-1)}\|_2^2 \right) \\
&\le \sup_{\|\boldsymbol{v}_1\|_2=1, \|V_2\|_F=1} \frac{\beta_\phi}{2m} \left( \|F^* R^{(l_1)} F \boldsymbol{v}_1\|_2^2 + \|V_2 \boldsymbol{\alpha}^{(l_1-1)}\|_2^2 \right) \\
&\le \sup_{\|\boldsymbol{v}_1\|_2=1, \|V_2\|_F=1} \frac{\beta_\phi}{2m} \left( \|F^* R^{(l_1)} F \boldsymbol{v}_1\|_2^2 + \|V_2 \boldsymbol{\alpha}^{(l_1-1)}\|_2^2 \right) \\
&\le \frac{\beta_\phi}{2} \left( \gamma_r^2 + \left( \gamma^{l_1-1} + |\phi(0)| \sum_{i=1}^{l_1-1} \gamma^{i-1} \right)^2 \right),
\end{aligned}
\tag{70}
$$

and, finally,

$$
\begin{aligned}
\left\| T_{3_{i,jj'k}} \right\|_{2,2,1} &= \sup_{\|\boldsymbol{v}_1\|_2=1, \|V_2\|_F=1} \sum_{i=1}^{m} \left| \frac{\phi'}{\sqrt{m}} V_{2_{ik}} v_{1_k} \right| \\
&\le \sup_{\|\boldsymbol{v}_1\|_2=1, \|V_2\|_F=1} \sum_{i=1}^{m} \frac{1}{\sqrt{m}} \|\boldsymbol{v}_1\|_2 \|V_{2,i,:}\|_2 \\
&\le \|V_2\|_F = 1.
\end{aligned}
\tag{71}
$$

Hence, we have

$$
\begin{aligned}
\left\| \frac{\partial^2 \boldsymbol{\alpha}^{(l_1)}}{\partial \boldsymbol{\alpha}^{(l_1-1)} \partial \mathbf{w}^{(l_1)}} \right\|_{2,2,1}^2 &\le \frac{\beta_\phi}{2} \left( \gamma_w^2 + \gamma_r^2 \right) + \beta_\phi \left( \gamma^{l_1-1} + |\phi(0)| \sum_{i=1}^{l_1-1} \gamma^{i-1} \right)^2 \\
&\le \beta_\phi \left( \gamma^2 + \gamma^{l_1-1} + |\phi(0)| \sum_{i=1}^{l_1-1} \gamma^{i-1} \right)^2.
\end{aligned}
\tag{72}
$$

This completes the proof of (68). $\qquad\square$

Finally, we remark that the analysis for the diagonal blocks $H_w^{(l_1,l_2)}$ and $H_r^{(l_1,l_2)}$ follows directly from (Banerjee et al., 2023). The interested reader is referred to Theorem 4.2 (and equivalently Theorem 3.1 in (Liu et al., 2021a)).

Then, based on the definitions in (52), we have

$$
\|H_{w,r}^{(l_1,l_2)}\|_2 \le C_1' \mathcal{Q}_{2,2,1}^{w,r}(f) \mathcal{Q}_\infty^{w,r}(f),
$$

where $C_1' = \cdots$. Here

**Case 2.A:** $1 \le l_1 \le L+1$. For Hessian terms involving $(v, w)$, since $\frac{\partial f}{\partial \mathbf{v}} = \frac{1}{\sqrt{m}} \alpha^{(L+1)}$, we have

$$
H_{v,w}^{(l_1)} = \frac{1}{\sqrt{m}} \frac{\partial \alpha^{(L+1)}}{\partial \mathbf{w}^{(l_1)}} = \frac{1}{\sqrt{m}} \left( \frac{\partial \alpha^{(l_1)}}{\partial \mathbf{w}^{(l_1)}} \prod_{l'=l_1+1}^{L+1} \frac{\partial \alpha^{(l')}}{\partial \alpha^{(l'-1)}} \right).
$$

Then,

$$
\|H_{w,v}^{(l_1,L+1)}\|_2 \le \frac{1}{\sqrt{m}} \left\| \frac{\partial \alpha^{(l_1)}}{\partial \mathbf{w}^{(l_1)}} \right\|_2 \prod_{l'=l_1+1}^{L+1} \left\| \frac{\partial \alpha^{(l')}}{\partial \alpha^{(l'-1)}} \right\|_2 \le \frac{1}{\sqrt{m}} L_\phi^L \mathcal{Q}_2^{(w,r)}(f).
$$

**Case 2.B:** $2 \le l_2 \le L+1$. For Hessian terms involving $(v, r)$, since $\frac{\partial f}{\partial \mathbf{v}} = \frac{1}{\sqrt{m}} \alpha^{(L+1)}$, we have

$$
H_{v,r}^{(l_2)} = \frac{1}{\sqrt{m}} \frac{\partial \alpha^{(L+1)}}{\partial \mathbf{r}^{(l_2)}} = \frac{1}{\sqrt{m}} \left( \frac{\partial \alpha^{(l_2)}}{\partial \mathbf{r}^{(l_2)}} \prod_{l'=l_2+1}^{L+1} \frac{\partial \alpha^{(l')}}{\partial \alpha^{(l'-1)}} \right).
$$

Then,

$$\|H_{v,r}^{(l_2)}\|_2 \leq \frac{1}{\sqrt{m}} \left\|\frac{\partial \alpha^{(l_2)}}{\partial \mathbf{r}^{(l_2)}}\right\|_2 \prod_{l'=l_2+1}^{L+1} \left\|\frac{\partial \alpha^{(l')}}{\partial \alpha^{(l'-1)}}\right\|_2 \leq \frac{1}{\sqrt{m}} L_\phi^L \mathcal{Q}_2^{(w,r)}(f) \,.$$

Thus, the hessian of the FNO predictor is bounded.

