# OpenReview forum: "Optimization for Neural Operator Learning: Wider Networks are Better"
_ICLR.cc/2024/Conference — ICLR 2024 Conference Withdrawn Submission_

### Official Review · Reviewer_JAvK · 2023-10-29

**Soundness:** 4 excellent
**Presentation:** 3 good
**Contribution:** 3 good
**Rating:** 3
**Confidence:** 5

**Summary:**

The paper theoretically analyzes the convergence of two representative neural operators under gradient descent algorithms, demonstrating that under the Restricted Strong Convexity condition, gradient descent can globally reduce loss. Overall, the theoretical results presented in this paper are very clear. However, my primary concern lies in the technical contributions of the paper, as it appears the authors did not mention where the main technical challenges lie compared to the training of ordinary feedforward neural networks. I believe the paper should also emphasize what the core technical innovations are. In conclusion, I think there is room for improvement in this paper, at least in terms of writing and experimental validation.

**Strengths:**

The main contribution of this paper lies in deriving the convergence of two representative neural operators, and elucidating that utilizing wider networks yields better results.

**Weaknesses:**

1. Although the writing in this paper is very concise and clear, it lacks professionalism, especially in highlighting its technical contributions and innovations. For instance, compared to ordinary Feedforward Neural Networks (FNNs), what are the significant challenges in error estimation for neural operators, and what new techniques are necessary to derive their convergence bound? Moreover, the better performance of wider and deeper neural networks in fitting data is already a well-acknowledged fact in the field, so deriving this conclusion cannot be considered a major contribution of this paper.
2. The paper derives the convergence rate for two neural operators (FNO and DeepONet) but does not provide a comparison between them, nor does it elaborate on their differences and connections. Additionally, there is no discussion on whether the derivation method proposed in this paper can be applied to or how it might be improved for neural operators beyond the aforementioned two structures.
3. The experimental section could benefit from additional validation. For instance, for the 2-dimensional Darcy flow problem and the 2D time-dependent Navier-Stokes (NS) equations, which are commonly used datasets, it is imperative that the authors also conduct experimental validation on them. Furthermore, since the data used for neural operators adhere to PDEs, verifying which datasets satisfy the assumptions made in the paper would also be valuable.
4. ")" is missing in the title of section 3.2.

**Questions:**

None

---

### Official Review · Reviewer_xbcP · 2023-10-30

**Soundness:** 1 poor
**Presentation:** 3 good
**Contribution:** 1 poor
**Rating:** 3
**Confidence:** 3

**Summary:**

This paper attempts to explain why gradient based optimization works for training DeepONets.

But the key claim being being seemingly made does not seem backed up with a proof.
The crux of the paper seems misleading.

**Strengths:**

The paper succeeds in measuring the constants of RSC and the smoothness property for standard operator nets.
This part of the paper is clearly a significant effort by the authors.

**Weaknesses:**

In Section 3.1, It seems from the notation of branch and trunk net that $\theta_f$ and $\theta_g$  are shared across all branch and trunk nets respectively, or in other words $\theta_f$ and $\theta_g$ are same for all values of $k$. This does not seem to at all be consistent with how DeepONets work - where usually at each $k$ its a different set of parameters - albeit with an overlap. So the very premise of the paper seems suspect!

Also in Section 3.1, the bold facing of $f, g, u$ and $x$ seem to be inconsistent.
For eg. $f_k$ is not bold in the first paragraph, but it is bold in the second.
This creates sufficient confusion with regards to following the paper.

In Section 4, Theorem 1 is almost identical to Theorem 5.3 in the Banerjee.et.al (2023).
It is not made clear in the writing if this theorem is exactly the same or if it is different, then how exactly does it differ from the other.

In Section 5, Definition 2, $f_{k}^{(i)}$ and $g_{k,j}^{(i)}$ are not defined anywhere.

The $t-$dependence of the potential contraction factor, $\frac{\alpha_t \omega_t (1-\gamma_t)}{\beta} (2 - \omega_t)$, in equation $9$ raises questions about why this should result in geometric convergence. Infact, it is not even transparent that equation 9 is a contraction.

**Questions:**

Does the factor $1 - \frac{\alpha_t \omega_t (1-\gamma_t)}{\beta} (2 - \omega_t)$, have a $t-$independent lower bound in the interval $(0,1)$ to ensure a minimum contraction in $L(\theta_{t+1})$ for every $t$ ? If not then how is any convergence getting guaranteed here?

---

### Official Review · Reviewer_MTc6 · 2023-11-01

**Soundness:** 3 good
**Presentation:** 2 fair
**Contribution:** 2 fair
**Rating:** 6
**Confidence:** 3

**Summary:**

The paper presents convergence guarantees for neural operator learning, which are shown to hold when assuming restricted strong convexity and smoothness of the loss. Two operator learning settings (DONs and FNOs) are shown to satisfy the conditions and it is concluded from the theoretical and experimental results that wider networks exhibit better optimisation results.

**Strengths:**

- The paper is well-structured and the motivations are clear.
- The assumptions seem reasonable and the theoretical results admit interesting interpretations.

**Weaknesses:**

The theoretical results admit interesting conclusions, such as wider networks perform better, but it is unclear to what extent this is specific for neural operators since similar results exist for feedforward networks.

**Questions:**

- The paper states that a takeaway of the theoretical results is that 'wider networks lead to better optimization convergence'. For both the FNO and DON the RSC property (one of the two sufficient conditions for convergence) holds whenever the predicted gradient is bounded ($\| \nabla_{\theta} \bar{G}_t \|^2 = \Theta (\frac{1}{\sqrt{m}})$) and RSC holds then in probability of at least $1 - \frac{4 L}{m}$. So it appears that wide networks are not only beneficial for convergence, but are actually required for the results to hold? I'd suggest to make this more explicit in the beginnig / abstract.
- As stated in the paper, the results are inspired by the work of Banerjee et al. (2023). There, a similar analysis is performed for feed forward networks, and for that setting the conditions for RSC is similarly $\| \nabla_{\theta} \bar{G}_t \|^2 = \Theta (\frac{poly(L)}{\sqrt{m}})$. Does that also yield the interpretation of 'wider networks are better', and if so, to what extent do the the results differ for the two settings (feedforward networks, neural operators)? Did the authors perform experiments with respect to the depth of the networks?
- In Condition 2 (smoothness) it might help the reader to mention that it will be shown in probability. Also, what is the set $\mathcal{N}(\Theta)$?
- The plots (Figure 1, 2) would benefit from concise (and consistent) titles and labels. What is the meaning of epochs%100?

---

### Official Review · Reviewer_15jL · 2023-11-03

**Soundness:** 4 excellent
**Presentation:** 3 good
**Contribution:** 1 poor
**Rating:** 5
**Confidence:** 5

**Summary:**

This paper presented an optimization analysis of operator networks and showed that the operator networks satisfy the RSC condition.  The authors show that the wider network makes it easier to satisfy the RSC condition.

**Strengths:**

This paper presents a solid optimization analysis for operator networks

**Weaknesses:**

The main concern of the reviewer is novelty, the analysis in the paper is no different from recent optimization analysis of neural networks. The author also still conjectures that wider is better can be an overclaim.
- Why dealing with operator networks is different from neural networks?
- From my understanding, the initialization makes the networks lie in the lazy regime. Is this an interesting regime for analyzing? What if I change the initialization scheme, and the wider the better still holds?
- The lazy regime is not learning a feature, is there a possibility that a finite/smaller width operator network can enforce feature learning.
- Is the wider the better an overclaim? For this paper don't present any result that the narrow network is provably hard (ie lower bound)

Most important, the reviewer can't find any difference between the ananlysis of operator learning and neural network and the conclusion differs from each other. The reviewer highly questions the novelty of the paper.

**Questions:**

See above